# `Polaris`: Coupled Orbital Polar Embeddings for Hierarchical Concept Learning

**Sahil Mishra** [* 1]  **Srinitish Srinivasan** [* 1]  **Sourish Dasgupta** [2]  **Tanmoy Chakraborty** [1 3]

## Abstract

Real-world knowledge is often organized as hierarchies such as product taxonomies, medical ontologies, and label trees, yet learning hierarchical representations is challenging due to asymmetric structure and noisy semantics. We introduce `Polaris`, a polar hyperspherical embedding framework that separates semanticity from hierarchy using angular geometry and radius, enabling the learning of meaning and structure without interference. To map a latent representation onto the sphere, we project it to the tangent space at the North pole, apply the exponential map, and learn unit-norm representations using spherical linear layers. `Polaris` then combines robust local constraints, global regularization that prevents geometric collapse, and uncertainty-aware asymmetric objectives that encourage directional containment. At inference time, `Polaris` uses structure-guided retrieval to efficiently narrow down candidate parents before final ranking. We evaluate `Polaris` on different settings of taxonomy expansion – spanning trees, multi-parent DAGs, and multimodal hierarchies, showing consistent improvements of up to $\sim$19 points in top-$K$ retrieval and up to $\sim$60% reduction in mean rank over fourteen strong baselines.

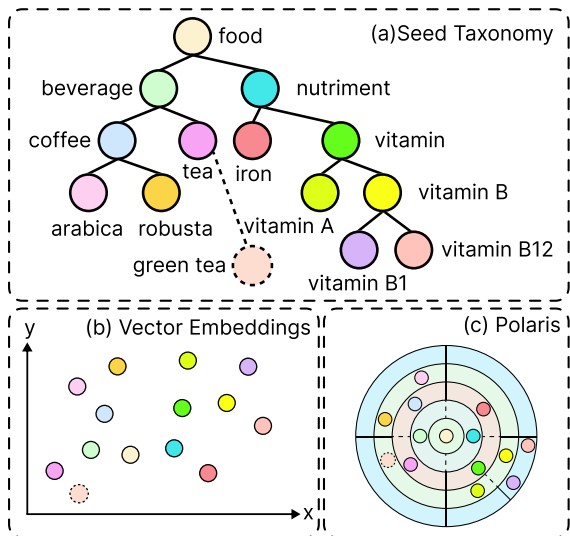

*Figure 1.* An overview of hierarchical structure and the comparison of `Polaris` with vector embeddings.

## 1. Introduction and Prior Works

In many real-world systems, knowledge is organized into hierarchies, where broad concepts are gradually refined into more specific ones. Specifically, in today's web-scale systems, these hierarchies provide a compact, reusable backbone for many applications, including query understanding (Wu et al., 2012), content browsing (Yang, 2012), personalized recommendation (Zhang et al., 2014; Tan et al., 2022), and web search (Gabrilovich et al., 2009). For example, large online retailers such as Amazon (Mao et al., 2020; Cheng et al., 2024) and eBay (Shen et al., 2012) organize products into categories at different levels of detail, so customers can quickly navigate from broad departments to fine-grained types when searching for an item (Mahabal et al., 2023). Similarly, search engines can use category structure to better interpret user intent and improve the quality of retrieval and ranking (Ding et al., 2013).

These hierarchical knowledge systems are mostly curated by human experts or crowdsourced (Vrandečić & Krötzsch, 2014; Lipscomb, 2000). A core requirement in these settings is to represent *asymmetric* relations: a child should be semantically compatible with its parent while also being *contained* by it in the hierarchy (Sastry et al., 2025). Such manual curations are time-consuming, labor-intensive, and rarely complete. New products, entities, and concepts appear continuously, creating a persistent need for methods to expand hierarchies so new concepts can be added to

---
[*]Equal contribution  [1]Indian Institute of Technology Delhi, New Delhi, India [2]KDM Lab, Dhirubhai Ambani University Gandhinagar, Gujarat, India [3]Indian Institute of Technology Delhi, Abu Dhabi, UAE. Correspondence to: Sahil Mishra <sahil.mishra@ee.iitd.ac.in>.

*Proceedings of the $43^{rd}$ International Conference on Machine Learning*, Seoul, South Korea. PMLR 306, 2026. Copyright 2026 by the author(s).

an existing hierarchy with the correct parent relationships, redefining the task as *taxonomy expansion*. To scale this process, we need methods that can learn the structural patterns already present in the seed hierarchy or taxonomy and use them to place new nodes automatically, with minimal human intervention.

**Prior studies and limitations.** Many taxonomy expansion methods embed each concept as a single point in Euclidean space and score parent-child links via symmetric distance or dot-product similarity (Shen et al., 2020b; Jiang et al., 2022b; Yu et al., 2020a), which mismatches the inherently asymmetric nature of hierarchies. Hyperbolic embeddings address this by exploiting exponential volume growth with radius, providing high capacity at deeper levels and alleviating crowding in tree-like structures (Nickel & Kiela, 2017; Xu et al., 2022; Ma et al., 2021a). Cone-based order embeddings instead encode asymmetry explicitly via directional containment, requiring a child to lie within its parent's entailment cone (Ganea et al., 2018). Related containment-based approaches use regions such as boxes (Li et al., 2019; Vilnis et al., 2018; Jiang et al., 2023) and probabilistic variants like Gaussian boxes (Mishra et al., 2026b). Despite making partial order more direct, these methods can still struggle in practice because taxonomy expansion must jointly balance semantic relatedness with correct hierarchical level; under noisy descriptions or non-tree structure, this coupling can destabilize optimization and amplify small semantic errors into large placement mistakes.

**Need for polar representation.** Polar embeddings are appealing for hierarchies because they can decouple semantic direction from hierarchical level, but they remain underused in practice due to unstable angular optimization. Prior polar methods often rely on ad hoc stabilizers such as modulus-based angle wrapping (Iwamoto et al., 2021; Zhang et al., 2020a), sector-based partitions with specialized losses (Zhu & Zeng, 2025), or sigmoid rescaling to restrict angles (Atzeni et al., 2023), which can compromise manifold continuity and yield inconsistent geometry. These issues are amplified under weak supervision common in real taxonomies, where noise and incompleteness manifest as high-variance angular drift, hindering reliable separation of semantics from depth. More related work is discussed in Appendix G.

**Our proposed method.** This motivates a representation that separates these two signals instead of forcing them into the same geometry. We propose `Polaris`, a polar hyperspherical framework that decouples semantic similarity from hierarchical position by modeling meaning as direction on the unit sphere and hierarchical level as a separate orbital potential derived from the existing hierarchy. To map pretrained encoder features onto the sphere without breaking continuity, we project them to the tangent space at the North

pole, apply the exponential map, and learn unit-norm representations using spherical linear layers. `Polaris` then combines robust local training objectives, global regularization that prevents geometric collapse, and uncertainty-aware asymmetric learning to better handle noisy semantics during expansion. At inference time, `Polaris` uses the orbital potential to prune candidates across levels before final angular ranking, making expansion both efficient and accurate in large hierarchies.

`Polaris` consists of four components. First, we introduce manifold-consistent spherical encoding. Starting from an existing latent representation, we map each node to the hypersphere by projecting its representation to the tangent space at the North pole and applying the Riemannian exponential map. We then enforce unit-norm features using spherical linear layers with explicit re-projection, so all transformations remain closed on the hyperspherical manifold, and avoid discontinuous tricks such as angle wrapping or modulus constraints. Second, `Polaris` performs robust local structure learning using a continuous geodesic triplet objective on intrinsic angular distance, which learns parent–child relations under noisy semantics without relying on multiple hard sector losses to define angular bounds. Third, to prevent high-dimensional collapse, such as equator concentration, we add global geometric regularization via an anisotropic spherical SVGD term with a von Mises–Fisher kernel and a pole-favoring score, encouraging stable, well-spread representations. Fourth, `Polaris` is uncertainty-aware, modeling each node as a vMF distribution and optimizing an asymmetric vMF–KL objective that keeps parent distributions directionally aligned while ensuring they are higher-entropy than their children, providing a principled notion of containment under ambiguity. Finally, during inference, `Polaris` performs coupled orbital retrieval, where the hierarchy-derived orbital potential defines a dynamic cosine decision boundary to gate candidates level-wise before angular re-ranking, reducing the search space while preserving accuracy and aligning retrieval with hierarchical organization. Experimental results show that `Polaris` achieves consistent gains across single-parent, multi-parent, and multimodal hierarchies. On benchmarks, it improves top-$K$ recovery by up to ∼19 points, while reducing mean rank by up to ∼60%. Ablations further verify the need of objectives and design choices.

Our contributions are summarized as follows.[1]

- We propose `Polaris`, a polar hyperspherical framework for hierarchy expansion that decouples semantic similarity (direction) from hierarchical position (orbital potential) while remaining stable under noisy semantics.
- We develop a four-component framework to separate

---

[1]Code: https://github.com/sahilmishra0012/Polaris

asymmetry and hierarchy — manifold-consistent spherical encoding, robust local structure learning, global geometric regularization to prevent collapse, and uncertainty-aware asymmetric learning. At inference time, coupled orbital retrieval gates candidates across levels before angular re-ranking to efficiently place them.

- We evaluate `Polaris` on hierarchy expansion benchmarks – single-parent trees, multi-parent DAGs, and multimodal hierarchies, and obtain consistent gains of 9, 6, 4 points, respectively, over baselines across all benchmarks. Our results are supplemented by detailed ablations to quantify the contribution of individual components.

## 2. `Polaris`

### 2.1. The Unit Hyperspherical Manifold

**Embedding space.** We model each concept as a point on the unit hypersphere $\mathcal{M} = \mathbb{S}^{d-1} = \{\mathbf{z} \in \mathbb{R}^d : \|\mathbf{z}\|_2 = 1\}$, equipped with the canonical round metric induced by the Euclidean inner product. To strictly enforce this topological constraint during learning, we apply a differentiable $\ell_2$-normalization operator $\phi(\mathbf{x}) = \mathbf{x}/(\|\mathbf{x}\|_2 + \epsilon)$ with a small $\epsilon > 0$, to all embedding vectors and projection weights.

**Proposition 2.1** (Geodesic distance on $\mathbb{S}^{d-1}$). *Let* $\mathcal{M} = (\mathbb{S}^{d-1}, g)$ *be the Riemannian manifold where g denotes the canonical round metric induced by the Euclidean inner product* $\langle \cdot, \cdot \rangle_{\mathbb{R}^d}$. *For any* $\mathbf{z}_i, \mathbf{z}_j \in \mathbb{S}^{d-1}$, *the geodesic distance under the round metric represents the length of the shortest path (great circle arc) between these points and is given as,*

$$d_{\mathcal{M}}(\mathbf{z}_i, \mathbf{z}_j) = \arccos\big(\langle \mathbf{z}_i, \mathbf{z}_j \rangle\big) \in [0, \pi]. \qquad (1)$$

The proof is provided in Appendix J.1. As $\arccos(\cdot)$ is a smooth, strictly decreasing bijection on $[-1, 1]$, minimizing the geodesic distance $d_{\mathcal{M}}(\mathbf{z}_i, \mathbf{z}_j)$ is equivalent in *ordering* to maximizing the inner product $\langle \mathbf{z}_i, \mathbf{z}_j \rangle$ for unit-norm embeddings. This lets us optimize with inner products while still respecting intrinsic spherical geometry, serving as a valid and exact surrogate for optimizing the intrinsic Riemannian distance on the manifold. In contrast, explicitly parameterizing polar angles $(\theta, \psi)$ and enforcing periodicity via discontinuous modular constraints such as $\theta \leftarrow \theta \bmod 2\pi$ introduces discontinuities and coordinate singularities, resulting in a topological mismatch (i.e., they implicitly model the embedding space as a flat cylindrical manifold with zero Gaussian curvature, rather than a constant-curvature hypersphere) that can destabilize optimization and break manifold-consistent learning. This leads to distorted geodesic metrics and optimization instability, analyzed in Appendix I.

### 2.2. Projection onto the Unit Hypersphere

We map latent representations in Euclidean space onto the unit hypersphere $\mathbb{S}^{d-1}$ using tangent-space projection at the North pole of the sphere, followed by the exponential mapping to the unit hypersphere along the geodesic path. Given a concept $l$ (text or image) and its latent representation $\mathbf{e} \in \mathbb{R}^{d_{\text{plm}}}$, we project $\mathbf{e} \in \mathbb{R}^{d_{\text{plm}}}$ to tangent space $T_{\mathbf{p}_N} \mathbb{S}^{d-1}$,

$$\mathbf{v} = \text{proj}_{\mathbf{p}_N}(\mathbf{e}) = \mathbf{e} - \langle \mathbf{e}, \mathbf{p}_N \rangle \mathbf{p}_N, \qquad \mathbf{v} \in T_{\mathbf{p}_N} \mathbb{S}^{d-1}, \tag{2}$$

where $\mathbf{p}_N \in \mathbb{S}^{d-1}$ is the basis vector $[0, ..., 0, 1]^{\mathsf{T}}$, representing the North pole. Further, we map it onto the sphere via the exponential map,

$$\mathbf{z}_0 = \exp_{\mathbf{p}_N}(\mathbf{v}) = \cos(\|\mathbf{v}\|_2)\, \mathbf{p}_N + \sin(\|\mathbf{v}\|_2)\, \frac{\mathbf{v}}{\|\mathbf{v}\|_2 + \epsilon}. \tag{3}$$

**Spherical linear layers.** Standard affine transformations do not preserve the unit-norm constraint of the hypersphere, and the addition of a bias term shifts the representation space away from the origin. We, therefore, use spherical linear layers with three modifications – **(a) Manifold Parameter Constraint.** Each row $\mathbf{w}_i$ of $\mathbf{W} \in \mathbb{R}^{d_{\text{out}} \times d_{\text{in}}}$ is constrained to $\|\mathbf{w}_i\|_2 = 1$ by $\ell_2$-normalizing rows at initialization and after each Riemannian gradient update. **(b) Bias Elimination.** We remove the bias term to avoid translations that are incompatible with origin-centered hyperspherical geometry. **(c) Output re-projection.** We re-project outputs to ensure closure on the manifold as, $\mathbf{y} = \frac{\mathbf{W}\mathbf{x}}{\|\mathbf{W}\mathbf{x}\|_2 + \epsilon}$.

From the final spherical layer, we obtain the geometric embedding $\mathbf{z} = f_{\text{sphere}}(\mathbf{z}_0)$, where $f_{\text{sphere}}$ is a neural network composed of spherical linear layers.

### 2.3. Learning on the Hypersphere

We learn embeddings directly in Cartesian coordinates under the constraint $\mathbf{z} \in \mathbb{S}^{d-1}$, without explicitly optimizing angular coordinates.

#### 2.3.1. GEOMETRIC LEARNING

We learn local parent–child structure using a robust metric-learning objective on the hypersphere. Given unit-norm embeddings $\mathbf{z} \in \mathbb{S}^{d-1}$, we measure dissimilarity by the intrinsic geodesic angle $\theta_{ij} = \arccos(\langle \mathbf{z}_i, \mathbf{z}_j \rangle)$. Because this angle depends only on inner products, any objective built from $\theta_{ij}$ should be independent of the global coordinate system. We state this invariance explicitly, since it clarifies that our loss captures *relative* hierarchical geometry rather than an arbitrary choice of orientation on the sphere.

**Definition 2.2** (Rotational Invariance of the Geodesic Welsch Loss). *Let* $\theta_{ij} = \arccos(\langle \mathbf{z}_i, \mathbf{z}_j \rangle)$ *be the geodesic angle between two concept embeddings* $\mathbf{z}_i, \mathbf{z}_j \in \mathbb{S}^{d-1}$. *Let the objective function be the Welsch M-estimator applied to this angular distance:*

$$\mathcal{L}_{\text{Welsch}}(\mathbf{z}_i, \mathbf{z}_j) = 1 - \exp\left(-\frac{\theta_{ij}^2}{2\sigma^2}\right), \tag{4}$$

where $\sigma > 0$ is a kernel width parameter. This loss function is invariant under the diagonal action of the Special Orthogonal Group $\mathsf{SO}(d)$. That is, for any rotation $\mathbf{R} \in \mathsf{SO}(d)$:

$$\mathcal{L}_{\text{Welsch}}(\mathbf{R}\mathbf{z}_i, \mathbf{R}\mathbf{z}_j) = \mathcal{L}_{\text{Welsch}}(\mathbf{z}_i, \mathbf{z}_j). \tag{5}$$

Definition 2.2 implies that the loss depends only on relative geometry on $\mathbb{S}^{d-1}$, and therefore does not privilege any particular global axis or coordinate chart. This is important in our setting because the sphere admits many equivalent parameterizations, and orientation-specific objectives can introduce spurious biases.

**Robust geodesic triplet objective.** We learn local hierarchy using a rotation-invariant objective on the unit hypersphere. Each concept is represented by a unit-norm embedding $\mathbf{z} \in \mathbb{S}^{d-1}$, and we measure semantic dissimilarity using the intrinsic geodesic angle $\theta$ between embeddings. To make training robust to noisy semantics and outliers, we pass this angular distance through a bounded Welsch M-estimator. Concretely, we encourage true parent–child pairs to have small Welsch-penalized angles and push the child away from negative candidates, so that intrinsic spherical distance aligns with hierarchical compatibility. Formally, for a triplet of embeddings $\mathbf{z}_p, \mathbf{z}_c, \mathbf{z}_n \subset \mathbb{S}^{d-1}$, we define the geometric learning objective as follows:

$$\theta_{ij} = \arccos(\langle \mathbf{z}_i, \mathbf{z}_j \rangle) \tag{6a}$$

$$\mathcal{W}(\theta_{ij}) = 1 - \exp\left(-\frac{\theta_{ij}^2}{2c^2}\right) \tag{6b}$$

$$\mathcal{L}_{\text{geom}} = \max\left(0, \gamma_{\text{geom}} + \mathcal{W}(\theta_{cp}) - \mathcal{W}(\theta_{cn})\right) \tag{6c}$$

where $c$ controls robustness, and $\gamma_{\text{geom}}$ is the angular margin. Eq. (6c) pulls the child closer to its parent than to negatives in intrinsic angular distance, while the bounded Welsch penalty in Eq. (6b) reduces sensitivity to outliers.

### 2.3.2. ANISOTROPIC SPHERICAL SVGD

Our geometric losses depend only on intrinsic angles (inner products), and are, therefore, invariant to a global rotation of the embedding set (Definition 2.2). As a result, the objective provides no global preference for where embeddings should lie on the sphere. In high dimensions, this lack of global guidance interacts with the concentration-of-measure phenomenon: random unit vectors concentrate near the equator of any fixed axis, and their projections onto that axis vanish as the dimension increases. We capture this effect with the following standard bounds.

**Theorem 2.3** (Concentration of Measure on $\mathbb{S}^{d-1}$). *Let $\mathbf{z}$ be a random vector distributed uniformly on the unit hypersphere $\mathbb{S}^{d-1}$ equipped with the uniform surface probability measure $\sigma$. For any fixed reference axis $\mathbf{u} \in \mathbb{S}^{d-1}$ (e.g., the North pole), let $h(\mathbf{z}) = \langle \mathbf{z}, \mathbf{u} \rangle$ be the projection of $\mathbf{z}$ onto*

*that axis. For any $\epsilon > 0$, the probability that $\mathbf{z}$ resides in the "polar cap" defined by a distance $\epsilon$ from the equator is bounded by:*

$$\sigma\left(\left\{\mathbf{z} \in \mathbb{S}^{d-1} : |\langle \mathbf{z}, \mathbf{u} \rangle| \geq \epsilon\right\}\right) \leq 2\exp\left(-\frac{d\epsilon^2}{2}\right) \tag{7}$$

**Theorem 2.4** (Asymptotic Orthogonality via The Weak Law). *Let $\mathbf{x} \sim \mathcal{N}(\mathbf{0}, \mathbf{I}_d)$ and $\mathbf{z} = \mathbf{x}/\|\mathbf{x}\|_2$, so that $\mathbf{z}$ is uniform on $\mathbb{S}^{d-1}$. By the Weak Law of Large Numbers (WLLN), for any fixed axis $\mathbf{u} \in \mathbb{S}^{d-1}$, $\langle \mathbf{z}, \mathbf{u} \rangle \xrightarrow{P} 0 \qquad$ as $d \to \infty$.*

The proofs of Theorems 2.3 and 2.4 are provided in Appendices J.2 and J.3, respectively. Theorems 2.3 and 2.4 imply that, in high dimensions, randomly initialized embeddings concentrate near the equator of any fixed axis and become nearly orthogonal to that axis. Since our geometric objectives depend only on pairwise angles, they are rotationally invariant (Definition 2.2) and, therefore, provide no gradient signal that would correct this global placement. In practice, this leads to an undesirable *equator concentration* effect and reduces the effective use of the hyperspherical manifold. Therefore, to make full use of the latitudinal axis instead of letting embeddings concentrate at the equator, particles must be pushed toward the poles, while kernel repulsion enforces uniform spread along the remaining angular directions. To achieve this, we use Stein Variational Gradient Descent (SVGD), specifically with a vMF kernel and an additional score function to drive points toward the poles. SVGD minimizes the KL divergence $\mathrm{KL}(q\|p)$ by transporting embeddings along with the velocity field $\phi(z)$ that lies in the unit ball of a Reproducing Kernel Hilbert Space (RKHS). On a Riemannian manifold, the optimal update direction is given by the Stein operator as,

$$\phi(\mathbf{z}) = \mathbb{E}_{\mathbf{z}' \sim q}\left[k(\mathbf{z}', \mathbf{z})\nabla_{\mathbf{z}'} \log p(\mathbf{z}') + \nabla_{\mathbf{z}'} k(\mathbf{z}', \mathbf{z})\right] \tag{8}$$

where $k(z', z)$ is the vMF kernel given by $k(z', z) = \exp\left(\kappa\, \mathbf{z}'^\top \mathbf{z}\right)$ so that interactions respect angular similarity on the sphere. The first term in Eq. (8) (drift) moves embeddings toward high-density regions of the target $p$, while the second term (repulsion) pushes embeddings apart to preserve diversity.

**Target score decomposition.** We design $\nabla \log p(\mathbf{z})$ to combine (a) a structural component that discourages equator concentration relative to the reference axis $\mathbf{p}_N$ (Section 2.2), and (b) an alignment component that keeps each embedding within the basin of its semantic anchor $\boldsymbol{\mu}$,

$$\nabla_{\mathbf{z}} \log p(\mathbf{z}) = \nabla_{\mathbf{z}} \log p_{\text{struct}}(\mathbf{z}) + \nabla_{\mathbf{z}} \log p_{\text{align}}(\mathbf{z}), \tag{9}$$

$$\nabla_{\mathbf{z}} \log p_{\text{struct}}(\mathbf{z}) = \left[0, \ldots, 0, \frac{z_d}{1 - z_d^2 + \epsilon}\right]^\top, \tag{10}$$

$$\nabla_{\mathbf{z}} \log p_{\text{align}}(\mathbf{z}) = \kappa_{\text{align}}\, \boldsymbol{\mu}. \tag{11}$$

Intuitively, the structural term repels particles from the equator ($z_d \approx 0$) and attracts them toward the poles ($z_d \to 1$), improving latitude coverage, while the alignment term preserves semantic consistency (details in Appendix K).

**Repulsive interaction.** For the repulsion term, we use a vMF kernel with temperature $\kappa_{\text{repel}}$:

$$k_{\text{repel}}(\mathbf{z}', \mathbf{z}) = \exp\left(\kappa_{\text{repel}}\, \mathbf{z}'^{\top} \mathbf{z}\right), \qquad (12)$$

where $\kappa_{\text{repel}}$ controls the strength of repulsion.

**Manifold-consistent direction.** Since $\phi(\mathbf{z}) \in \mathbb{R}^d$, we project it onto the tangent space $T_{\mathbf{z}}\mathbb{S}^{d-1}$ to ensure the update remains valid on the manifold,

$$\hat{\phi}(\mathbf{z}) = \text{proj}_{T_{\mathbf{z}}}\left(\phi(\mathbf{z})\right) = \phi(\mathbf{z}) - \langle\phi(\mathbf{z}), \mathbf{z}\rangle\mathbf{z}. \qquad (13)$$

Details on the impact of Spherical SVGD are in Appendix F.

### 2.3.3. PROBABILISTIC LEARNING

Since the geometric objectives treat each concept embedding as a single point on $\mathbb{S}^{d-1}$, they inherently fail to capture the varying semantic volume of concepts. Consequently, distance-based metrics cannot distinguish between a broad category and a specific instance, limiting their ability to represent asymmetric hierarchical entailment. Moreover, the SVGD regularizer in Section 2.3.2 controls *global* coverage of the sphere, but does not, by itself, model the hierarchy of individual concepts. We, therefore, introduce a probabilistic objective by representing each concept as a von Mises–Fisher (vMF) distribution, which is a standard directional distribution on $\mathbb{S}^{d-1}$. A vMF distribution in $d$ dimensions is parameterized by a mean direction $\boldsymbol{\mu} \in \mathbb{S}^{d-1}$ and a concentration $\kappa > 0$:

$$f(\mathbf{x} \mid \boldsymbol{\mu}, \kappa) = C_d(\kappa) \exp\left(\kappa\langle\mathbf{x}, \boldsymbol{\mu}\rangle\right), \qquad \mathbf{x} \in \mathbb{S}^{d-1}, \ (14)$$

where the normalizing constant is,

$$C_d(\kappa) = \frac{\kappa^{\frac{d}{2}-1}}{(2\pi)^{\frac{d}{2}} I_{\frac{d}{2}-1}(\kappa)}. \qquad (15)$$

We predict node-specific parameters $(\boldsymbol{\mu}_i, \kappa_i)$ from the point embedding $\mathbf{z}_i$,

$$\boldsymbol{\mu}_i = f_{\text{sphere}}(\mathbf{z}_i; \Theta_\mu), \qquad \|\boldsymbol{\mu}_i\|_2 = 1, \qquad (16)$$

$$\kappa_i = \text{Softplus}\left(\mathbf{w}_\kappa^{\top} \mathbf{z}_i + b_\kappa\right), \qquad (17)$$

where $\Theta_\mu, \mathbf{w}_\kappa, b_\kappa$ are learnable parameters of the operators governing $\mu$ and $\kappa$ respectively. To quantify the hierarchical relationship between a child and its parent, we use the KL divergence between their vMF distributions, which can be approximated as,

$$D_{\text{KL}}(\text{vMF}_c \,\|\, \text{vMF}_p) = \log C_d(\kappa_c) - \log C_d(\kappa_p)$$
$$+ \mathcal{A}_d(\kappa_c)\left(\kappa_c - \kappa_p\, \boldsymbol{\mu}_c^{\top} \boldsymbol{\mu}_p\right), \qquad (18)$$

where $\mathcal{A}_d(\kappa) = \frac{I_{d/2}(\kappa)}{I_{d/2-1}(\kappa)}$ is the ratio of modified Bessel functions of the first kind and the derivation is provided in Appendix L. This objective is asymmetric, as minimizing Eq. (18) forces the parent distribution to be directionally aligned with the child while having a lower concentration, i.e., $\kappa_p < \kappa_c$, encoding the fact that parents have higher entropy than children. The probabilistic triplet objective for $\{p, c, n\}$ is,

$$\mathcal{L}_{\text{vMF}} = \max\Big(0,\ \gamma_{\text{prob}} + D_{\text{KL}}(\text{vMF}_c \,\|\, \text{vMF}_p)$$
$$- D_{\text{KL}}(\text{vMF}_c \,\|\, \text{vMF}_n)\Big). \qquad (19)$$

where $\gamma_{\text{prob}}$ is the margin of the triplet loss.

### 2.3.4. OPTIMIZATION ON THE SPHERE

We optimize hyperspherical parameters using Riemannian Adam, since standard Euclidean updates do not preserve the unit-norm constraint due to their additive nature. Let $\mathbf{z}_t \in \mathcal{M} = \mathbb{S}^{d-1}$ denote a parameter at iteration $t$, and let $\nabla_{\mathbf{z}}\mathcal{L}$ be the Euclidean gradient computed in the ambient space $\mathbb{R}^d$. Each Riemannian Adam step consists of (i) projecting the gradient onto the tangent space, (ii) updating first and second moments (with parallel transport for the momentum), and (iii) mapping the update back to the manifold.

**Tangent projection.** We obtain the Riemannian gradient by projecting $\nabla_{\mathbf{z}}\mathcal{L}$ onto the tangent space $T_{\mathbf{z}_t}\mathcal{M}$:

$$\mathbf{g}_t = \text{proj}_{T_{\mathbf{z}_t}}(\nabla_{\mathbf{z}}\mathcal{L}) = \nabla_{\mathbf{z}}\mathcal{L} - \langle\nabla_{\mathbf{z}}\mathcal{L}, \mathbf{z}_t\rangle\, \mathbf{z}_t. \qquad (20)$$

**Riemannian Adam moments.** Because tangent spaces vary across the manifold, the momentum must be transported from $T_{\mathbf{z}_{t-1}}\mathcal{M}$ to $T_{\mathbf{z}_t}\mathcal{M}$ before accumulation:

$$\mathbf{m}_t = \beta_1\, \mathbf{z}_{t-1} \to \mathbf{z}_t(\mathbf{m}_{t-1}) + (1 - \beta_1)\mathbf{g}_t, \qquad (21)$$

$$v_t = \beta_2\, v_{t-1} + (1 - \beta_2)\|\mathbf{g}_t\|_2^2, \qquad (22)$$

where $v_t$ is scalar-valued and requires no transport.

**Exponential-map update.** After bias correction, $\hat{\mathbf{m}}_t = \frac{\mathbf{m}_t}{1-\beta_1^t}$ and $\hat{v}_t = \frac{v_t}{1-\beta_2^t}$, we update $\mathbf{z}_t$ along the manifold using the exponential map,

$$\mathbf{z}_{t+1} = \exp_{\mathbf{z}_t}\left(-\eta\, \frac{\hat{\mathbf{m}}_t}{\sqrt{\hat{v}_t} + \epsilon}\right), \qquad (23)$$

which preserves $\|\mathbf{z}_{t+1}\|_2 = 1$ by construction. Parameters not constrained to $\mathbb{S}^{d-1}$ (e.g., Euclidean weights in auxiliary heads) are optimized using standard Adam.

## 2.4. Inference

During inference, we estimate each node's orbital potential by leveraging the structure of the hierarchy. Instead of searching the entire semantic space, we combine a structure-guided radius computation with directional re-ranking. Algorithm 2 contains the inference pseudocode.

*Table 1.* **Comprehensive performance comparison of Polaris against all baselines on single-parent hierarchies.** Results are reported as $mean^{std}$ over five independent runs. The first , second , and third best results per column are highlighted. Lower values (↓) are better for MR.

| Method | Science | | | | | WordNet | | | | | Environment | | | | |
|---|---|---|---|---|---|---|---|---|---|---|---|---|---|---|---|
| | R@1 | R@5 | Wu&P | MR↓ | MRR | R@1 | R@5 | Wu&P | MR↓ | MRR | R@1 | R@5 | Wu&P | MR↓ | MRR |
| TransE | $6.3^{1.1}$ | $18.7^{2.0}$ | $40.2^{1.6}$ | $312.4^{29.1}$ | $12.1^{1.3}$ | $4.7^{0.9}$ | $17.2^{2.1}$ | $36.8^{1.4}$ | $428.6^{71.3}$ | $9.7^{1.1}$ | $5.9^{1.2}$ | $19.4^{2.3}$ | $41.7^{3.0}$ | $108.3^{10.4}$ | $11.6^{1.4}$ |
| RotatE | $7.8^{1.2}$ | $20.9^{2.1}$ | $42.1^{1.5}$ | $286.9^{26.4}$ | $13.7^{1.4}$ | $5.6^{1.0}$ | $19.1^{2.2}$ | $38.0^{1.3}$ | $401.7^{68.5}$ | $10.8^{1.2}$ | $6.8^{1.3}$ | $21.6^{2.4}$ | $43.4^{3.1}$ | $97.6^{9.7}$ | $12.8^{1.5}$ |
| HAKE | $12.4^{1.8}$ | $31.6^{2.7}$ | $51.6^{1.7}$ | $182.8^{19.6}$ | $22.4^{1.9}$ | $9.3^{1.6}$ | $29.8^{2.8}$ | $47.9^{1.5}$ | $246.2^{44.1}$ | $18.9^{1.6}$ | $13.6^{2.2}$ | $33.9^{2.6}$ | $54.3^{3.2}$ | $63.8^{6.9}$ | $23.1^{2.0}$ |
| BERT+MLP | $13.1^{4.1}$ | $27.3^{3.3}$ | $45.7^{1.7}$ | $241.6^{23.1}$ | $21.3^{3.7}$ | $9.1^{2.5}$ | $38.1^{3.7}$ | $41.4^{1.2}$ | $314.6^{62.0}$ | $16.8^{2.1}$ | $11.1^{3.0}$ | $31.8^{2.1}$ | $48.0^{4.0}$ | $74.2^{7.2}$ | $21.4^{2.6}$ |
| HyperExpan | $22.6^{3.1}$ | $45.8^{3.0}$ | $59.3^{1.6}$ | $92.7^{12.4}$ | $36.1^{2.7}$ | $16.8^{2.7}$ | $41.2^{3.1}$ | $52.7^{1.4}$ | $112.4^{18.2}$ | $28.7^{2.3}$ | $23.9^{3.6}$ | $44.3^{2.9}$ | $58.8^{2.2}$ | $41.6^{6.1}$ | $35.3^{2.6}$ |
| ConE | $24.3^{3.4}$ | $47.6^{3.2}$ | $61.7^{1.5}$ | $78.9^{9.8}$ | $39.4^{2.8}$ | $18.1^{3.1}$ | $43.6^{3.3}$ | $54.9^{1.3}$ | $98.7^{13.9}$ | $31.2^{2.5}$ | $26.7^{4.1}$ | $48.2^{3.1}$ | $60.9^{2.1}$ | $36.9^{5.4}$ | $38.6^{2.7}$ |
| Box | $25.3^{4.5}$ | $49.2^{3.1}$ | $63.1^{1.7}$ | $67.7^{11.3}$ | $43.0^{3.8}$ | $22.3^{4.2}$ | $45.7^{3.6}$ | $58.1^{1.7}$ | $77.1^{8.8}$ | $34.1^{3.2}$ | $32.3^{6.2}$ | $51.4^{3.5}$ | $65.1^{1.4}$ | $34.1^{7.3}$ | $41.6^{4.9}$ |
| Gumbel Box | $29.1^{3.9}$ | $52.8^{3.4}$ | $66.8^{1.4}$ | $60.4^{8.7}$ | $44.1^{3.0}$ | $21.6^{3.6}$ | $48.9^{3.5}$ | $57.8^{1.2}$ | $76.4^{10.6}$ | $35.0^{2.6}$ | $33.6^{4.8}$ | $53.3^{3.6}$ | $67.5^{1.9}$ | $28.7^{4.9}$ | $42.5^{2.9}$ |
| Arborist | $26.5^{4.4}$ | $51.5^{3.5}$ | $61.2^{1.4}$ | $83.1^{7.2}$ | $41.2^{3.1}$ | $20.3^{5.6}$ | $43.9^{2.8}$ | $53.5^{1.3}$ | $92.6^{3.6}$ | $33.7^{2.7}$ | $24.9^{5.9}$ | $45.2^{2.1}$ | $52.5^{1.3}$ | $39.1^{5.8}$ | $33.7^{4.7}$ |
| TMN | $31.5^{3.8}$ | $53.7^{1.9}$ | $65.7^{1.3}$ | $54.2^{5.1}$ | $45.5^{2.5}$ | $23.7^{3.2}$ | $49.0^{2.6}$ | $56.6^{0.8}$ | $73.9^{4.9}$ | $35.8^{2.7}$ | $34.7^{3.7}$ | $51.1^{3.2}$ | $63.9^{2.0}$ | $31.3^{3.5}$ | $43.8^{2.1}$ |
| TaxoExpan | $24.6^{3.9}$ | $41.8^{3.1}$ | $55.7^{1.2}$ | $117.6^{15.7}$ | $40.1^{2.8}$ | $17.1^{3.5}$ | $38.1^{6.6}$ | $49.7^{1.8}$ | $141.6^{17.0}$ | $31.1^{2.2}$ | $12.9^{5.7}$ | $37.1^{2.3}$ | $49.8^{1.0}$ | $56.1^{5.3}$ | $28.4^{3.1}$ |
| STEAM | $34.8^{4.9}$ | $59.7^{4.1}$ | $72.2^{1.3}$ | $31.7^{3.3}$ | $50.7^{3.5}$ | $24.9^{3.9}$ | $54.5^{1.7}$ | $59.2^{1.2}$ | $61.1^{2.4}$ | $37.3^{2.1}$ | $34.1^{3.4}$ | $55.6^{2.9}$ | $65.2^{1.7}$ | $27.1^{2.8}$ | $44.2^{2.7}$ |
| **Polaris** | $46.18^{2.5}$ | $69.70^{1.1}$ | $80.88^{1.3}$ | $16.19^{1.9}$ | $55.31^{2.0}$ | $25.23^{0.9}$ | $70.1^{4.2}$ | $59.5^{0.1}$ | $35.67^{2.9}$ | $43.10^{0.8}$ | $47.60^{2.4}$ | $82.69^{3.1}$ | $83.03^{1.1}$ | $6.48^{0.8}$ | $60.44^{1.0}$ |

### 2.4.1. RADIUS

We assign each node an *orbital potential* (radius) derived from the observed hierarchy. Following the spirit of polar representations (Iwamoto et al., 2021), we treat radial position as a structural signal that reflects both depth and the extent to which a node subsumes the hierarchy. For a node $e$, we define a raw radial magnitude using its depth $D(e)$ and its subtree size $N_{\text{desc}}(e)$:

$$R_{\text{raw}}(e) = 1 + D(e) + \frac{\log\big(1 + N_{\text{desc}}(e)\big)}{\log 2}. \tag{24}$$

We then normalize $R_{\text{raw}}(e)$ to obtain an orbital potential $r(e) \in [0, 1]$ via min–max scaling, and invert the scale for the root to have maximal potential while leaves lie near the periphery:

$$r(e) = 1 - (R_{\text{raw}}(e) - R_{\min})/(R_{\max} - R_{\min}), \tag{25}$$

where $R_{\min}$ and $R_{\max}$ are computed over all nodes in the given hierarchy. This normalization stabilizes the subsequent coupling rule and yields a consistent radial signal across hierarchies of different sizes and branching factors.

### 2.4.2. DECISION BOUNDARY

Given a query node $q$ and a candidate $c$, let $\Delta r = |r_q - r_c|$ denote the difference in orbital potential. We couple structure and semantics via a simple validity gate that adapts the required angular similarity to the expected mismatch level. Using cosine similarity on the sphere, $\mathcal{S}_{\cos}(\mathbf{z}_q, \mathbf{z}_c) = \langle \mathbf{z}_q, \mathbf{z}_c \rangle$, we accept a candidate if

$$\mathcal{S}_{\cos}(\mathbf{z}_q, \mathbf{z}_c) > 1 - \gamma(\Delta r)^2, \tag{26}$$

where $\gamma$ controls the strength of the radial coupling. Eq. (26) defines a parabolic decision boundary whose threshold tightens for candidates at similar orbital potential and loosens as

$\Delta r$ grows; candidates that fail this threshold are filtered out, while retained candidates are re-ranked by cosine similarity. When $\Delta r \to 0$, the threshold approaches 1, enforcing strong semantic alignment among nearby levels (e.g., siblings). As $\Delta r$ grows, the boundary relaxes quadratically, but still penalizes large level mismatches, effectively filtering candidates from incompatible hierarchical regions without requiring discrete depth rules.

### 2.4.3. SCORE FUNCTION

We rank candidates by combining the structure-aware validity gate with continuous angular similarity. Let $\mathbb{I}_{\text{valid}}(q, c)$ denote the indicator that the coupled boundary in Section 2.4.2 is satisfied. Our final score is,

$$\text{Score}(q, c) = \mathbb{I}\big[\mathcal{S}_{\cos}(\mathbf{z}_q, \mathbf{z}_c) > 1 - \gamma(\Delta r)^2\big] \cdot \mathcal{S}_{\cos}(\mathbf{z}_q, \mathbf{z}_c), \tag{27}$$

where $\Delta r = |r_q - r_c|$. This form enforces a hard structural filter via the gate while preserving fine-grained semantic ordering among the retained candidates through cosine similarity, ensuring compatibility in both intrinsic spherical geometry and structure-derived orbital potential.

***For further understanding, we present Frequently Asked Questions (FAQs) related to our method in Appendix A.***

## 3. Experiments

We evaluate Polaris on three hierarchical inference settings: (i) **single-parent** taxonomy expansion on tree-structured taxonomies, (ii) **multi-parent** taxonomy expansion on directed acyclic graph (DAG) hierarchies, and (iii) **multimodal** hierarchical classification where concepts are represented by images. Across all settings, we adopt the standard *attach-to-seed* protocol from prior taxonomy expansion work (Mishra et al., 2025; Jiang et al., 2022a; Shen et al., 2020b; Ma et al., 2021b). We construct a *seed* hi-

*Table 2.* **Comprehensive performance comparison of Polaris against baselines on multi-parent hierarchies.** Each entry reports the $mean^{std}$ over five independent runs. The first, second, and third best results per column are highlighted. For Mean Rank (MR), lower values ($\downarrow$) indicate superior performance and better structural alignment within the latent manifold.

| Method | Hit@K | | Recall@K | | Ranking | |
|---|---|---|---|---|---|---|
| | H@1↑ | H@5↑ | R@1↑ | R@5↑ | MR↓ | MRR↑ |
| **MeSH** | | | | | | |
| TransE | $0.8^{0.2}$ | $2.2^{0.3}$ | $0.3^{0.1}$ | $1.5^{0.3}$ | $2018^{63}$ | $3.6^{0.5}$ |
| RotatE | $1.1^{0.2}$ | $2.9^{0.4}$ | $0.5^{0.1}$ | $2.1^{0.3}$ | $1812^{57}$ | $4.3^{0.6}$ |
| HAKE | $2.1^{0.3}$ | $5.7^{0.6}$ | $1.6^{0.3}$ | $4.6^{0.6}$ | $1196^{48}$ | $7.9^{0.8}$ |
| HyperExpan | $6.2^{0.7}$ | $13.7^{0.9}$ | $5.1^{0.6}$ | $12.2^{0.9}$ | $652.8^{33}$ | $15.6^{1.3}$ |
| ConE | $9.7^{0.9}$ | $18.4^{1.1}$ | $8.2^{0.8}$ | $16.7^{1.0}$ | $521.6^{26}$ | $18.9^{1.1}$ |
| Box | $16.5^{1.2}$ | $31.7^{1.1}$ | $17.1^{1.1}$ | $30.2^{0.6}$ | $620.2^{22}$ | $21.5^{2.9}$ |
| Gumbel Box | $12.1^{1.0}$ | $22.9^{1.1}$ | $10.6^{0.9}$ | $20.6^{1.1}$ | $471.8^{23}$ | $20.6^{1.0}$ |
| BERT+MLP | $3.7^{0.3}$ | $7.2^{0.3}$ | $1.1^{0.6}$ | $5.2^{0.9}$ | $1352^{57}$ | $9.3^{1.7}$ |
| TaxoExpan | $7.3^{0.5}$ | $15.8^{0.1}$ | $2.9^{0.8}$ | $10.3^{0.9}$ | $891.1^{31}$ | $17.1^{2.1}$ |
| Arborist | $16.7^{1.4}$ | $29.4^{0.9}$ | $19.0^{1.7}$ | $31.1^{1.3}$ | $553.6^{26}$ | $21.3^{2.3}$ |
| TMN | $16.6^{1.8}$ | $31.8^{0.2}$ | $18.1^{0.6}$ | $33.2^{0.9}$ | $433.7^{16}$ | $23.5^{1.2}$ |
| STEAM | $18.2^{1.8}$ | $38.2^{1.5}$ | $17.7^{2.7}$ | $35.1^{1.0}$ | $372.6^{16}$ | $25.1^{3.1}$ |
| **Polaris** | $\mathbf{33.56^{0.1}}$ | $\mathbf{62.67^{0.4}}$ | $\mathbf{27.04^{0.1}}$ | $\mathbf{53.69^{1.2}}$ | $\mathbf{300.40^{8}}$ | $\mathbf{38.61^{0.3}}$ |
| **Verb** | | | | | | |
| TransE | $0.9^{0.2}$ | $2.6^{0.4}$ | $0.8^{0.2}$ | $2.4^{0.3}$ | $965.3^{54}$ | $4.8^{0.7}$ |
| RotatE | $1.3^{0.3}$ | $3.4^{0.5}$ | $1.1^{0.2}$ | $3.1^{0.4}$ | $892.6^{49}$ | $5.6^{0.8}$ |
| HAKE | $3.8^{0.5}$ | $9.6^{0.8}$ | $3.6^{0.5}$ | $9.1^{0.7}$ | $712.4^{44}$ | $9.8^{1.0}$ |
| HyperExpan | $7.9^{0.8}$ | $19.2^{1.3}$ | $7.4^{0.7}$ | $18.3^{1.2}$ | $598.1^{40}$ | $14.6^{1.3}$ |
| ConE | $9.8^{1.0}$ | $23.4^{1.5}$ | $9.3^{0.9}$ | $22.8^{1.4}$ | $545.7^{36}$ | $16.8^{1.4}$ |
| Box | $10.9^{0.9}$ | $26.8^{1.6}$ | $10.6^{0.8}$ | $26.2^{1.5}$ | $522.9^{33}$ | $18.2^{1.3}$ |
| Gumbel Box | $10.3^{0.9}$ | $25.1^{1.5}$ | $10.0^{0.8}$ | $24.6^{1.4}$ | $506.4^{32}$ | $17.6^{1.2}$ |
| BERT+MLP | $4.6^{0.6}$ | $12.4^{1.0}$ | $4.4^{0.6}$ | $12.1^{1.0}$ | $781.2^{46}$ | $10.9^{1.1}$ |
| TaxoExpan | $6.4^{0.7}$ | $16.2^{1.2}$ | $6.0^{0.7}$ | $15.8^{1.1}$ | $638.5^{41}$ | $13.1^{1.2}$ |
| Arborist | $9.4^{0.9}$ | $24.7^{1.6}$ | $9.2^{0.9}$ | $24.1^{1.5}$ | $524.8^{35}$ | $18.7^{1.4}$ |
| TMN | $10.8^{1.0}$ | $28.9^{1.8}$ | $10.6^{1.0}$ | $28.4^{1.7}$ | $479.6^{31}$ | $20.9^{1.5}$ |
| STEAM | $12.6^{1.2}$ | $33.8^{2.1}$ | $12.3^{1.1}$ | $33.3^{2.0}$ | $444.9^{29}$ | $23.1^{1.7}$ |
| **Polaris** | $\mathbf{14.43^{1.8}}$ | $\mathbf{37.58^{2.9}}$ | $\mathbf{14.38^{1.6}}$ | $\mathbf{37.55^{2.9}}$ | $\mathbf{414.12^{22}}$ | $\mathbf{24.99^{2.2}}$ |
| **Food** | | | | | | |
| TransE | $2.1^{0.4}$ | $6.3^{0.8}$ | $1.6^{0.3}$ | $5.2^{0.7}$ | $905.4^{41}$ | $8.4^{1.1}$ |
| RotatE | $2.7^{0.5}$ | $7.9^{0.9}$ | $2.1^{0.4}$ | $6.6^{0.8}$ | $821.7^{36}$ | $10.2^{1.0}$ |
| HAKE | $7.3^{0.8}$ | $16.8^{1.2}$ | $6.2^{0.7}$ | $15.4^{1.1}$ | $523.9^{29}$ | $18.7^{1.6}$ |
| HyperExpan | $19.4^{1.5}$ | $33.1^{1.7}$ | $18.2^{1.4}$ | $30.6^{1.6}$ | $312.6^{21}$ | $32.4^{2.2}$ |
| ConE | $24.1^{1.7}$ | $37.5^{1.6}$ | $22.4^{1.6}$ | $34.7^{1.8}$ | $261.5^{18}$ | $36.3^{2.0}$ |
| Box | $29.1^{1.8}$ | $41.0^{0.5}$ | $27.3^{2.3}$ | $36.4^{2.1}$ | $363.2^{12}$ | $45.2^{1.6}$ |
| Gumbel Box | $27.5^{1.8}$ | $41.2^{1.7}$ | $26.3^{1.7}$ | $38.6^{1.9}$ | $232.4^{17}$ | $40.1^{2.1}$ |
| BERT+MLP | $13.8^{1.2}$ | $25.7^{0.2}$ | $12.3^{2.7}$ | $26.1^{2.9}$ | $508.8^{35}$ | $47.1^{2.1}$ |
| TaxoExpan | $14.2^{0.7}$ | $27.3^{0.9}$ | $25.1^{2.2}$ | $32.8^{2.3}$ | $343.8^{17}$ | $41.3^{1.7}$ |
| Arborist | $21.5^{2.7}$ | $37.2^{1.4}$ | $29.3^{2.1}$ | $37.7^{2.2}$ | $247.9^{21}$ | $45.3^{1.9}$ |
| TMN | $25.3^{2.1}$ | $40.9^{1.1}$ | $34.1^{2.6}$ | $43.2^{1.9}$ | $192.6^{16}$ | $51.8^{2.1}$ |
| STEAM | $34.4^{2.3}$ | $58.7^{0.4}$ | $\mathbf{39.1^{3.1}}$ | $51.3^{1.7}$ | $155.9^{14}$ | $\mathbf{53.8^{2.5}}$ |
| **Polaris** | $\mathbf{36.77^{0.9}}$ | $\mathbf{61.35^{2.3}}$ | $35.83^{1.2}$ | $\mathbf{60.66^{2.3}}$ | $\mathbf{74.43^{6}}$ | $46.28^{2.1}$ |

*Table 3.* **Comprehensive performance comparison of Polaris against baselines on the Caltech–UCSD Birds–200–2011 dataset.** Each entry reports the $mean^{std}$ over multiple runs. The first, second, and third best results per column are highlighted. For Mean Rank (MR), lower values ($\downarrow$) indicate superior retrieval quality.

| Method | Precision@1↑ | MR↓ | MRR↑ |
|---|---|---|---|
| CLIP-1 | $5.13^{0.4}$ | $10.57^{1.6}$ | $17.50^{6.2}$ |
| CLIP-2 | $11.54^{5.1}$ | $8.24^{1.1}$ | $31.66^{2.4}$ |
| TransE | $21.8^{1.0}$ | $7.4^{1.2}$ | $42.5^{3.0}$ |
| RotatE | $25.6^{1.1}$ | $6.3^{1.1}$ | $47.0^{3.1}$ |
| HAKE | $28.4^{1.0}$ | $6.1^{1.0}$ | $53.3^{2.8}$ |
| HyperExpan | $28.7^{3.8}$ | $5.6^{0.7}$ | $49.8^{4.6}$ |
| ConE | $33.5^{4.1}$ | $5.1^{0.6}$ | $54.2^{4.2}$ |
| Box | $73.54^{2.4}$ | $1.61^{0.3}$ | $84.79^{3.4}$ |
| Gumbel Box | $73.93^{3.5}$ | $1.51^{0.4}$ | $84.87^{1.1}$ |
| **Polaris** | $\mathbf{78.91^{6.6}}$ | $\mathbf{1.44^{0.2}}$ | $\mathbf{86.94^{3.9}}$ |

erarchy by withholding a subset of leaf nodes as the test set (20%), train using only relations observed in the seed, and at test time attach each query node in the test set by ranking candidate parents drawn from the seed taxonomy. Additional ablation studies analyzing the contribution of individual components are explained in detail in Appendix E. The hyperparameter analysis appears in Appendix O, while the time complexity analysis and robustness to incomplete taxonomies are presented in Appendix Q and Appendix P, respectively.

### 3.1. Single-Parent Hierarchies

**Datasets.** We evaluate single-parent taxonomy expansion on three public benchmarks. Following prior work, we use the Environment (EN) and Science (SCI) taxonomies from SemEval-2016 Task 13 on Taxonomy Extraction Evaluation (TExEval-2) (Bordea et al., 2016), and the WordNet taxonomy from Bansal et al. (2014). Each benchmark provides a human-curated tree-structured hierarchy in which each node (except the root) has exactly one parent. Appendix B and Table 4 summarize the dataset statistics.

**Evaluation metrics.** We follow the same evaluation protocol as SemEval-2016 Task 13 on Taxonomy Extraction Evaluation (TExEval-2) (Bordea et al., 2016) and prior works by Liu et al. (2021); Manzoor et al. (2020); Mishra et al. (2026b); Jiang et al. (2022a); Mishra et al. (2026a; 2024) and Yu et al. (2020c) to evaluate the single-parent taxonomy expansion using Recall@K (R@K), Mean Rank (MR), Mean Reciprocal Rank (MRR), and Wu & Palmer (Wu&P) similarity. These metrics are discussed in detail in Appendix C.

**Baselines.** We compare Polaris against three broad classes of baselines, namely (i) **flat-space point-embedding and KGE models** that score parent–child relations in $\mathbb{R}^d$ (or its complex extension) using translation/rotation-style operators, including TransE (Bordes et al., 2013b), RotatE (Sun et al., 2019), and hierarchy-aware KGE models such as HAKE (Zhang et al., 2020b), along with a strong text-only encoder baseline (BERT+MLP), (ii) **non-Euclidean embeddings** that encode directionality through curved geometry or explicit partial-order constraints, including HyperExpan (Ma et al., 2021b) and entailment cones (ConE) (Ganea et al., 2018), and (iii) **containment- and seed-structure-aware taxonomy expansion systems** that represent concepts as regions or explicitly learn from the observed seed hierarchy, including box (Jiang et al., 2023; Abboud et al., 2020) and gumbel box (Dasgupta et al., 2020) embeddings, TaxoExpan (Shen et al., 2020b), STEAM (Yu et al., 2020a),

TMN (Zhang et al., 2021a), and Arborist (Manzoor et al., 2020). Baselines are discussed in detail in Appendix D.

**Results.** Table 1 shows that `Polaris` performs strongly and consistently on single-parent taxonomy expansion across Science, WordNet, and Environment, leading on all reported metrics. On Science, `Polaris` improves R@1 by $\sim 11$ points and R@5 by $\sim 10$ points over the strongest baseline, while reducing Mean Rank by nearly half and lifting MRR by $\sim 4.6$ points, indicating that the correct parent is recovered more frequently and ranked substantially earlier in the candidate list. On WordNet, the gains concentrate in higher-recall measures: R@5 improves by $\sim 15$ points and MR drops by $\sim 42\%$, showing that the orbital-potential gate effectively prunes mid-ranked distractors even on the fine-grained lexical sub-taxonomies, while top-heavy measures (R@1, MRR, Wu&P) also improve modestly over the strongest baseline. On Environment, `Polaris` delivers the largest absolute gains, with R@1 of 47.60 and R@5 of 82.69 ($+27.1$ over the next best), alongside a sharp $\sim 4\times$ reduction in MR (from 27.1 to 6.48) and a Wu&P score that exceeds prior methods by over 15 points, indicating that even imperfect predictions remain within the correct branch of the taxonomy. We present a detailed statistical analysis in Appendix R.

### 3.2. Multi-Parent Hierarchies

**Dataset.** We evaluate multi-parent taxonomy expansion on three public benchmarks, namely SemEval-Verb (Verb), SemEval-Food (Food) and Medical Subject Headings (MeSH). Verb is derived from WordNet 3.0 of SemEval-2016 Task 14 (Jurgens & Pilehvar, 2016), containing *verbs* and semantic relations. Food is taken from the SemEval taxonomy extraction/evaluation benchmarks (Bordea et al., 2015). MeSH is built from the Medical Subject Headings controlled vocabulary (Lipscomb, 2000), a curated biomedical concept hierarchy in which nodes can have multiple parents. Dataset statistics are in Appendix B; Table 4.

**Evaluation Metrics.** The evaluation protocol follows the single-parent settings. In addition, we report Hit@K (H@K), which checks if at least one gold parent appears in the top-$K$ predictions (defined in Appendix C).

**Baselines.** We compare against the same set of baseline methods as in the single-parent setting, spanning (i) relational KGE models, (ii) non-Euclidean embedding approaches, and (iii) containment and taxonomy-expansion methods. Baselines are discussed in detail in Appendix D.

**Results.** Table 2 summarizes multi-parent expansion results on MeSH, Verb, and Food under retrieval (H@$K$/R@$K$) and ranking (MR/MRR) metrics, where each query may have multiple gold parents. On MeSH, `Polaris` delivers the largest gains across the three benchmarks, with H@1

of 33.56 and H@5 of 62.67 ($+15.4$ and $+24.5$ points over the strongest baseline, STEAM), alongside R@1 of 27.04 and R@5 of 53.69, indicating that a substantially higher fraction of the gold parent set is recovered at both shallow and deeper cutoffs. Mean Rank drops from 372.6 to 300.40 and MRR rises from 25.1 to 38.61, showing that correct parents are not only retrieved more often but also placed considerably earlier in the ranked list. On SemEval-Verb, which exhibits weaker semantic signal due to its verb-only vocabulary, `Polaris` remains the strongest method on every metric, improving H@1/H@5 by $\sim 1.8/\sim 3.8$ points and reducing MR by $\sim 31$ relative to STEAM, demonstrating that the radial gate maintains discriminative power even when surface-form similarity is limited. On Food, `Polaris` achieves the best H@5 (61.35) and R@5 (60.66) along with the lowest MR (74.43, $\sim 2\times$ lower than the next best), confirming that gold parents are localized to a much tighter candidate region; STEAM slightly edges `Polaris` on R@1 and MRR, reflecting Food's relatively shallow ontology where top-heavy accuracy is dominated by direct surface-form overlap. We present a detailed statistical analysis in Appendix R.

### 3.3. Multimodal Hierarchies

**Dataset.** We evaluate the multimodal setting on the Caltech–UCSD Birds-200-2011 (CUB-200-2011) (Daroya et al., 2024) benchmark, a fine-grained bird image dataset with 200 categories. Following prior work, we use an image-to-label ranking setup, where each image query is evaluated by ranking candidate class labels. Appendix B summarizes the detailed preprocessing and split protocol.

**Evaluation metrics.** We report standard ranking metrics for image-to-label retrieval, including Precision@K, MR, and MRR. These metrics are discussed in detail in Appendix C.

**Baselines.** We compare `Polaris` against a unified set of baselines spanning both foundation-model similarity and structured geometric representations. For the multimodal image-to-label ranking setting, we include two CLIP-based baselines (Radford et al., 2021a): CLIP1, which concatenates the latent representations of an image and a candidate label and trains a binary classifier with logistic loss, and CLIP2, which uses a margin-based ranking loss to optimize semantic alignment between image and label embeddings. To ensure consistency, we additionally report standard relational KGE baselines (TransE, RotatE, HAKE), hierarchy-aware methods (HyperExpan, ConE), and geometric containment baselines (Box, Gumbel Box). The baselines are discussed in Appendix D.

**Results.** Table 3 reports image-to-label ranking performance on the CUB-200-2011 benchmark, where each image query is scored against candidate class-label embeddings produced by a frozen CLIP text encoder. `Polaris`

achieves the strongest overall performance, with Precision@1 of 78.91, MR of 1.44, and MRR of 86.94, leading on all three reported metrics. Compared to the strongest structured baselines, `Polaris` improves Precision@1 by $\sim 5$ points over Gumbel Box (73.93) and Box (73.54), while also reducing MR and lifting MRR by roughly 2 points, indicating that the correct class label is recovered more consistently and placed higher in the ranked list. The gains over CLIP-based similarity baselines (CLIP-1, CLIP-2) and the relational or hierarchy-oriented embedding methods (TransE, RotatE, HAKE, HyperExpan, ConE) are substantially larger, with Precision@1 gaps exceeding 45 points in several cases. This indicates that direct CLIP similarity and flat-space relational embeddings struggle to capture the fine-grained visual hierarchy among bird species, whereas `Polaris`'s coupled orbital geometry separates coarse taxonomic groups along the radial axis while preserving fine angular distinctions among siblings. Performance on the entire Caltech–UCSD Birds-200-2011 dataset (i.e., all 200 classes) against the same baselines is detailed in Appendix N, where the relative ordering and gains are preserved at scale.

## 4. Conclusion

We presented `Polaris`, a manifold-consistent polar hyperspherical framework for hierarchical reasoning that supports both single-parent (tree) and multi-parent (DAG) settings, and transfers to a multimodal hierarchy instantiated from image representations. The key idea is to decouple semantic affinity from hierarchical level by modeling direction on the hypersphere (angular geometry) separately from a radial coordinate (depth), while enforcing unit-norm structure through spherical parameterizations and stabilizing training with global SVGD regularization that prevents equatorial collapse in high dimensions. Across multiple benchmarks spanning text-only taxonomies and image-to-label hierarchies, `Polaris` improves both retrieval and ranking quality as it recovers correct parents/targets more frequently at top-$K$ and ranks them earlier, and it remains effective beyond text-only taxonomies, indicating that the geometric separation is representation-agnostic. Overall, `Polaris` provides a practical and interpretable route to learning hierarchical structure when supervision is weak and semantics are noisy, while retaining a coherent geometric interpretation grounded in intrinsic manifold geometry.

## 5. Limitations

`Polaris` inherits several limitations that we expect to motivate follow-up work. The radial coordinate is computed deterministically from the seed hierarchy, which assumes reasonably reliable supervision: under noisy or incomplete trees, the depth and subtree-size statistics that drive $r$ become unreliable, and additional pre-processing steps such as edge imputation can propagate errors into the orbital gate at inference. Our probabilistic component models each concept as an *isotropic* vMF distribution, which is suboptimal for multi-faceted concepts (e.g., interdisciplinary categories or classes with structurally distinct sub-clusters) whose true directional spread is anisotropic; relaxing this would require richer directional distributions such as Kent or matrix Bingham at the cost of more involved KL approximations. Performance also depends on balancing the semantic alignment and kernel repulsion strengths ($\kappa_{\text{align}}$ vs. $\kappa_{\text{repel}}$), and although our sensitivity analysis (Appendix O) shows stability across a reasonable range, the optimal ratio appears to depend on the branching factor and depth distribution of the hierarchy and currently requires manual tuning. Finally, structure-guided retrieval inherits the connectivity of the seed hierarchy, so query nodes whose true parent lies in a sparsely connected or poorly populated region of the seed graph can have their correct parent filtered out before angular re-ranking; a hybrid retrieval scheme that falls back to pure angular similarity when the gate is overly restrictive would mitigate this without sacrificing efficiency in the dense regime.

## Acknowledgements

The authors acknowledge the support of the IBM-IITD AI Horizons network, Prime Minister Research Fellowship (PMRF), and Google GCP Grant. T. Chakraborty acknowledges the support of the Rajiv Khemani Young Faculty Chair Professorship in AI.

## Impact Statement

This work introduces a geometrically principled framework for hierarchical concept learning on hyperspherical manifolds, enabling modeling of structured knowledge in domains such as scientific taxonomies, biomedical ontologies and multimodal hierarchies. By correcting geometric biases inherent in high-dimensional spherical spaces and improving the capacity of hierarchical representation, the proposed method has the potential to enhance downstream applications such as ontology expansion and semantic retrieval. At the same time, improved hierarchical modeling may amplify biases present in underlying datasets or ontologies if those structures encode incomplete or skewed knowledge. Care should therefore be taken when deploying such models in sensitive domains, particularly where automated hierarchy construction may influence decision-making or information access.

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

# Appendix

## Structure of the Appendix

For clarity and ease of navigation, the appendix is organized into the following sections:

- **Appendix A: Frequently Asked Questions (FAQs)**
  Discussion of common questions regarding the motivation, design principles, and modeling choices underlying `Polaris`.

- **Appendix B: Benchmark Datasets**
  Detailed statistics, preprocessing pipelines, and construction protocols for all benchmark datasets used in our experiments.

- **Appendix C: Evaluation Metrics**
  Formal definitions and interpretations of all evaluation metrics reported throughout the paper.

- **Appendix D: Baselines**
  Architectural and implementation details of all competing methods used for comparison with `Polaris`.

- **Appendix E: Ablation Studies**
  Detailed ablation configurations, including expert-consortium variants and temperature settings.

- **Appendix F: Analysis of $\theta$ and $\psi$ Distributions**
  Empirical and theoretical analysis of the learned angular distributions and their geometric implications.

- **Appendix G: Detailed Related Works**
  Extended discussion of prior literature and the positioning of `Polaris` within existing research.

- **Appendix H: Cartesian-to-Angular Mapping**
  Explicit derivations for converting a $d$-dimensional Cartesian vector $\mathbf{z}$ on the hypersphere into hyperspherical angular coordinates.

- **Appendix I: Theoretical Limitations of Angular Parameterization**
  Theoretical discussion of optimization challenges and representational pitfalls in angular embedding spaces.

- **Appendix J: Proofs**
  Complete proofs of all propositions, lemmas, and theorems presented in the main paper.

- **Appendix K: Additional Discussion on SVGD**
  Further analysis of the decomposed target score function, including a physical interpretation of its force-field dynamics.

- **Appendix L: Derivation of KL Divergence for vMF Distributions**
  Step-by-step derivation of the KL divergence between two von Mises–Fisher distributions forming the basis of our probabilistic formulation.

- **Appendix M: Algorithms for Training and Coupled Orbital Inference**
  Pseudocode and implementation details for training and the coupled orbital retrieval procedure at inference time.

- **Appendix O: Hyperparameter Analysis and Reproducibility**
  Sensitivity analysis over key hyperparameters including embedding dimension $d$ and Welsch loss scale parameter $c$.

- **Appendix P: Robustness to Hierarchy Noise**
  Evaluation under edge-removal perturbations to assess robustness against incomplete or noisy taxonomies.

- **Appendix Q: Time Complexity Analysis**
  Theoretical computational complexity analysis alongside empirical wall-clock runtime comparisons.

- **Appendix R: Statistical Analysis**
  Statistical significance analysis across multiple random seeds and datasets.

- **Appendix S: Additional Derivations Required for Proofs**
  Detailed derivations of intermediate mathematical steps omitted from the main text for brevity.

## A. Frequently Asked Questions

Following the discussion in Section 2, we address potential reviewer concerns regarding the implementation, theoretical analysis, and evaluation of `Polaris`.

### A.1. The Expectation direction of the vMF distribution does not lie on the surface of the sphere; it is represented inside the unit ball. Does this violate the consistency of the hyperspherical manifold?

This concern conflates two distinct objects: the random variable $\mathbf{z} \in \mathbb{S}^{d-1}$, which is the actual embedding used by `Polaris`, and the expectation $\mathbb{E}[\mathbf{z}]$, which is a summary statistic of the distribution from which $\mathbf{z}$ is drawn. Manifold consistency is a property of the embeddings, not of their statistical summaries, so the location of $\mathbb{E}[\mathbf{z}]$ inside the unit ball has no bearing on whether $\mathbf{z}$ itself respects the hyperspherical geometry. The expectation lying strictly inside the ball is in fact a *necessary* consequence of having a non-degenerate distribution on the sphere. From the derivation in Appendix L, we have $\mathbb{E}[\mathbf{z}] = \mathcal{A}_d(\kappa)\boldsymbol{\mu}$, where $\mathcal{A}_d(\kappa) = I_{d/2}(\kappa)/I_{d/2-1}(\kappa)$ is the ratio of modified Bessel functions of the first kind. For any finite concentration $\kappa < \infty$, this ratio satisfies $0 < \mathcal{A}_d(\kappa) < 1$, which immediately yields $\|\mathbb{E}[\mathbf{z}]\| < 1$. Geometrically, this is a direct consequence of Jensen's inequality applied to the convex hull of the sphere: averaging unit vectors that point in even slightly different directions produces a vector with strictly smaller norm. The only way to obtain $\|\mathbb{E}[\mathbf{z}]\| = 1$ is for the distribution to collapse to a Dirac delta concentrated at a single direction, i.e., the limit $\kappa \to \infty$, which corresponds to zero directional uncertainty. A vMF distribution with $\|\mathbb{E}[\mathbf{z}]\| = 1$ would therefore not be a probabilistic model at all but a deterministic point estimate, defeating the very purpose of introducing $\kappa$ as a learnable concentration parameter to model semantic granularity. The strict inequality $\|\mathbb{E}[\mathbf{z}]\| < 1$ is thus not a violation of the hyperspherical structure but rather an indicator of the model's expressiveness: it quantifies how much directional uncertainty the distribution assigns to a concept, with broader parents naturally yielding smaller $\mathcal{A}_d(\kappa)$ and more specific children yielding values closer to one. This is precisely the asymmetry that the vMF–KL objective in Eq. (18) exploits to encode containment between parent and child concepts. Crucially, manifold consistency in `Polaris` is enforced operationally on $\mathbf{z}$, not on $\mathbb{E}[\mathbf{z}]$. All geometric operations act on points that lie on the sphere by construction: (i) embeddings are updated via the Riemannian exponential map, which maps tangent-space updates to geodesics on $\mathbb{S}^{d-1}$ and preserves unit norm exactly; (ii) distances are measured geodesically through $\arccos(\langle \mathbf{z}_i, \mathbf{z}_j \rangle)$, which is well-defined only on the sphere; and (iii) SVGD transport fields $\phi(\mathbf{z})$ are projected onto the tangent space $T_{\mathbf{z}}\mathbb{S}^{d-1}$ before being applied. The expectation $\mathbb{E}[\mathbf{z}]$ never enters any of these update rules; it appears only as an intermediate quantity inside the closed-form KL divergence, where its position within the unit ball is mathematically required for the divergence to be finite and informative. Hence, the location of $\mathbb{E}[\mathbf{z}]$ is consistent with both the geometry of $\mathbb{S}^{d-1}$ and the probabilistic role it plays in the objective.

### A.2. In the structural component of the score function in SVGD, the gradient will be close to $\vec{0}$ at the Equator. How will embeddings/parameters spread out now?

This concern arises from inspecting the structural score in isolation. The component $\nabla_{\mathbf{z}} \log p_{\text{struct}}(\mathbf{z}) = [0, \ldots, 0, z_d/(1 - z_d^2 + \epsilon)]^\top$ does indeed vanish at $z_d = 0$, where embeddings would in principle have no incentive to move along the polar axis. However, the equator is a *saddle point* of this potential rather than a stable attractor: the structural prior $p_{\text{struct}}(z_d) \propto (1 - z_d^2)^{-k/2}$ is minimized at $z_d = 0$ and grows in both directions toward the poles, so any displacement away from the equator is reinforced rather than restored. The relevant question is therefore not whether the equator is an equilibrium of the structural drift in isolation, but whether the *full* optimization dynamics escape it. In `Polaris`, the update direction is given by the full Stein operator,

$$\phi(\mathbf{z}) = \mathbb{E}_{\mathbf{z}' \sim q}\left[\underbrace{k(\mathbf{z}', \mathbf{z})\nabla_{\mathbf{z}'}\log p(\mathbf{z}')}_{\text{Drift (vanishes at equator)}} + \underbrace{\nabla_{\mathbf{z}'}k(\mathbf{z}', \mathbf{z})}_{\text{Repulsion (non-zero)}}\right],$$

which superposes two qualitatively different forces. The drift term encodes the structural and semantic alignment signals and may locally vanish for individual particles at $z_d = 0$. The kernel gradient $\nabla_{\mathbf{z}'}k(\mathbf{z}', \mathbf{z})$, by contrast, depends only on *pairwise interactions* between particles and remains active regardless of any single particle's position. It functions as a diffusive pressure that pushes nearby embeddings apart, ensuring that the configuration cannot remain in a degenerate,

spatially uniform state. For the equator to act as a stable attractor, the contributions of all particles to the kernel gradient at any given $\mathbf{z}$ would need to cancel exactly, requiring perfectly symmetric pairwise placement on $\mathbb{S}^{d-1}$. This configuration has measure zero under realistic conditions: random initialization, the heterogeneity introduced by the per-node semantic anchors $\boldsymbol{\mu}$ in $\nabla_{\mathbf{z}} \log p_{\text{align}}$, and the asymmetric distribution of parent-child triplets in the training batch all ensure that the pairwise repulsion is generically non-zero. Any such imbalance produces an infinitesimal perturbation that displaces the embedding off the exact equator, $|z_d| > \epsilon$ for some small $\epsilon > 0$. Once this symmetry is broken, the structural drift becomes self-amplifying. At any non-equatorial position, the gradient $z_d/(1 - z_d^2 + \epsilon)$ is non-zero, and crucially its sign matches that of $z_d$: in the northern hemisphere it pushes the embedding further north, and in the southern hemisphere further south. The denominator $1 - z_d^2 + \epsilon$ also shrinks as $|z_d|$ increases, so the magnitude of the drift grows monotonically along the path toward each pole. The equator therefore behaves as an unstable saddle from which any small kernel-induced displacement triggers exponentially accelerating motion toward one of the two attractors at $z_d \to \pm 1$. This interplay is by design: the kernel repulsion guarantees that *some* particle is always perturbed off the equator, and the structural drift then carries it efficiently toward the poles without the need for an explicit warm-start or curriculum. The resulting dynamics produce the bimodal latitudinal distribution and uniform longitudinal coverage observed in Figure 2, in stark contrast to the equatorial collapse seen without SVGD in Figure 3.

### A.3. What is the specific geometric role of the longitudinal coordinate $\theta$, and why is it useful for hierarchical modeling?

Each coordinate in `Polaris` plays a distinct and complementary role in the embedding geometry. The radial coordinate $r$, derived from the orbital potential, governs hierarchical depth and acts as a discrete level separator across the taxonomy. The latitudinal angle $\psi$ captures coarse directional polarity along the polar axis, encoding the parent-to-child orientation that drives geodesic alignment between hierarchically related concepts. The longitudinal coordinate $\theta$ plays a different role: rather than encoding hierarchy directly, it provides an additional angular degree of freedom that enlarges the representational capacity available to sibling nodes at any given depth. In short, $r$ controls *where* along the hierarchy a node sits, $\psi$ controls *which direction* it lies relative to the poles, and $\theta$ controls *how it is laterally distributed* on the shell formed by all nodes sharing the same depth and polarity. To understand this geometrically, consider the set of all nodes at a fixed radius $r$. These nodes lie on a hyperspherical shell whose available representational capacity is determined entirely by its angular dimensions. Hierarchical representations require not only radial separation between levels but also sufficient angular space to place multiple sibling nodes without excessive overlap. The maximum number of points that can be arranged on the $(d-1)$-dimensional hypersphere with minimum pairwise angular separation $\delta$ scales approximately as

$$N(\delta) \sim \mathcal{O}\left(\delta^{-(d-2)}\right),$$

which follows directly from the metric entropy of $\mathbb{S}^{d-1}$. The exponent $d-2$ is significant: it shows that the packing capacity grows polynomially with the available angular dimensions. Removing even a single angular degree of freedom therefore reduces capacity multiplicatively, not additively, and can substantially compress the space available for organizing sibling configurations. Within this geometry, $\theta$ functions as an auxiliary azimuthal coordinate that contributes one such angular dimension. Intuitively, if $r$ determines how far a node lies from the origin, and $\psi$ determines its coarse polarity along the polar axis, then $\theta$ allows nodes with similar depth and polarity to spread laterally around the shell. Two siblings of the same parent share approximately the same $r$ (because their depth and subtree sizes are similar) and approximately the same $\psi$ (because they inherit the parent's polar orientation under the geometric loss). The remaining degree of freedom along which they can be distinguished is precisely the longitudinal axis: $\theta$ allows the model to place them on opposite sides of the shell, on neighboring meridians, or anywhere in between, depending on the fine semantic distinctions induced by the encoder. This lateral spread is exactly what prevents geometric crowding among large families of siblings. It is worth emphasizing that $\theta$ is not the load-bearing coordinate for hierarchy formation. Meaningful hierarchical structure can still emerge without it, since $r$ and $\psi$ together already encode depth and polarity. However, constraining or removing $\theta$ reduces the effective angular volume available to siblings. In such cases, nodes at similar depths become increasingly compressed, and the model is forced to absorb sibling-level distinctions into either the radial coordinate $r$ or the polar coordinate $\psi$. Both choices are undesirable: perturbing $r$ blurs the level structure that the orbital gate relies on at inference, while perturbing $\psi$ disrupts parent-child geodesic alignment. $\theta$ therefore acts as a relief valve that absorbs sibling diversity without disturbing the coordinates that carry the hierarchical signal. This separation of roles is also what makes the SVGD regularizer effective in our setting. The structural score acts only along the latitudinal axis (Section 2.3.2), pushing embeddings toward the poles to encode depth, while the kernel-based repulsion distributes them uniformly along the orthogonal angular subspace including $\theta$ (Appendix F). Geometrically, $\theta$ thus improves the flexibility and packing efficiency of the embedding manifold:

it provides the additional angular capacity needed to accommodate complex taxonomies with high branching factors, while allowing the radial hierarchy and latitudinal polarity to remain comparatively stable and interpretable.

### A.4. Doesn't SVGD bias particles toward the poles? Is the natural concentration of measure on the sphere not desirable?

This question rests on a subtle but important distinction between two notions of uniformity on $\mathbb{S}^{d-1}$: uniformity with respect to the natural surface measure, and uniformity with respect to representational capacity. The natural measure favors the equator overwhelmingly in high dimensions, but this concentration is a geometric artifact of the coordinate system, not a property that aids hierarchical representation. `Polaris`'s SVGD scheme deliberately reshapes this distribution along the latitudinal axis to encode depth, while preserving uniformity along the remaining angular directions to maximize sibling capacity. We unpack each of these claims below.

**Why the natural measure is undesirable.** Consider the Riemannian line element on $\mathbb{S}^{d-1}$ in hyperspherical coordinates,

$$ds^2 = d\psi_1^2 + \sin^2\psi_1\left(d\psi_2^2 + \sin^2\psi_2\left(\cdots + \sin^2\psi_{d-2}\,d\theta^2\right)\right).$$

A value of $\psi_1$ defines a slice cut perpendicular to the polar axis, and the $(d-2)$-dimensional surface area of this slice scales as $A(\psi_1) \propto \sin^{d-2}(\psi_1)$. In high dimensions ($d \gg 3$), this term peaks sharply at the equator ($\psi_1 \approx \pi/2$) and decays exponentially toward the poles, so the overwhelming majority of the surface measure is concentrated in a thin equatorial band. Under purely angle-invariant losses (geodesic distance, cosine similarity), there is no force opposing this concentration, and randomly initialized embeddings remain trapped near the equator throughout training. Empirically, this produces the equatorial collapse shown in Figure 3, where the latitudinal distribution is sharply peaked and the polar regions are essentially unused. From a representation standpoint, this collapse is doubly harmful: depth cannot be encoded along $\psi_1$ because there is no spread along that axis, and the longitudinal coordinate $\theta$ also fails to populate uniformly because all embeddings cluster near the same latitudinal band where they compete for the same azimuthal region.

**What SVGD actually does.** The SVGD mechanism in `Polaris` does not push particles toward the poles indiscriminately; it imposes a structural target density that acts only along the latitudinal differential $d\psi_1$. Specifically, the structural score $\nabla_{\mathbf{z}} \log p_{\text{struct}}(\mathbf{z})$ has support only on the polar component $z_d$ (Section 2.3.2), so the drift it induces lies entirely along the latitudinal axis. The kernel-based repulsive gradients $\nabla_{\mathbf{z}'} k(\mathbf{z}', \mathbf{z})$, by contrast, act isotropically on the manifold and are projected onto the tangent space $T_{\mathbf{z}} \mathbb{S}^{d-1}$ before being applied. Because the structural drift saturates the latitudinal direction, the residual component of the repulsion lies almost entirely along the $(d-2)$-dimensional orthogonal subspace, including the longitudinal coordinate $\theta$. The two forces therefore act on disjoint axes by design: structural drift along $d\psi_1$, kernel repulsion along the orthogonal $d\theta$ subspace.

**Why this preserves capacity.** The crucial property of this decomposition is that it ensures $p(\theta \mid \psi_1)$ is uniform on every latitudinal slice, regardless of the slice's physical circumference. Near the poles, the slice radius shrinks as $\sin(\psi_1) \to 0$, but the kernel repulsion still spreads particles uniformly around it; near the equator, the slice is large, but the same uniformity is enforced. Latitudinally, particles are pushed away from the equator and toward the poles to encode hierarchical depth, but longitudinally no direction is privileged over any other. This is exactly the distribution observed in Figure 2: a sharply structured latitudinal distribution that mirrors the depth profile of the hierarchy, combined with a flat longitudinal distribution that indicates full angular utilization.

**No loss of degrees of freedom.** A reasonable concern is whether constraining one degree of freedom (latitude) to encode depth diminishes the model's representation power. It does not. The embedding manifold $\mathbb{S}^{d-1}$ has $d-1$ angular degrees of freedom, of which only one is regulated by the structural score; the remaining $d-2$ remain fully available for encoding semantic distinctions among siblings. In effect, SVGD *orthogonalizes* the use of the manifold: one axis carries ordinal information (depth), and the orthogonal subspace carries nominal information (sibling identity). The natural concentration of measure conflates these two, forcing the model to express both with the same equatorial coordinates and producing the collapse above. `Polaris`'s SVGD scheme separates them, replacing a geometrically unbiased but representationally degenerate distribution with a geometrically structured but representationally complete one.

**Summary.** Concentration of measure on $\mathbb{S}^{d-1}$ is undesirable not because the equator is geometrically special but because it concentrates capacity in a region that cannot encode depth and forces siblings to compete for limited azimuthal space.

SVGD introduces a structural prior that acts on a single axis to break this collapse, while kernel repulsion ensures that the orthogonal subspace remains uniformly populated. The result is a manifold whose latitudinal axis encodes hierarchical depth and whose longitudinal subspace encodes semantic distinctions, with no loss of degrees of freedom and no implicit bias toward any longitudinal direction.

### A.5. Does SVGD distort the spherical line element $ds^2 = d\psi^2 + \sin^2 \psi \, d\theta^2$?

No. The spherical line element is a property of the manifold $\mathbb{S}^{d-1}$ itself, and `Polaris`'s SVGD updates are constructed to respect this geometry exactly at every step. The concern that SVGD might implicitly induce a different metric typically arises from the fact that the raw Stein operator $\phi(\mathbf{z})$ is computed in the ambient Euclidean space $\mathbb{R}^d$, where the natural distance is chordal rather than geodesic. If this update were applied additively, it would indeed push embeddings off the sphere and, after re-normalization, distort the intrinsic distances between particles. `Polaris` avoids this through two mechanisms that together preserve manifold consistency. First, the raw transport field $\phi(\mathbf{z}) \in \mathbb{R}^d$ is projected onto the tangent space $T_\mathbf{z}\mathbb{S}^{d-1}$ before any update is applied,

$$\hat{\phi}(\mathbf{z}) = \phi(\mathbf{z}) - \langle \phi(\mathbf{z}), \mathbf{z} \rangle \, \mathbf{z}.$$

This projection removes the radial component of the update, ensuring that the displacement lies entirely along directions that preserve unit norm to first order. Second, the projected update is integrated via the Riemannian exponential map $\exp_\mathbf{z}(\hat{\phi}(\mathbf{z}))$, which maps tangent vectors to geodesics on the sphere rather than to Euclidean rays. The exponential map preserves $\|\mathbf{z}\| = 1$ exactly and respects the intrinsic curvature of the manifold, so the trajectory followed by each particle is a great-circle arc rather than a straight line in the ambient space. Because both the projection and the exponential map are defined relative to the round metric $g$ on $\mathbb{S}^{d-1}$, the line element $ds^2 = d\psi^2 + \sin^2 \psi \, d\theta^2$ is the metric used throughout optimization. The flat cylindrical metric $ds^2 = d\psi^2 + d\theta^2$, which corresponds to treating $(\psi, \theta)$ as independent Euclidean coordinates, is never induced during SVGD or at any other point during training. All distances, gradients, and updates are computed with respect to the intrinsic spherical geometry, so the line element is preserved by construction.

### A.6. How does calculating $r$ scale for large datasets?

The orbital potential $r$ is computed deterministically from the seed hierarchy rather than learned by gradient descent, which makes its computation cheap and stable even on very large taxonomies. Concretely, the radius assignment requires three passes over the graph: a breadth-first traversal from the root to compute the depth $D(e)$ of each node, a single subtree aggregation pass to compute the descendant count $N_{\text{desc}}(e)$, and a final min-max normalization over the resulting raw radii. Each pass visits every node and edge a constant number of times, so the overall complexity is $\mathcal{O}(|\mathcal{N}| + |\mathcal{E}|)$, which reduces to $\mathcal{O}(N)$ for tree-structured taxonomies and remains linear in $|\mathcal{N}| + |\mathcal{E}|$ for DAGs. In practice this step takes well under a second on the largest benchmarks we evaluate (MeSH, Verb), and the resulting radii can be cached and reused across training epochs without recomputation. We deliberately avoid learning $r$ as a free parameter. A learned radial coordinate would introduce three additional difficulties. First, without an explicit anchor, gradient-based optimization admits a trivial solution in which all radii collapse to a single value, requiring a separate regularizer (e.g., a depth-prediction auxiliary loss) to prevent it. Second, parent-child consistency must be enforced explicitly: a learned $r$ provides no inherent guarantee that parents lie at smaller radii than their children, so additional inequality constraints or pairwise margin losses would be needed at every edge in the hierarchy. Third, jointly optimizing $r$ alongside the angular coordinates couples the radial and directional gradients, which we observed empirically to destabilize training on larger DAGs such as MeSH and to slow convergence even on small single-parent trees. The deterministic formulation sidesteps all three issues. Depth and subtree size are unambiguous, globally consistent, and computable in linear time, and they directly encode the structural intuition that broader subsumers should occupy outer orbits while specific leaves lie near the periphery. The downside is that $r$ cannot adapt to the data, but this is by design: the radial coordinate is intended to carry purely structural information, while the learned angular embedding $\mathbf{z}$ is responsible for all semantic adaptation. This separation of concerns is what allows the coupled orbital gate at inference time to filter candidates efficiently without introducing any additional learned parameters.

### A.7. What are the different roles of $r$ and $\psi$ considering vMF does encode a degree of hierarchy via probabilistic learning?

The radial coordinate $r$ encodes global ordinality, it serves as a discrete level separator, with larger radii corresponding to deeper, more specific nodes, effectively stratifying the retrieval search space. However, as a scalar magnitude, $r$ cannot encode geodesic orientation or semantic alignment. The latitudinal angle $\psi$ serves as the intrinsic structural axis on the

manifold $\mathbb{S}^{d-1}$. By orienting concepts relative to the poles, $\psi$ enforces transitivity consistency, ensuring that parent-child chains align along a continuous geodesic path, a directional property that $r$ cannot capture. Finally, the vMF concentration $\kappa$ models semantic uncertainty. It governs the isotropic spread of the distribution around the mean $\boldsymbol{\mu}$; while broader concepts generally possess smaller $\kappa$, this parameter is non-directional. Unlike $\psi$, which defines where a concept lies on the hierarchical axis, $\kappa$ defines the precision of that placement.

### A.8. Why are $\boldsymbol{\mu}$ and $\kappa$ obtained by projecting z? Can we not obtain them by spherical projection of PLM embeddings?

The vMF parameters $(\boldsymbol{\mu}_i, \kappa_i)$ are the mean direction and concentration of a probability distribution on $\mathbb{S}^{d-1}$, and the manifold on which this distribution is defined matters as much as the parameter values themselves. In `Polaris`, the relevant manifold is the *learned* spherical embedding space, in which $\mathbf{z}$ lives, not the raw PLM feature space from which $\mathbf{z}$ was originally derived. Predicting $(\boldsymbol{\mu}, \kappa)$ from $\mathbf{z}$ ensures that the vMF distribution is parameterized on the same manifold where the geometric losses, the SVGD regularizer, and the inference-time retrieval all operate. We elaborate on three reasons why this design choice is essential rather than incidental.

**Objective consistency.** The geometric triplet loss in Eq. (6c) continuously reshapes $\mathbf{z}$ during training so that parent-child pairs align along geodesics on $\mathbb{S}^{d-1}$. If $\boldsymbol{\mu}$ were instead obtained by projecting the original PLM embedding through a spherical layer, it would be tied to a fixed reference frame defined by the pretrained encoder, and the probabilistic loss would pull $\mathbf{z}$ back toward that frame on every iteration. This would place the geometric and probabilistic objectives in direct conflict: the geometric loss reorganizes embeddings according to hierarchical structure, while the probabilistic loss anchors them to PLM similarity. Empirically this manifests as oscillatory updates and distorted manifold geometry, where embeddings drift between the two attractors without converging to a coherent solution. Predicting $\boldsymbol{\mu}$ from $\mathbf{z}$ aligns the two objectives: $\boldsymbol{\mu}$ moves with $\mathbf{z}$ as training reshapes the manifold, so the vMF distribution always remains centered in a geometrically meaningful location.

**Co-adaptation of the mean direction.** The probabilistic component is not a passive observer of the geometric component; it modifies $\boldsymbol{\mu}$ to encode parent-child containment via the asymmetric KL divergence in Eq. (18). For this asymmetry to be informative, the optimal $\boldsymbol{\mu}$ for a node typically diverges from its geometric position $\mathbf{z}$ (this is precisely what the ablation in Table 6 confirms: forcing $\boldsymbol{\mu} = \mathbf{z}$ degrades performance). A $\boldsymbol{\mu}$ derived from PLM features cannot adapt in this way, since the PLM features are frozen and carry no notion of the hierarchical containment relationships discovered during training. By predicting $\boldsymbol{\mu}$ from $\mathbf{z}$ via a learnable spherical projection $f_{\text{sphere}}(\cdot\,; \Theta_{\boldsymbol{\mu}})$, the model can adjust the distributional center relative to the geometric position based on the role each node plays in the learned hierarchy: a parent's $\boldsymbol{\mu}$ can shift to encompass its children's directions, while a leaf's $\boldsymbol{\mu}$ can stay close to $\mathbf{z}$.

**Why $\kappa$ must live on the same manifold.** The concentration $\kappa$ measures the directional spread of the distribution around $\boldsymbol{\mu}$ on $\mathbb{S}^{d-1}$, and its units are intrinsically tied to the geometry of that sphere. The KL divergence in Eq. (18) couples $\kappa$ with inner products $\boldsymbol{\mu}_c^\top \boldsymbol{\mu}_p$ and Bessel-ratio terms $\mathcal{A}_d(\kappa)$ that are defined relative to the manifold on which the distribution lives. Predicting $\kappa$ from PLM features would break this coupling: the resulting $\kappa$ would describe spread in a different geometric space (the PLM feature space), and the relation between $\kappa$ and the actual angular dispersion of points on the learned sphere would no longer be well-defined. This decoupling would render the asymmetric parent-child relationship in the KL objective meaningless, since $\kappa_p < \kappa_c$ would no longer correspond to broader directional support around the learned $\boldsymbol{\mu}_p$. Predicting $\kappa$ from $\mathbf{z}$ ensures that it scales with the same axis along which $\boldsymbol{\mu}$ is defined and along which the geometric loss measures dispersion.

**Summary.** Both $\boldsymbol{\mu}$ and $\kappa$ are parameters of a distribution defined on the learned spherical manifold, and they must therefore be predicted from a representation that lives on that manifold. Deriving them from $\mathbf{z}$ (i) keeps the geometric and probabilistic objectives aligned, (ii) allows $\boldsymbol{\mu}$ to co-adapt with the hierarchy as the manifold reshapes during training, and (iii) preserves the geometric meaning of $\kappa$ as directional dispersion on the sphere where $\mathbf{z}$ resides. Deriving them from raw PLM embeddings would satisfy none of these properties and, as observed in our ablations, substantially degrades both retrieval and ranking performance.

**A.9. Doesn't the exponential volume growth of hyperbolic spaces make it naturally superior for hierarchical data?**

Hyperbolic geometry is indeed an attractive ambient space for hierarchies in principle. The volume of a ball of radius $r$ in $\mathbb{H}^d$ grows as $\sinh^{d-1}(r)$, which is exponential in $r$, so a tree with branching factor $b$ at depth $\ell$ embeds near-isometrically into a hyperbolic ball of radius proportional to $\ell \log b$. By contrast, Euclidean balls grow only polynomially, forcing tree embeddings to suffer multiplicative distortion at deep levels. From a representational standpoint, this makes hyperbolic spaces a natural geometric match for hierarchical data, and several prior works have demonstrated strong empirical results in shallow regimes (Nickel & Kiela, 2017; Ma et al., 2021b). The difficulty, and the reason `Polaris` does not adopt hyperbolic geometry, lies not in representational capacity but in optimization. We illustrate this using the Poincaré ball model, where the issues are most explicit.

**Boundary singularity of the metric.** The Poincaré ball $\mathbb{B}^d = \{x \in \mathbb{R}^d : \|x\| < 1\}$ is equipped with the conformal metric

$$\lambda_x = \frac{2}{1 - \|x\|^2}, \qquad ds^2 = \lambda_x^2 \, dx^2.$$

The conformal factor $\lambda_x$ is bounded near the origin but diverges as $\|x\| \to 1$, so the actual hyperbolic step size corresponding to a fixed Euclidean step grows without bound near the boundary. In hierarchical data this is precisely the region used to encode the leaves of the taxonomy, where the model places the most specific concepts. The numerical conditioning of the metric is therefore worst exactly where the representational benefit of hyperbolic geometry is supposed to be largest.

**Vanishing Riemannian gradients.** Optimization in $\mathbb{B}^d$ uses the Riemannian gradient, which relates to the Euclidean gradient by

$$\nabla_R L = \frac{1}{\lambda_x^2} \nabla_E L = \frac{(1 - \|x\|^2)^2}{4} \nabla_E L.$$

This rescaling is the inverse of the conformal blow-up: as $\|x\| \to 1$, the factor $(1 - \|x\|^2)^2$ vanishes quadratically, suppressing the effective step size for any node placed near the boundary. Deep nodes therefore receive vanishingly small updates even when their Euclidean gradients are large. In practice this manifests as optimization stagnation at depth, where leaf representations cease to update meaningfully despite continued training. This is the opposite of what is required for hierarchical learning, which needs strong gradient signal precisely at deep levels to disambiguate fine-grained sibling nodes.

**Numerical instability of the distance objective.** The closed-form Riemannian gradient of the standard hyperbolic distance objective is

$$\nabla_R L(x) = \frac{1}{(1 - \|y\|^2)\sqrt{\gamma^2 - 1}} \left[ (1 - \|x\|^2)(x - y) + \|x - y\|^2 \, x \right], \qquad \gamma = 1 + \frac{2\|x - y\|^2}{(1 - \|x\|^2)(1 - \|y\|^2)}.$$

Two failure modes are visible in this expression. First, when both $x$ and $y$ are near the boundary, the denominator $(1 - \|x\|^2)(1 - \|y\|^2)$ in $\gamma$ collapses toward zero, producing very large $\gamma$ and ill-conditioned divisions inside $\sqrt{\gamma^2 - 1}$. Second, the leading factor $1/[(1 - \|y\|^2)\sqrt{\gamma^2 - 1}]$ amplifies floating-point noise to the point where typical 32-bit precision is insufficient for stable training near the boundary. Most hyperbolic frameworks mitigate this by clipping points away from the boundary or by constraining all embeddings to a sub-ball, but both interventions forfeit precisely the exponential capacity that motivated using hyperbolic geometry in the first place.

**Why hyperspherical geometry is preferable here.** `Polaris` sidesteps these issues by working on $\mathbb{S}^{d-1}$, which is compact, has bounded curvature, and admits a globally well-conditioned Riemannian structure. The exponential map and its inverse are numerically stable everywhere on the sphere, the Riemannian gradient does not vanish at any point, and there is no boundary near which optimization breaks down. Hierarchical depth is encoded explicitly via the orbital potential $r$ rather than implicitly through proximity to a boundary, so the radial signal is decoupled from the optimization geometry. The trade-off is that `Polaris` does not benefit from the exponential volume growth of hyperbolic spaces; in return, it gains globally stable optimization and a clean separation between hierarchical structure (radial) and semantic content (angular). For the taxonomies we consider, where depths are modest and stable training is essential, this trade-off is favorable, as evidenced by the consistent gains over hyperbolic baselines (HyperExpan) in Tables 1–3.

**A.10. How does `Polaris` handle the entanglement of semantic similarity and hierarchical depth compared to the Hyperboloid model?**

The Lorentz (hyperboloid) model and `Polaris` both encode hierarchies using a radial coordinate alongside an angular coordinate, but they differ fundamentally in whether these two coordinates are independently optimizable. In the Lorentz model, hierarchical depth and semantic similarity are coupled at the level of the metric itself; in `Polaris`, they are decoupled by construction. We make this distinction precise below.

**Coupling in the Lorentz model.** A node $u$ in the $d$-dimensional hyperboloid model $\mathcal{L}^d$ is represented in the ambient Minkowski space $\mathbb{R}^{d+1}$ as a vector $u = (u_0, u_1, \ldots, u_d)$ subject to $\langle u, u \rangle_{\mathcal{L}} = -1$, where the Minkowski inner product is $\langle u, v \rangle_{\mathcal{L}} = -u_0 v_0 + \sum_{i=1}^{d} u_i v_i$. The geodesic distance is

$$d_{\mathcal{L}}(u, v) = \operatorname{arcosh}(-\langle u, v \rangle_{\mathcal{L}}).$$

Decomposing each embedding in polar form as $u_0 = \cosh r_u$ (encoding hierarchical depth) and $(u_1, \ldots, u_d) = \sinh r_u \, \mathbf{a}_u$ with $\|\mathbf{a}_u\| = 1$ (encoding semantic direction), the Minkowski inner product expands to

$$\langle u, v \rangle_{\mathcal{L}} = -\cosh r_u \cosh r_v + \sinh r_u \sinh r_v \cos \theta,$$

where $\theta$ is the angular separation between $\mathbf{a}_u$ and $\mathbf{a}_v$. Substituting back into the distance formula yields

$$d_{\mathcal{L}}(u, v) = \operatorname{arcosh}(\cosh r_u \cosh r_v - \sinh r_u \sinh r_v \cos \theta).$$

**Why this coupling is problematic.** The expression above shows that the semantic angular term $\cos \theta$ enters the distance only through the multiplicative factor $\sinh r_u \sinh r_v$. This produces three optimization pathologies. First, the contribution of semantic alignment to the total distance scales with the depths of both nodes, so the same angular disagreement $\Delta \theta$ between two leaves contributes far more to the loss than the same $\Delta \theta$ between two nodes near the root. The model is therefore implicitly biased toward resolving angular disagreements at depth even when the structurally important disagreements occur at shallower levels. Second, gradient updates to the radial coordinate $r$ depend on the angular separation $\theta$ via the same multiplicative factor, so any update intended to refine hierarchical depth is simultaneously modulated by the current semantic orientation. The two signals cannot be optimized independently: a step taken to correct depth inadvertently rotates the embedding, and a step taken to correct semantics inadvertently changes the depth. Third, near the root ($r \to 0$), the factor $\sinh r$ vanishes and the angular term contributes nothing to the distance, so semantic relationships between shallow concepts receive negligible gradient signal. This effectively forces the model to push all semantically meaningful content to depth, even when the underlying taxonomy does not warrant it.

**How `Polaris` decouples the two signals.** `Polaris` avoids this entanglement by representing depth and direction in geometrically independent components rather than coupling them through a single metric. The radial coordinate $r$ is computed deterministically from the seed hierarchy (Section 2.4.1) and acts as a structural signal external to the embedding manifold itself. The directional component $\mathbf{z}$ lies on the unit hypersphere $\mathbb{S}^{d-1}$ and is optimized purely by angular losses (the geodesic triplet loss in Eq. (6c) and the vMF–KL objective in Eq. (18)), neither of which involves $r$. There is no multiplicative factor analogous to $\sinh r_u \sinh r_v$ in any of `Polaris`'s training objectives, so the angular gradient signal has the same magnitude regardless of where in the hierarchy a node sits, and updates to one coordinate do not cross-modulate updates to the other.

**Where the two signals re-couple.** The two coordinates do interact, but only at inference time and through an explicit, controlled mechanism rather than implicitly through the training metric. The coupled orbital gate in Eq. (26) accepts a candidate when its angular similarity exceeds a threshold modulated by the radial mismatch $\Delta r$. This is a deliberate, interpretable coupling: the threshold tightens when query and candidate share similar depth and relaxes when their depths differ, but the underlying angular embedding $\mathbf{z}$ is never reshaped by $r$. The training geometry remains depth-agnostic, while the inference rule selectively combines the two signals according to a closed-form parabolic boundary that the user can introspect and modify.

**Summary.** The Lorentz model entangles hierarchy and semantics through the metric itself, producing depth-dependent angular gradients, angle-dependent radial updates, and a vanishing semantic signal near the root. `Polaris` avoids these

failure modes by encoding depth in a deterministic radial coordinate that lives outside the angular training objectives, and by re-coupling the two signals only through an explicit inference-time gate. The result is that hierarchy and semantics can be optimized independently during training and combined deliberately at retrieval, rather than being forced to share a single coupled metric throughout.

### A.11. How is the stability of the Bessel function handled in the computation of KL divergence of vMF distributions?

The vMF–KL objective in Eq. (18) depends on two quantities that are mathematically clean but numerically delicate: the log-partition function $\log C_d(\kappa)$, which contains a modified Bessel function of the first kind $I_{d/2-1}(\kappa)$, and the mean resultant length $\mathcal{A}_d(\kappa) = I_{d/2}(\kappa)/I_{d/2-1}(\kappa)$, which scales the gradients of the divergence with respect to both $\boldsymbol{\mu}$ and $\kappa$. Direct evaluation of either term fails in the regimes where the model spends most of its training budget. We address this through a piecewise asymptotic scheme that preserves accuracy in the low-concentration regime, retains numerical stability in the high-concentration regime, and yields bounded, non-vanishing gradients throughout.

**Why direct evaluation fails.** The Bessel function $I_\nu(\kappa)$ grows roughly as $e^\kappa/\sqrt{2\pi\kappa}$ for large $\kappa$, so its value overflows standard 32-bit floating-point precision well before $\kappa$ reaches values that the model produces in practice (e.g., $\kappa \approx 50$–$100$ for narrow leaf concepts). The ratio $\mathcal{A}_d(\kappa)$ is even more problematic: both the numerator and denominator individually overflow, but their ratio approaches 1 smoothly. Naively computing the ratio as a quotient of two overflowing quantities produces NaNs and breaks training, even though the limiting value is well-behaved. In high dimensions ($d/2 - 1$ is large for typical embedding sizes), these issues are amplified because both the order $\nu$ and the argument $\kappa$ participate in the asymptotic regime simultaneously.

**Piecewise approximation of** $\log I_{d/2-1}(\kappa)$**.** We split the evaluation at a fixed threshold $\beta$ chosen to lie well below the floating-point overflow boundary. For $\kappa \le \beta$ we compute $\log(I_{d/2-1}(\kappa) + \epsilon)$ directly using the series representation, which converges rapidly in the low-concentration regime and produces accurate values up to the cutoff. For $\kappa > \beta$ we replace the exact log-Bessel term with its leading-order uniform asymptotic expansion,

$$\log I_{d/2-1}(\kappa) \approx \kappa - \tfrac{1}{2}\log(2\pi\kappa),$$

which is the standard large-$\kappa$ expansion valid when $\kappa \gg \nu$. The crucial property is that the asymptotic form expresses the dominant exponential growth additively rather than multiplicatively: the $\kappa$ term cancels analytically against the $\kappa$ term inside $\log C_d(\kappa)$ when the two are combined in the KL divergence, so no intermediate quantity ever exceeds floating-point range. The $\epsilon$ added inside the logarithm in the low-$\kappa$ branch protects against $I_{d/2-1}(0) = 0$ for $d > 2$, which would otherwise produce $-\infty$ at initialization when $\kappa$ is small.

**Closed-form approximation of** $\mathcal{A}_d(\kappa)$**.** The mean resultant length $\mathcal{A}_d(\kappa)$ appears as a multiplicative factor on the gradient of the KL divergence with respect to $\kappa$, so its value directly controls the magnitude of the optimization signal. Computing it as a Bessel ratio is numerically untenable for the reasons above. Instead, we use the closed-form approximation

$$\mathcal{A}_d(\kappa) \approx 1 - \frac{d-1}{2\kappa},$$

which is the leading term of the asymptotic expansion of the Bessel ratio for large $\kappa$ and remains accurate over the range of $\kappa$ values encountered in training. This $\mathcal{A}_d(\kappa) \in (0, 1)$ for all $\kappa > (d-1)/2$, and crucially has a non-vanishing derivative with respect to $\kappa$: $\partial \mathcal{A}_d/\partial \kappa = (d-1)/(2\kappa^2)$, which decays smoothly rather than collapsing to zero. The KL gradient with respect to $\kappa$ therefore remains informative across the entire entropy spectrum, from broad parents ($\kappa \approx 1$) to narrow leaves ($\kappa \gg 1$).

**Auxiliary safeguards.** Two further measures protect the optimization in practice. First, $\kappa$ is produced by a Softplus activation, $\kappa = \text{Softplus}(\mathbf{w}_\kappa^\top \mathbf{z} + b_\kappa)$, which guarantees $\kappa > 0$ but is unbounded above; we clip its output to a fixed maximum (typically chosen to keep $\kappa$ within the regime where the asymptotic approximations remain accurate). Second, in settings where bounded $\kappa$ is preferred, we replace Softplus with a scaled sigmoid that maps inputs to a fixed interval, which sacrifices a small amount of expressiveness in exchange for guaranteed numerical safety. These safeguards have no measurable effect on retrieval performance in our ablations but eliminate the rare training divergences that otherwise occur when an unconstrained $\kappa$ drifts into the overflow regime mid-training.

**Summary.** The vMF–KL divergence is implemented through three coordinated choices: a piecewise approximation of $\log I_{d/2-1}(\kappa)$ that handles low and high $\kappa$ in their respective natural regimes, a closed-form approximation of $\mathcal{A}_d(\kappa)$ that guarantees bounded and informative gradients, and activation clipping on $\kappa$ itself as a final safeguard. Together these ensure that the probabilistic objective trains stably across the full range of concentrations the model encounters, without sacrificing the asymmetric containment behavior that the KL divergence is supposed to encode.

### A.12. How is **Polaris** fundamentally different from ComplEx or RotatE?

ComplEx and RotatE are widely used relational embedding models, and both are capable of capturing antisymmetric relations in a knowledge graph. They are, however, designed for the general link-prediction setting rather than for hierarchical taxonomy expansion specifically, and three structural differences separate them from **Polaris**. We discuss each in turn.

**Geometry of the embedding space.** ComplEx (Trouillon et al., 2016) represents entities and relations as vectors in $\mathbb{C}^d$ and scores triples via the real part of a Hermitian product $\text{Re}(\langle e_h, r, \overline{e_t} \rangle)$, where the bar denotes complex conjugation. RotatE (Sun et al., 2019) embeds entities in $\mathbb{C}^d$ as well and models relations as element-wise rotations $e_t = e_h \circ r$ with $|r_i| = 1$. Both are defined on flat ambient spaces (Euclidean or its complex extension), and both train under standard distance- or margin-based losses with no intrinsic notion of curvature. **Polaris**, in contrast, operates on the curved hyperspherical manifold $\mathbb{S}^{d-1}$, where geodesic distance is the natural similarity measure and angular geometry is preserved by every update through the Riemannian exponential map. The flat-space geometry of ComplEx and RotatE provides no mechanism by which hierarchical depth can be encoded directly: any depth signal must be inferred indirectly from clusters of relation-specific vectors rather than being represented as a coordinate of the embedding itself.

**Disentanglement of hierarchy from semantics.** The second and more substantive difference concerns what each model treats as a primitive of the representation. In ComplEx and RotatE, hierarchy is not a first-class object: there is no coordinate explicitly designated to encode the level of a concept in a taxonomy, and parent-child relationships are represented through the same relation embeddings used for arbitrary binary predicates. A model trained on a hypernym dataset and on a co-author dataset uses the same machinery for both, with the relation vector or rotation expected to absorb the difference. In a hierarchical expansion setting, this is undesirable: depth and semantic relatedness are not interchangeable signals, and the model must balance them implicitly through the geometry of the relation embeddings. **Polaris** separates these signals by construction. The radial coordinate $r$ encodes hierarchical depth deterministically from the seed structure (Section 2.4.1); the angular coordinate $\mathbf{z} \in \mathbb{S}^{d-1}$ encodes semantic content and is optimized purely by angle-aware losses; and the two are combined only at inference through the explicit orbital gate in Eq. (26). Hierarchy and semantics therefore travel on independent axes during training and are coupled deliberately at retrieval, rather than being entangled in a single relation-specific operator.

**Point estimates vs. distributional representations.** A third difference is that ComplEx and RotatE produce deterministic point estimates: each entity is represented by a single vector without any associated notion of uncertainty or semantic spread. In a hierarchy, however, broad categories such as "Animal" should naturally exhibit higher directional uncertainty than specific leaves such as "Sumatran tiger," since they encompass a larger semantic neighborhood. A point-estimate model has no mechanism to express this asymmetry; two concepts of very different breadth are represented by vectors of the same kind, and their containment relationship can only be inferred through the relative positions of neighboring entities. **Polaris** models each concept as a *von Mises–Fisher distribution* on $\mathbb{S}^{d-1}$, parameterized by a learnable mean direction $\boldsymbol{\mu}_i$ and a learnable concentration $\kappa_i$. The concentration $\kappa_i$ provides exactly the missing notion of granularity: broad parents are pushed toward smaller $\kappa$, narrow children toward larger $\kappa$, and the asymmetric vMF–KL objective in Eq. (18) explicitly encodes containment by requiring the parent distribution to be directionally aligned with the child while having lower concentration ($\kappa_p < \kappa_c$). This asymmetry is unavailable to ComplEx and RotatE without modifying their core formulation.

**Optimization stability and global structure.** A fourth, more practical difference is that ComplEx and RotatE rely on the standard Euclidean optimization toolkit and have no mechanism to prevent the global pathologies that arise on high-dimensional embedding spaces, most notably representation collapse and equator concentration. **Polaris** uses an anisotropic spherical SVGD regularizer (Section 2.3.2) that explicitly counteracts these effects by combining a pole-attracting structural score with a vMF-kernel repulsion, producing the structured latitudinal and uniform longitudinal distributions shown in Figure 2. This kind of manifold-aware regularization is not part of the ComplEx or RotatE training pipeline.

**Summary.** ComplEx and RotatE are designed for general relational reasoning on flat ambient spaces and represent entities as deterministic point estimates, with hierarchy treated as just another binary relation. Polaris targets hierarchical taxonomy expansion directly: it operates on a curved hyperspherical manifold, separates hierarchical depth (radial) from semantic content (angular) by construction, models each concept as a distribution rather than a point so that semantic granularity becomes an explicit parameter, and regularizes the embedding distribution globally via SVGD. These choices reflect a different design objective and are not minor tweaks of the ComplEx or RotatE recipe.

## B. Benchmark Datasets

*Table 4.* Statistics of benchmark datasets. $|\mathcal{N}^0|$ and $|\mathcal{E}^0|$ denote the number of nodes and edges in the seed taxonomy, while $|D|$ is the taxonomy depth. For WordNet, values are averaged across 114 sub-taxonomies.

| Dataset | $|\mathcal{N}^0|$ | $|\mathcal{E}^0|$ | $|D|$ |
|---|---|---|---|
| **Single-parent taxonomies** | | | |
| SemEval-Env | 261 | 261 | 6 |
| SemEval-Sci | 429 | 452 | 8 |
| WordNet | 20.5 | 19.5 | 3 |
| **Multi-parent taxonomies** | | | |
| SemEval-Food | 1486 | 1533 | 8 |
| MeSH | 9710 | 10498 | 12 |
| Verb | 13936 | 13407 | 12 |

As discussed in Sections 3.1, 3.2, 3.3, we evaluate Polaris on three hierarchical inference settings covering **single-parent** trees, **multi-parent** DAG taxonomies, and a **multimodal** image-to-label hierarchy. Across all settings, we adopt the standard *attach-to-seed* protocol from prior taxonomy expansion work (Mishra et al., 2025; Jiang et al., 2022a; Shen et al., 2020b; Ma et al., 2021b): we form a *seed* hierarchy by withholding 20% of leaf nodes as queries, train using only relations observed in the seed, and at test time attach each query by ranking candidate parents drawn from the seed taxonomy. We ensure that each query's gold parent(s) remain in the seed, so evaluation measures attachment quality rather than missing candidates. Table 4 shows the dataset statistics.

**Single-parent taxonomies.** We evaluate single-parent taxonomy expansion on three public benchmarks: Environment (EN), Science (SCI), and WordNet. Each benchmark provides a human-curated tree-structured hierarchy where every node (except the root) has exactly one parent, aligning with the classical taxonomy completion setting.

**Multi-parent taxonomies.** We evaluate multi-parent taxonomy expansion on three benchmarks: SemEval-Verb (Verb), SemEval-Food (Food), and MeSH. SemEval-Verb contains a verb taxonomy derived from SemEval-2016 Task 14 (Jurgens & Pilehvar, 2016), which itself is a hierarchy of verbs from WordNet 3.0. SemEval-Food is taken from the SemEval taxonomy extraction/evaluation benchmarks introduced by Bordea et al. (2015). MeSH is constructed from the Medical Subject Headings controlled vocabulary (Lipscomb, 2000), a curated biomedical concept hierarchy where nodes may have multiple parents.

**Multimodal hierarchy (images).** For multimodal evaluation, we consider an image-to-label ranking task on the CUB-200-2011 dataset. Following standard practice, we sample a subset of 20 classes and apply an 80:20 train–test split within this subset. Each image is embedded using the standard open_clip preprocessing pipeline with a CLIP ViT-H/14 encoder pretrained on LAION-2B, yielding dense image representations. To construct label embeddings, for each class label $l \in \mathcal{L}$ we form a prompt of the form "A label of {class name}" and encode it with the same CLIP text encoder. This yields a fixed label set for retrieval, enabling direct evaluation of hierarchical inference from visual inputs.

## C. Evaluation Metrics

As discussed in Sections 3.1, 3.2, 3.3, we use the following evaluation metrics. Let $\mathcal{C}$ denote the set of query nodes (test instances). For each query $i \in \mathcal{C}$, let $\mathcal{Y}_i$ be the set of gold parent labels and $\hat{\mathcal{Y}}_i^{(k)}$ be the set of top-$k$ predicted labels.

**Hit@k.** Hit@k measures the fraction of queries for which a correct label appears among the top-$k$ predictions:

$$\text{Hit@}k \;=\; \frac{1}{|\mathcal{C}|} \sum_{i=1}^{|\mathcal{C}|} \mathbb{I}\Big( \mathcal{Y}_i \cap \hat{\mathcal{Y}}_i^{(k)} \neq \emptyset \Big). \tag{28}$$

For the single-parent case (i.e., $|\mathcal{Y}_i| = 1$), this is equivalent to checking whether the unique gold parent is in $\hat{\mathcal{Y}}_i^{(k)}$.

**Recall@k.** Recall@k measures the proportion of gold labels recovered within the top-$k$ predictions:

$$\text{Recall@}k \;=\; \frac{1}{|\mathcal{C}|} \sum_{i=1}^{|\mathcal{C}|} \frac{\Big| \{ y \in \mathcal{Y}_i : y \in \hat{\mathcal{Y}}_i^{(k)} \} \Big|}{|\mathcal{Y}_i|}. \tag{29}$$

Note that when $|\mathcal{Y}_i| = 1$, $\text{Recall@}k$ reduces to $\text{Hit@}k$.

**Mean Rank (MR).** Let $\text{rank}_i(y)$ denote the rank position (1 is best) of label $y$ in the model's sorted list for query $i$. In multi-parent hierarchies, we use the best-ranked gold parent:

$$\text{MR} \;=\; \frac{1}{|\mathcal{C}|} \sum_{i=1}^{|\mathcal{C}|} \min_{y \in \mathcal{Y}_i} \text{rank}_i(y). \tag{30}$$

**Mean Reciprocal Rank (MRR).**

$$\text{MRR} \;=\; \frac{1}{|\mathcal{C}|} \sum_{i=1}^{|\mathcal{C}|} \frac{1}{\min_{y \in \mathcal{Y}_i} \text{rank}_i(y)}. \tag{31}$$

**Wu & Palmer (Wu&P).** For single-parent trees, we additionally report Wu&P similarity between the predicted parent $c_1$ and gold parent $c_2$:

$$\text{Wu\&P}(c_1, c_2) \;=\; \frac{2 \cdot \text{depth}(\text{LCA}(c_1, c_2))}{\text{depth}(c_1) + \text{depth}(c_2)}, \tag{32}$$

where $\text{depth}(\cdot)$ is the distance from the root and $\text{LCA}(\cdot, \cdot)$ is the lowest common ancestor.

## D. Baselines

As discussed in Sections 3.1, 3.2, 3.3, we compare `Polaris` against a broad set of baselines used across our **text** (single-/multi-parent) and **multimodal** (image) settings, covering contextual encoders, taxonomy-expansion models, relational KGE methods, and geometric/non-Euclidean embeddings.

- **BERT+MLP** (Devlin et al., 2019) encodes term surface forms with BERT and applies an MLP to score parent–child (hypernym) relations.
- **TaxoExpan** (Shen et al., 2020a) represents anchor nodes by encoding their ego-networks with a GNN and scores candidate parent–child pairs via a log-bilinear feed-forward layer.
- **Arborist** (Manzoor et al., 2020) models heterogeneous edge semantics and optimizes a large-margin ranking objective with a dynamically adapting margin.
- **TMN** (Zhang et al., 2021a) introduces a triplet matching network to find the appropriate pairs for a given query concept consisting of a single primal scorer and multiple auxiliary scorers.
- **STEAM** (Yu et al., 2020b) samples mini-paths from the existing taxonomy and formulates a node attachment prediction task between anchor mini paths and query terms.
- **TransE** (Bordes et al., 2013a) learns translational relational embeddings and scores links with a distance-based energy function.
- **RotatE** (Sun et al., 2019) models relations as rotations in complex space, capturing diverse relational patterns through phase-based transformations.
- **HAKE** (Zhang et al., 2020b) encodes hierarchical structure using polar-coordinate embeddings, disentangling semantic similarity from hierarchical level to model asymmetric relations.

*Table 5.* **Ablation Study on SVGD Kernels.** Performance comparison of different kernel functions on MeSH and WordNet Verb datasets. **H@K**: Hit@K, **R@K**: Recall@K. The best scores are marked in **bold**.

| Kernel | MeSH | | | | | WordNet Verb | | | | |
|---|---|---|---|---|---|---|---|---|---|---|
| | H@1 | H@5 | R@1 | R@5 | MRR | H@1 | H@5 | R@1 | R@5 | MRR |
| Radial (RBF) | 31.12 | 54.18 | 25.07 | 47.24 | 37.40 | 13.10 | 36.54 | 13.07 | 35.65 | 23.92 |
| IMQ | 31.01 | 55.28 | 24.98 | 48.42 | 34.10 | 13.71 | 35.41 | 11.23 | 34.72 | 24.31 |
| vMF | 30.23 | 54.27 | 23.25 | 44.80 | 34.73 | 11.36 | 32.45 | 10.32 | 29.32 | 22.60 |
| **vMF + Score** | **33.59** | **62.66** | **27.03** | **51.95** | **38.72** | **15.05** | **39.20** | **15.01** | **38.77** | **26.34** |
| % ↑ | +7.94 | +13.35 | +7.82 | +7.29 | +3.53 | +9.78 | +7.28 | +14.85 | +8.76 | +8.35 |

- **HyperExpan** (Ma et al., 2021b) performs taxonomy expansion with hyperbolic representations to better capture hierarchical geometry and long-range ancestor–descendant structure
- **ConE** (Zhang et al., 2021b) embeds concepts as Cartesian products of two-dimensional cones where, the intersection and union of cones naturally model the conjunction and disjunction operations.
- **Box** (Jiang et al., 2023) learns box/region embeddings that capture partial order via overlap and containment, serving as a strong geometric baseline.
- **Gumbel Box** (Dasgupta et al., 2020) extends box embeddings with a Gumbel-based relaxation to improve optimization and robustness under uncertainty.
- **CLIP-1** (Radford et al., 2021b) uses a CLIP image–text encoder as a direct vision baseline by ranking class labels using CLIP similarity scores.
- **CLIP-2** (Radford et al., 2021b) is a stronger CLIP-based baseline (e.g., different backbone / pooling / prompting variant) that likewise ranks labels by CLIP similarity.

# E. Ablation Studies

As discussed in Section 3, we provide detailed explanations and results of all experiments conducted.

### E.1. Experiments on SVGD Kernels

We perform a comprehensive ablation study to validate the design choices of the `Polaris` regularization scheme. The choice of kernel $k(\mathbf{z}, \mathbf{z}')$ dictates the topology of the repulsive forces between embeddings, which is critical for preventing mode collapse on the compact hyperspherical manifold. We first evaluate two Euclidean baselines: the Radial Basis Function (RBF), which relies on chordal distances, and the heavy-tailed Inverse Multi-Quadratic (IMQ) kernel, which tests the efficacy of long-range repulsive forces. We then assess the benefit of manifold-consistent interactions by evaluating the von Mises-Fisher (vMF) kernel, which ensures that particle interactions respect geodesic distances. Finally, we evaluate the full formulation combining the vMF repulsive kernel with the proposed structural score function $\nabla \log p_{\text{struct}}$. This final ablation isolates the contribution of the pole-attracting prior, verifying that uniformity alone (via repulsion) is insufficient without the explicit hierarchical breaking of symmetry provided by the structural score. As shown in Table 5, the results clearly indicate that the vMF kernel combined with the structural score function provides superior regularization compared to the standard RBF or IMQ kernels, empirically validating the necessity of both manifold-aware interactions and an explicit hierarchical prior.

### E.2. Experiments on vMF with constant $\kappa$ and $\mu$ in Probabilistic Learning

In `Polaris`, the probabilistic embedding for a concept is parameterized by $\text{vMF}(\boldsymbol{\mu}, \kappa)$, where both the mean direction $\boldsymbol{\mu}$ and the concentration $\kappa$ are predicted as functions of embedding $\mathbf{z}$ via learnable spherical projections. We perform experiments to isolate the contribution of the adaptive method. First, we keep $\boldsymbol{\mu}$ constant by setting it as $z$. This ablation evaluates whether the probabilistic centroid of a concept needs to diverge from its geometric position on the manifold. Second, we also keep $\kappa$ constant to a global scalar constant C. In our experiment we set $C = 0.5$. This experiment tests the hypothesis that hierarchical concepts exhibit varying degrees of semantic granularity. We show that capturing semantic diversity between concepts is essential for representing the varying volume of concepts across hierarchical levels. Table 6 describes the results of vMF experiments on MeSH and Verb. The results indicate that keeping both $\boldsymbol{\mu}$ and $\kappa$ performs better

*Table 6.* **Ablation on Probabilistic Parameterization.** We evaluate the impact of learning adaptive distributions versus fixed parameters. **Constant** $\kappa$: Fixed concentration $\kappa = 0.5$. **Identity** $\mu$: Mode fixed to geometric embedding $\mu = \mathbf{z}$. Best scores are marked in **bold**.

| Configuration | MeSH | | | | | WordNet Verb | | | | |
|---|---|---|---|---|---|---|---|---|---|---|
| | H@1 | H@5 | R@1 | R@5 | MRR | H@1 | H@5 | R@1 | R@5 | MRR |
| Constant $\kappa$ ($\kappa = 0.5$) | 31.57 | 62.32 | 26.69 | 51.76 | 38.22 | 13.21 | 35.93 | 12.32 | 35.16 | 24.76 |
| Identity $\mu$ ($\mu = \mathbf{z}$) | 31.14 | 60.78 | 25.07 | 52.39 | 35.57 | 13.20 | 35.72 | 13.12 | 35.47 | 23.65 |
| **Learnable $\mu$ and $\kappa$** | **33.59** | **62.66** | **27.03** | **53.66** | **38.91** | **15.05** | **39.20** | **15.01** | **38.77** | **26.34** |
| % ↑ | +6.40 | +0.55 | +1.27 | +2.42 | +1.80 | +13.93 | +9.10 | +14.41 | +9.30 | +6.38 |

*Table 7.* **Ablation on Target Anchoring in SVGD.** We compare the proposed learned semantic anchor against a self-targeting baseline where the target mode is fixed to the current embedding ($\mu = \mathbf{z}$). The best scores are marked in **bold**.

| Configuration | MeSH | | | | | WordNet Verb | | | | |
|---|---|---|---|---|---|---|---|---|---|---|
| | H@1 | H@5 | R@1 | R@5 | MRR | H@1 | H@5 | R@1 | R@5 | MRR |
| Self-Targeting | 30.89 | 61.57 | 24.87 | 52.31 | 36.63 | 11.97 | 35.01 | 11.75 | 33.50 | 22.24 |
| **Learned Anchor** | **33.59** | **62.66** | **27.03** | **54.97** | **38.65** | **15.05** | **39.20** | **15.01** | **38.77** | **26.34** |
| % ↑ | +8.74 | +1.77 | +8.68 | +5.09 | +5.51 | +25.74 | +11.97 | +27.74 | +15.73 | +18.44 |

than keeping one of the parameters constant. Comparing the constraints reveals distinct failure modes: the performance drop in Identity $\mu$ suggests that the optimal probabilistic centroid must diverge from the geometric coordinate $\mathbf{z}$ to correct for structural regularization artifacts. Conversely, the $\kappa$ baseline performs slightly worse because it enforces uniform semantic volume, reducing the model's capability of distinguishing between broad, high-entropy parent concepts and narrow, low-entropy leaf nodes.

### E.3. Experiments on Target Distribution Anchoring in SVGD

In the proposed SVGD, the target score function $\nabla \log p(\mathbf{z})$ comprises two competing vector fields: a structural drift and semantic alignment. The alignment term is defined as $\nabla_{\mathbf{z}} \log p_{\text{align}}(\mathbf{z}) = \kappa_{\text{align}} \mu$ where $\mu$ is the predicted probabilistic center. To assess the necessity of this learnable anchor, we set the model of the local target distribution to be the current geometric position of the embedding itself. This experiment tests whether the regularization needs an external semantic reference. Table 7 consists of experiments with learnable and fixed anchors $\mu$ on MeSH and Verb, indicating that learnable anchors outperform fixed semantic anchors. This is probably because if the target is centered at $\mathbf{z}$, the alignment gradient becomes proportional to $\mathbf{z}$. Crucially, the projection of this vector onto the tangent space is zero. Without this restoring force, the structural force $\nabla \log p_{\text{struct}}$ and the repulsive kernel $\nabla k$ dominate the dynamics. We hypothesize that this leads to semantic drift where embeddings satisfy hierarchical depth constraints but lose their angular distinctiveness, resulting in a slight degradation in performance.

### E.4. Component Analysis of the SVGD Gradient Field

The SVGD update vector $\phi(z)$ is composed of two distinct gradient fields: the drift field derived from the target distribution (weighted by $\kappa_{\text{align}}$) and the repulsive field derived from the kernel gradients (weighted by $\kappa_{\text{repel}}$). To decouple their contributions, we evaluate the model by selectively masking these terms in the update equation. The results demonstrate that removing the repulsive term leads to a performance drop i.e MeSH H@1 decreases from 33.59 to 23.82. Mathematically, when $\kappa_{\text{repel}} \to 0$, the Stein operator degenerates into standard gradient ascent on the log-density $\log p(z)$. Without the kernel gradient $\nabla_{z'} k(z', z)$ acting as a dispersive force, the particles collapse toward the modes of the target distribution (the semantic anchors $\mu$). This results in a degenerate embedding distribution with minimal variance along the longitudinal manifold $\mathbb{S}^{d-2}$, effectively reducing the rank of the representation and causing indistinguishability among sibling nodes. Now, removing the alignment term also degrades performance, though less severely. In this regime, optimization is driven solely by the structural score and kernel repulsion. While the repulsive component ensures uniform coverage of the sphere, the absence of the alignment tether $\kappa_{\text{align}} \mu$ allows embeddings to drift geodesically away from their semantic initialization. This semantic drift implies that although the geometric structure remains well-formed (i.e., high manifold coverage), the correspondence to the input feature space is distorted, leading to suboptimal retrieval performance. Table 8 explains these

*Table 8.* **Ablation on SVGD Components.** We analyze the contribution of the Alignment Drift ($\kappa_{\text{align}}$) and Kernel Repulsion ($\kappa_{\text{repel}}$). The **Full Model** utilizes both. Best scores are marked in **bold**.

| | MeSH | | | | | WordNet Verb | | | | |
| --- | --- | --- | --- | --- | --- | --- | --- | --- | --- | --- |
| Configuration | H@1 | H@5 | R@1 | R@5 | MRR | H@1 | H@5 | R@1 | R@5 | MRR |
| w/o Repulsion ($\kappa_{\text{repel}} = 0$) | 23.82 | 46.70 | 19.20 | 43.25 | 29.82 | 12.79 | 33.88 | 12.77 | 32.91 | 23.00 |
| w/o Alignment ($\kappa_{\text{align}} = 0$) | 30.90 | 53.60 | 24.89 | 46.79 | 34.44 | 12.18 | 32.65 | 11.86 | 31.88 | 21.83 |
| w/o Both | 32.25 | 58.88 | 26.85 | 50.95 | 36.23 | 14.12 | 35.00 | 13.77 | 32.57 | 23.21 |
| **Full Model** | **33.59** | **62.66** | **27.03** | **51.95** | **37.85** | **16.27** | **39.71** | **16.24** | **39.63** | **27.03** |
| % ↑ | +4.16 | +6.4 | +0.67 | +1.96 | +4.47 | +15.23 | +13.46 | +17.93 | +20.45 | +16.46 |

results in detail.

### E.5. Sensitivity Analysis on Alignment Concentration $\kappa_{\text{align}}$

The parameter $\kappa_{\text{align}}$ governs the magnitude of the drift term derived from the semantic target distribution, $\nabla \log p_{\text{align}}(z) = \kappa_{\text{align}} \mu$, which balances the optimization objective by ensuring that embeddings remain semantically consistent with their learned anchors $\mu$, while the structural score and kernel gradients redistribute them geometrically. We evaluate the robustness of the model by varying $\kappa_{\text{align}} \in \{1, 2, 3, 4, 5\}$ on MeSH. As shown in Table 9, the method exhibits stability across this range, indicating that the regularization framework is not brittle with respect to hyperparameter tuning. Lower values (e.g., $\kappa_{\text{align}} = 1$) yield slightly reduced retrieval performance, suggesting that a weak alignment signal allows the structural gradients to overly distort the semantic placement. Conversely, higher values (e.g., $\kappa_{\text{align}} = 5$) constrain the embeddings tightly to the mean $\mu$, potentially limiting the geometric adjustments required for optimal hierarchical separation. Empirically, $\kappa_{\text{align}} = 3$ provides the optimal trade-off, maximizing ranking metrics (MRR) while maintaining high recall.

*Table 9.* **Sensitivity Analysis of $\kappa_{\text{align}}$ on MeSH.** Shading intensity reflects performance degradation relative to the best value in each column i.e darker gray implies higher degradation.

| $\kappa_{\text{align}}$ | H@1 | H@5 | R@1 | R@5 | MRR |
| --- | --- | --- | --- | --- | --- |
| 1 | 33.25 | 61.01 | **27.79** | 51.54 | 38.36 |
| 2 | 32.32 | **62.02** | 26.33 | **54.12** | 38.40 |
| 3 | 33.71 | 61.80 | 27.15 | 52.57 | **38.77** |
| 4 | 33.60 | 61.01 | 27.06 | 52.31 | 38.21 |
| 5 | **34.61** | 60.00 | 27.88 | 51.04 | 38.42 |

### E.6. Sensitivity Analysis on Repulsion Strength $\kappa_{\text{repel}}$

The parameter $\kappa_{\text{repel}}$ controls the strength of the kernel-based repulsive interactions within the SVGD update, which are responsible for enforcing geometric diversity among embeddings on the hypersphere. Specifically, it scales the influence of the vMF kernel $k(z', z) = \exp(\kappa_{\text{repel}} z'^{\top} z)$ thereby determining the magnitude of the repulsive force that prevents particle collapse and promotes uniform angular coverage. We examine the effect of varying $\kappa_{\text{repel}} \in \{0.5, 1.5, 2.5, 3.5, 4.5\}$ on MeSH. As reported in Table 10, performance improves consistently as $\kappa_{\text{repel}}$ increases, reflecting stronger enforcement of angular separation and improved utilization of longitudinal capacity. Lower values yield insufficient repulsion, leading to partial particle clustering and degraded retrieval metrics. Conversely, excessively large values begin to saturate gains, indicating diminishing returns once sufficient geometric diversity is achieved. Empirically, $\kappa_{\text{repel}} \approx 3.5–4.5$ provides the best trade-off between separation and semantic stability, yielding peak MRR and recall across datasets.

### E.7. Ablation on Geometric and Probabilistic Loss Components

To quantify the individual contributions of the geometric regularization and probabilistic alignment objectives, we evaluate the model under three configurations: (i) removing the geometric loss (w/o Geometric), (ii) removing the probabilistic vMF-based loss (w/o Probabilistic), and (iii) employing both components jointly. The geometric loss enforces strict manifold consistency by forcing the alignment of parent and child directions by minimizing the geodesic distance on the sphere,

*Table 10.* **Sensitivity Analysis of $\kappa_{\mathrm{repel}}$ on MeSH.** Darker shading indicates increased performance degradation from the optimal repulsion strength.

| $\kappa_{\mathrm{repel}}$ | H@1 | H@5 | R@1 | R@5 | MRR |
|---|---|---|---|---|---|
| 0.5 | 28.54 | 53.93 | 22.99 | 45.34 | 33.01 |
| 1.5 | 26.40 | 56.40 | 21.45 | 47.78 | 33.61 |
| 2.5 | 29.32 | 58.65 | 23.62 | 50.14 | 35.57 |
| 3.5 | 31.57 | **60.22** | 25.43 | **51.58** | 36.93 |
| 4.5 | **34.61** | 60.00 | **27.87** | 51.04 | **37.99** |

*Table 11.* **Ablation Study on Loss Components across MeSH and Verb datasets.** Best results are marked in **bold**.

| Configuration | MeSH | | | | | WordNet Verb | | | | |
|---|---|---|---|---|---|---|---|---|---|---|
| | H@1 | H@5 | R@1 | R@5 | MRR | H@1 | H@5 | R@1 | R@5 | MRR |
| w/o Geometric | 28.31 | 54.00 | 23.26 | 43.98 | 33.71 | 9.31 | 26.41 | 9.29 | 26.35 | 17.07 |
| w/o Probabilistic | 28.20 | 54.94 | 22.35 | 45.88 | 33.19 | 11.37 | 27.23 | 10.38 | 27.17 | 18.61 |
| w/o Both | 32.25 | 58.88 | 26.85 | 50.95 | 36.23 | 14.12 | 35.00 | 13.77 | 32.57 | 23.21 |
| **Full Model** | **33.59** | **62.66** | **27.03** | **51.95** | **37.85** | **16.27** | **39.71** | **16.24** | **39.63** | **27.03** |
| % ↑ | +4.16 | +6.40 | +0.67 | +1.96 | +4.47 | +15.23 | +13.46 | +17.93 | +20.45 | +16.46 |

while the probabilistic component aligns embeddings with semantic anchors via the vMF distribution. As shown in Table 11, removing either component results in performance degradation across both MeSH and Verb datasets. Excluding the geometric component leads to the model not being consistent with the geometry, whereas excluding the probabilistic objective induces semantic drift despite maintaining geometric spread. The full model consistently outperforms both ablated variants, confirming that accurate hierarchical representation requires the joint optimization of geometric structure and probabilistic semantic alignment.

### E.8. Ablation on Orbital Gate Potential

To quantify the orbital potential component, we conduct experiments with and without the gated potential on the overall objective and on each loss component and mention them in table 12. The gate consistently improves performance, confirming gains stem from decoupling rather than auxiliary accumulation. Improvements in experiments with geometric loss only indicate better hierarchical structuring and vMF-only gains reflect superior semantic alignment. Combining both yields optimal results. Further, even without the orbital gate, `Polaris` outperforms Euclidean baselines, showing learned geometry inherently captures hierarchy.

## F. Analysis of $\theta$ and $\psi$ Distributions of `Polaris`

As discussed in Section 2.3.2, we analyze the distributions of $\theta$ and $\psi$ on MeSH for all candidates with and without SVGD in figures 2 and 3 respectively. In Figure 3 the latitudinal angles show extreme kurtosis, indicating that optimization degenerates into a narrowband leaving polar regions empty, effectively nullifying radial capacity. Likewise, the longitudinal coordinate fails to achieve uniformity by clustering at the boundaries and thus limiting semantic separability. In contrast, figure 2 demonstrates a complete utilization of the hyperspherical volume. Despite the varying available surface area at different depths, the repulsive forces successfully maximize entropy within the semantic subspace, ensuring that sibling concepts remain distinguishable across all levels.

## G. Prior and Related Works

As discussed in Section 1, prior and related works to polar embeddings are as follows.

**Concept learning using Polar Coordinates.** Prior polar and angular embedding approaches typically rely on coordinate-wise optimization rather than intrinsic manifold geometry. (Iwamoto et al., 2021) learn longitudinal and latitudinal angles through distance-based losses defined directly on angular coordinates and enforce constraints via modulus operations. While effective in practice, this procedure treats the angular domain as a flat parameter space and induces geometric distortions

*Table 12.* **Effect of orbital potential gating.** Gating consistently improves retrieval quality across datasets and objective variants, showing that adaptive structural modulation strengthens both geometric and directional representations. Best results are shown in **bold**.

| Setting | Variant | Recall@1 | | MRR | |
|---|---|---|---|---|---|
| | | **Score** | **Δ** | **Score** | **Δ** |
| *Full model* | | | | | |
| Environment | ungated | 46.3 | | 56.3 | |
| Environment | gated | **51.9** | +5.6 | **60.8** | +4.5 |
| MeSH | ungated | 24.6 | | 36.5 | |
| MeSH | gated | **27.1** | +2.5 | **37.7** | +1.2 |
| *Objective ablation on MeSH* | | | | | |
| Geometric only | ungated | 20.5 | | 30.7 | |
| Geometric only | gated | **22.8** | +2.3 | **34.2** | +3.5 |
| vMF only | ungated | 21.8 | | 32.2 | |
| vMF only | gated | **23.1** | +1.3 | **33.4** | +1.2 |

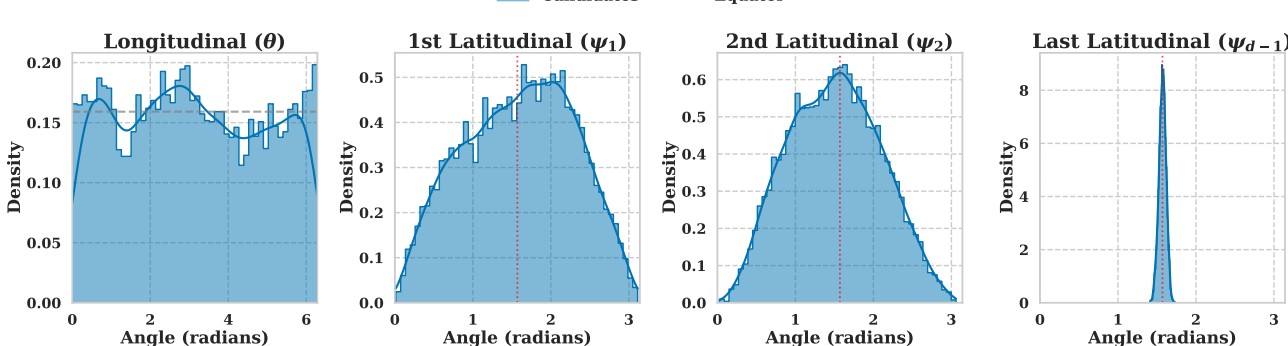

*Figure 2.* **Global Distributional Analysis with SVGD on MeSH.** Histograms showing the frequency of learned Angular coordinates $(\theta, \psi)$ for all concepts in the DAG. The uniform spread of the Longitudinal Angle (leftmost) confirms that the SVGD regularizer successfully prevents collapse, while the Latitudinal angles exhibit structural peaking consistent with hierarchical depth.

that ignore the curvature of the hypersphere. (Atzeni et al., 2023) extend polar formulations to hierarchical representation learning by modeling concepts as box regions on the hypersphere; however, their latitudinal coordinate $\psi$ is generated through a scaled sigmoid transformation, which does not correspond to a valid hyperspherical parameterization and breaks intrinsic manifold consistency. HAKE (Zhang et al., 2020b) introduces a polar-coordinate embedding framework that decomposes entity representations into modulus and phase components, where the modulus captures hierarchical depth and the phase models relational semantics. Although HAKE successfully encodes hierarchical ordering, it operates within a Euclidean polar space and relies on hard geometric constraints, limiting its ability to fully exploit intrinsic manifold structure and curvature-aware optimization. (Gregucci et al., 2023) proposed an attention-based framework that integrates the query representations from multiple diverse knowledge graph embedding models into a unified embedding. By learning a relation-specific attention mechanism, their model can dynamically weight the contribution of each base model for a given query, allowing it to adapt to heterogeneous patterns. Furthermore, they extend this combination to non-Euclidean spaces by projecting the unified query onto a Poincaré ball to better model hierarchical structure. However, it inherits the geometric priors of its constituent models, hierarchy is primarily captured by projecting a fused Euclidean vector into a hyperbolic space, rather than being an intrinsic, disentangled component of the representation itself. Further, it does not explicitly regularize the global distribution of embeddings to prevent pathologies like representation collapse or anisotropy, relying instead on the implicit geometries of the base models. ***In contrast, `Polaris` adopts an intrinsic hyperspherical formulation in which embeddings evolve directly on the manifold through tangent-space projections and exponential maps, preserving the spherical line element throughout optimization. By coupling vMF-based semantic alignment with geometry-aware SVGD dynamics and an explicit uniform occupancy prior, `Polaris` achieves principled manifold-consistent hierarchical representations while avoiding the coordinate distortions and heuristic constraints present in prior polar approaches.***

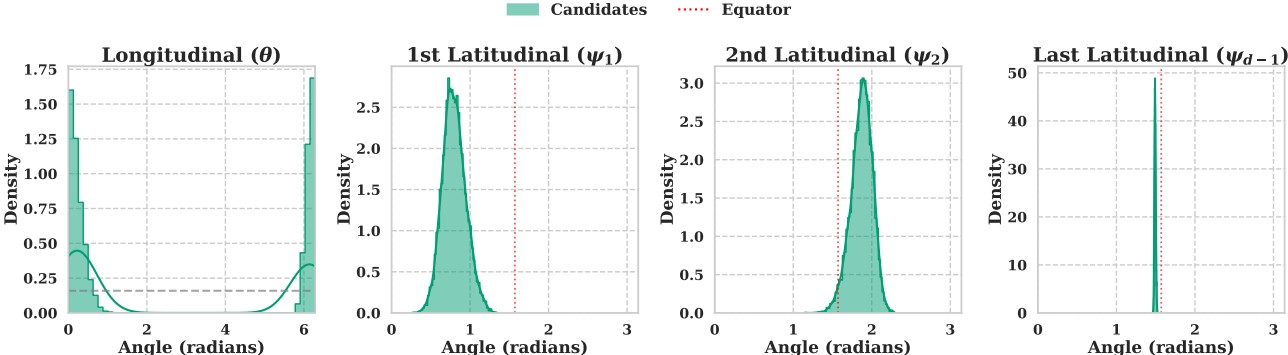

*Figure 3.* **Evidence of Equatorial Collapse on MeSH.** Global angle distributions learned *without* the proposed SVGD regularization. In the absence of the structural score and repulsive kernel, the latitudinal angles (rightmost plots) exhibit a sharp concentration of probability mass around the equator. Unlike the `Polaris` model which populates the poles to encode depth, this unregularized model suffers from the high-dimensional concentration of measure phenomenon, where embeddings degenerate into a narrow orthogonal band, effectively wasting the volume of the hypersphere.

**Hyperspherical Learning.** (Mettes et al., 2019) proposed a prototype-based framework where, instead of learning the positions of class centroids, they are fixed a priori on the hypersphere to guarantee maximal angular separation. The learning task is thus simplified to optimizing a direct mapping from inputs to these static targets via a squared cosine loss. (Loshchilov et al., 2025) constrains all vectors including token embeddings, linear projection weights, attention matrices and hidden states to lie on a hypersphere during learning by normalization after each operation and (Karagodin et al., 2026) studies normalization schemes on token representations as interacting particles on the sphere, revealing how they influence clustering dynamics and representation collapse. (Trosten et al., 2023) formally shows that uniformly distributing embeddings on the unit hypersphere eliminates hubness by ensuring both zero mean and vanishing density gradients across all tangent directions. (Bendada et al., 2025) introduced a technique that leverages hyperspherical embedding vectors and sampling from the von Mises–Fisher distribution to efficiently explore large action spaces. Unlike Boltzmann exploration, which scales poorly with the number of actions, it uses hyperspherical sampling to approximate exploration probabilities while requiring only nearest neighbor operations. (Zhang & Chen, 2026) learns angular representations on the hypersphere. Their anchor-free SpherePair loss uses cosine similarity to enforce pairwise constraints, theoretically guaranteeing that clusters form an equidistant regular simplex. ***Building on these ideas, `Polaris` combines intrinsic hyperspherical encoding, vMF-based semantic alignment, and geometry-preserving tangent-space updates with a uniform manifold prior to ensure unbiased, structured representation learning.***

## H. Cartesian-Angular Mapping

**Definition H.1** (Cartesian-to-Angular Mapping). There exists a mapping $\Phi : \mathbb{S}^{d-1} \setminus \mathcal{S} \to [0, \pi]^{d-2} \times [0, 2\pi)$ transforming a Cartesian vector $\mathbf{z} = (z_1, \ldots, z_d)$ into angular coordinates $\phi = (\psi_1, \ldots, \psi_{d-2}, \theta)$. This mapping is defined recursively:

$$\psi_i = \arccos\left(\frac{z_i}{\sqrt{\sum_{j=i}^{d} z_j^2}}\right), \quad 1 \leq i \leq d-2$$

$$\theta = \begin{cases} \arccos\left(\frac{z_{d-1}}{\sqrt{z_{d-1}^2 + z_d^2}}\right) & \text{if } z_d \geq 0 \\ 2\pi - \arccos\left(\frac{z_{d-1}}{\sqrt{z_{d-1}^2 + z_d^2}}\right) & \text{if } z_d < 0 \end{cases}$$

$\theta, \psi$ represent the longitudinal and latitudinal angles, respectively. $\theta$ is defined on the full circle, while $\psi_i$ is defined on the half circle, where $i$ represents components from 1 to $d-2$.

# I. Theoretical Limitations of Angular Parameterization

In this section, we formally analyze the geometric deficiencies inherent in prior polar embedding methods. We categorize these limitations into three distinct pathologies that arise when optimizing angular coordinates directly.

## I.1. Coordinate Singularities

The mapping from angular parameters to the spherical manifold is not a global diffeomorphism. It contains singularities where the coordinate system collapses.

**Definition I.1** (Local Identifiability of the Embedding Space). Let $\mathcal{Z}$ be the geometric embedding space and $\{P_\theta : \theta \in \mathcal{Z}\}$ be the family of model distributions. The model is said to be locally identifiable at $\theta_0 \in \mathcal{Z}$ if there exists an open neighborhood $U \subseteq \mathcal{Z}$ containing $\theta_0$ such that:

$$P_{\theta_1} = P_{\theta_2} \implies \theta_1 = \theta_2, \quad \forall \theta_1, \theta_2 \in U. \tag{33}$$

**Theorem I.2** (Jacobian Singularity of Angular Coordinates). *Let $\Psi : [0, \pi]^{d-2} \times [0, 2\pi) \to \mathbb{S}^{d-1}$ be the coordinate transformation mapping angles $\phi = (\psi_1, \ldots, \psi_{d-1})$ to a Cartesian vector $\mathbf{z}$. The Jacobian matrix $\mathbf{J}_\Psi(\psi) = \frac{\partial \mathbf{z}}{\partial \psi}$ becomes singular whenever any latitudinal angle becomes $\psi_k \in \{0, \pi\}$ for $1 \leq k \leq d-2$.*

*Proof.* The mapping of angular coordinates to the rectangular coordinates on the unit hypersphere $\mathbb{S}^{d-1}$ is as follows,

$$z_1 = \cos(\psi_1), \tag{34}$$

$$z_2 = \sin(\psi_1)\cos(\psi_2), \tag{35}$$

$$\vdots \tag{36}$$

$$z_k = \left( \prod_{j=1}^{k-1} \sin(\psi_j) \right) \cos(\psi_k), \tag{37}$$

$$\vdots \tag{38}$$

$$z_d = \left( \prod_{j=1}^{d-2} \sin(\psi_j) \right) \sin(\theta). \tag{39}$$

The volume element of a unit hypersphere $\mathbb{S}^{d-1} \subset \mathbb{R}^d$ is given by,

$$d\,\mathrm{Vol}_{S^{d-1}} = \sin^{d-2}(\psi_1) \sin^{d-3}(\psi_2) \cdots \sin(\psi_{d-2})\, d\psi_1\, d\psi_2 \cdots d\psi_{d-2}\, d\theta \tag{40}$$

From, (34), the Jacobian determinant factor is obtained as follows,

$$\sqrt{\det(\mathbf{J}^T \mathbf{J})} = \prod_{j=1}^{d-2} \sin^{d-1-j} \psi_j \tag{41}$$

The singularity of the transformation between Cartesian and angular coordinates is determined by the rank of the Jacobian $\mathbf{J}_\Psi$. For the mapping to be full rank, the volume element must be non-zero. For any latitudinal coordinate $\psi_k \in \{0, \pi\}$, $\sin \psi_k = 0$. The exponent of the power series in (41) is $p = d - 1 - k$ where $k$ ranges between $0 \leq k \leq d - 2$. Therefore, the minimum value of the exponent $p_{\min}$ is,

$$p_{\min} = (d-1) - (d-2) = 1 \tag{42}$$

This means $\sin \psi$ always exists in the product. Substituting $\sin \psi_k = 0$ in the determinant term,

$$\sqrt{\det(\mathbf{J}^\top \mathbf{J})} = \left( \cdots \underbrace{\left( \sin(\psi_k) \right)}_{0}^{d-1-k} \cdots \right) = 0 \tag{43}$$

Therefore, the Jacobian $\mathbf{J}_\Psi$ loses full column rank indicating that angular sampling biases embeddings. $\square$

Theorem $I.2$ plays a significant role in learning polar coordinates. At points of singularity, variations in specific angular parameters result in zero displacement in the embedding space, rendering the space no longer locally identifiable.

Further, if the Jacobian $\mathbf{J}_\Psi(\phi) = \frac{\partial \mathbf{z}}{\partial \phi}$ is singular, it projects the gradients into the null space effectively nullifying the learning signal of the model parameters associated with the loss functions, causing the training process to suffer from numeric instabilities.

### I.2. Superficial Similarity

A common heuristic for polar embedding learning involves optimizing a separable loss function on the angular coordinates. Let $\mathcal{L}_{\text{sep}}$ be a composite objective (e.g., using Welsch loss $\rho$) that minimizes the latitudinal difference and the shortest longitudinal wrap-around difference independently:

$$\mathcal{L}_{\text{sep}}(\phi_i, \phi_j) = \rho(|\psi_i - \psi_j|) + \rho(\min(|\theta_i - \theta_j|, \ 2\pi - |\theta_i - \theta_j|)) \tag{44}$$

Optimizing angles directly as a combination of losses for longitude and latitude is analogous to minimizing distances on a parameter grid rather than on a spherical manifold. This inherently defines the geometry of a flat Euclidean lattice wrapped along one axis. This cylindrical model holds true at the equator, where $\psi = \frac{\pi}{2}$. The manifold is maximally wide, and Euclidean approximations are valid, $\sin \psi \approx 1$. However, this model collapses as $\psi$ approaches the poles. On a sphere, the grid lines of the longitude converge at the poles, whereas on a cylinder, the lines run parallel and equidistant to each other. Therefore, the model ends up attempting to make large gradient updates for small semantic differences at the poles, resulting in negligible improvement in performance and true semantic alignment of concepts. We show this formally in statements I.3 and I.4.

**Theorem I.3** (Implicit Cylindrical Geometry of Angular Differences). *The separable objective function $\mathcal{L}_{\text{sep}}$ implies an underlying Riemannian manifold $\mathcal{M}_{param}$ equipped with a flat Euclidean metric tensor $\mathbf{G}_{param} = \mathbf{I}$. This geometry corresponds to a rectilinear grid on the parameter domain $[0, \pi] \times [0, 2\pi)$.*

*Topologically, due to the periodicity of $\theta$, this manifold is isomorphic to a cylinder $\mathcal{C} = [0, \pi] \times \mathbb{S}^1$ with zero Gaussian curvature. This stands in direct contradiction to the intrinsic geometry of the hypersphere $\mathbb{S}^{d-1}$, which possesses constant positive curvature and a metric tensor coupled by the sine of the latitude: $ds^2 = d\psi^2 + \sin^2(\psi)d\theta^2$.*

*Proof.* Consider the separable loss to be,

$$\mathcal{L}_{\text{sep}}(\phi_i, \phi_j) = \rho(|\psi_i - \psi_j|) + \rho(\min(|\theta_i - \theta_j|, \ 2\pi - |\theta_i - \theta_j|)) \tag{45}$$

where the parameterization $\phi = (\psi, \theta) \in [0, \pi] \times [0, 2\pi)$. Assume $\rho$ is smooth and strictly increasing with $\rho(0) = 0$. Let $\phi = (\psi, \theta)$ and $\phi' = (\psi + d\psi, \theta + d\theta)$. For sufficiently small displacements $d\psi, d\theta$, we have

$$\min(|\theta - \theta'|, \ 2\pi - |\theta - \theta'|) = |d\theta|.$$

Assume $\rho$ is twice continuously differentiable with $\rho(0) = 0$ and $\rho'(0) = 0$. Its second-order Taylor expansion about 0 is

$$\rho(x) = \frac{1}{2}\rho''(0)\, x^2 + o(x^2) \tag{46}$$

Applying this expansion to each coordinate difference yields

$$\rho(|d\psi|) = \frac{1}{2}\rho''(0)\,(d\psi)^2 + o((d\psi)^2) \tag{47}$$

$$\rho(|d\theta|) = \frac{1}{2}\rho''(0)\,(d\theta)^2 + o((d\theta)^2) \tag{48}$$

Therefore, for infinitesimally close points,

$$\mathcal{L}_{\text{sep}}(\phi, \phi + d\phi) = \frac{1}{2}\rho''(0)\Big((d\psi)^2 + (d\theta)^2\Big) + o(\|d\phi\|^2) \tag{49}$$

Up to the positive scalar factor $\frac{1}{2}\rho''(0)$ the induced squared line element is therefore,

$$ds^2 = d\psi^2 + d\theta^2 \tag{50}$$

Hence, the metric tensor on $\mathcal{M}_{\text{param}}$ is,

$$\mathbf{G}_{\text{param}} = \begin{pmatrix} 1 & 0 \\ 0 & 1 \end{pmatrix} = \mathbf{I}. \tag{51}$$

which is constant and diagonal. The Riemann curvature tensor is identically zero. Therefore, the induced geometry is flat. Since the coordinate $\theta$ is periodic with period $2\pi$, the parameter space is topologically equivalent to the cylinder. (50) shows that the coordinate $\psi \in [0, \pi]$ moves along the height from 0 to $\pi$. Geometrically, this indicates a band with periodic boundary conditions in one direction, resembling a finite cylinder,

$$\mathcal{M}_{\text{param}} \cong [0, \pi] \times \mathbb{S}^1$$

A cylinder is locally isometric to the Euclidean plane and has zero Gaussian curvature. By contrast the metric of the hypersphere $\mathbb{S}^{d-1}$ in spherical coordinates is given by,

$$ds^2 = d\psi^2 + \sin^2(\psi)d\theta^2 \tag{52}$$

which explicitly couples longitudinal and latitudinal directions and yields constant positive curvature. Since $\mathcal{L}_{\text{sep}}$ induces a metric without such coupling, it cannot recover the intrinsic geometry of $\mathbb{S}^{d-1}$. $\square$

**Corollary I.4** (Unbounded Distortion at the Poles). *The mismatch between the induced cylindrical metric and the true spherical metric results in a distortion ratio $\mathcal{D}$ that diverges at the singularities. Let $\delta_{\mathcal{L}}$ be the gradient magnitude induced by the loss and $\delta_{geo}$ be the true geodesic gradient for a longitudinal displacement $d\theta$:*

$$\lim_{\psi \to 0} \mathcal{D}(\psi) = \lim_{\psi \to 0} \frac{\delta_{\mathcal{L}}}{\delta_{geo}} \propto \lim_{\psi \to 0} \frac{1}{\sin(\psi)} = \infty \tag{53}$$

*Thus, the optimization landscape treats the poles not as points, but as expanded circles of circumference $2\pi$, forcing the model to minimize distances that do not exist on the manifold.*

*Proof.* We know that a map $f : (X, d_X) \to (Y, d_Y)$ is bi-Lipschitz if there exist constants $0 < c \leq C < \infty$ such that

$$c\, d_X(x, x') \leq d_Y(f(x), f(x')) \leq C\, d_X(x, x') \quad \forall x, x' \in X.$$

Fix $\psi > 0$ and consider two points differing only in longitude,

$$p_\psi = (\psi, 0), \qquad q_\psi = (\psi, \varepsilon), \quad 0 < \varepsilon \ll 1.$$

Under the cylindrical metric,

$$d_{\text{cyl}}(p_\psi, q_\psi) = \varepsilon.$$

Under the spherical metric,

$$d_{\text{sphere}}(p_\psi, q_\psi) = \sin \psi\, \varepsilon.$$

Hence, the distortion ratio is

$$\frac{d_{\text{cyl}}(p_\psi, q_\psi)}{d_{\text{sphere}}(p_\psi, q_\psi)} = \frac{1}{\sin \psi}.$$

As $\psi \to 0$ (or $\psi \to \pi$), $\sin \psi \to 0$ and the ratio diverges. Therefore, no finite Lipschitz constant exists. Now, let $L(\psi, \theta)$ be a smooth loss function. The Riemannian gradient under a metric $g$ is given by

$$\nabla_g L = g^{-1} \nabla L.$$

For the spherical metric,

$$g_{\text{sphere}}^{-1} = \begin{pmatrix} 1 & 0 \\ 0 & \sin^{-2}\psi \end{pmatrix},$$

so the longitudinal component of the gradient is

$$(\nabla_{\text{sphere}} L)_\theta = \frac{1}{\sin^2 \psi} \partial_\theta L.$$

Under the cylindrical metric,

$$g_{\text{cyl}}^{-1} = \mathbf{I}, \qquad (\nabla_{\text{cyl}} L)_\theta = \partial_\theta L.$$

As $\psi \to 0$, the spherical geometry enforces $\partial_\theta L \to 0$ since all longitudinal directions coincide at the pole. However, the cylindrical metric assigns a constant norm to $\partial_\theta L$. Consequently, the effective gradient amplification relative to the true spherical geometry scales as

$$\|\nabla_{\text{cyl}} L\| \; \sim \; \frac{1}{\sin \psi} \|\nabla_{\text{sphere}} L\|,$$

which diverges at the poles. This mismatch causes spurious, unbounded gradient updates in the longitudinal direction near $\psi = 0, \pi$, resulting in gradient explosion. $\qquad\square$

### I.3. Topological Mismatch of Modulus Constraints

A naive approach to strictly enforcing polar domains involves applying modulus constraints during optimization, specifically, restricting longitudinal coordinates via modulus operations. However, this imposes a discontinuous "wrapping" operation on the optimization landscape. Consequently, the gradient descent update rule implies a Euclidean step followed by a coordinate reset:

$$\theta_{t+1} \leftarrow (\theta_t - \eta \nabla_\theta \mathcal{L}) \bmod 2\pi \tag{54}$$

$$\psi_{t+1} \leftarrow (\psi_t - \eta \nabla_\psi \mathcal{L}) \bmod \pi \tag{55}$$

where $\eta$ is the learning rate and $\mathcal{L}$ denotes the loss function. While these updates technically bound the parameters within angular domains, the optimization dynamic remains fundamentally Euclidean. The algorithm traverses a flat, periodic hyper-rectangle rather than respecting the continuous curvature of the hyperspherical manifold $\mathbb{S}^{d-1}$.

## J. Proof of Theorems

In this section, we present proofs for all theorems, propositions and corollaries. We state each theorem before elucidating the proof.

### J.1. Proof of Proposition 2.1

**Theorem J.1.** *Let $\mathcal{M} = (\mathbb{S}^{d-1}, g)$ be the Riemannian manifold embedded in $\mathbb{R}^d$, defined by the constraint set $\mathbb{S}^{d-1} = \{\mathbf{z} \in \mathbb{R}^d : \|\mathbf{z}\|_2 = 1\}$, where $g$ denotes the canonical round metric induced by the Euclidean inner product $\langle \cdot, \cdot \rangle_{\mathbb{R}^d}$. The geodesic distance $d_\mathcal{M} : \mathcal{M} \times \mathcal{M} \to \mathbb{R}_{\geq 0}$ between any two points $\mathbf{z}_i, \mathbf{z}_j \in \mathcal{M}$ is given by:*

$$d_\mathcal{M}(\mathbf{z}_i, \mathbf{z}_j) = \arccos(\langle \mathbf{z}_i, \mathbf{z}_j \rangle). \tag{56}$$

*Thus, the optimization of intrinsic geometric relationships on $\mathcal{M}$ is isometric to the optimization of cosine similarities in the ambient space $\mathbb{R}^d$ subject to $\|\mathbf{z}\|_2 = 1$.*

*Proof.* Consider two vectors $\mathbf{z}_i, \mathbf{z}_j$, such that, $\|\mathbf{z}_i\| = \|\mathbf{z}_j\| = 1$. We know that $\mathbf{z}_i, \mathbf{z}_j$ and the origin span a two-dimensional plane passing through the origin. The intersection of the plane and the hypersphere is a standard circle of radius $r = 1$ known as the Great Circle. The shortest distance between two points on a sphere always lies along the Great Circle connecting them. Let $\theta$ be the angle subtended by $\mathbf{z}_i$ and $\mathbf{z}_j$ at the origin. Now, on a circle of radius $r$, the length of the arc $L$ is given by,

$$L = r\theta \tag{57}$$

but, we know that $r = 1$, therefore,

$$L = \theta \tag{58}$$

Now, the inner product of two vectors in $\mathbb{R}^d$ is given by,

$$\langle \mathbf{z}_i, \mathbf{z}_j \rangle = \|\mathbf{z}_i\| \|\mathbf{z}_j\| \cos \theta \tag{59}$$

Since $\|\mathbf{z}_i\| = \|\mathbf{z}_j\| = 1$, this simplifies to:

$$\langle \mathbf{z}_i, \mathbf{z}_j \rangle = \cos \theta \tag{60}$$

$$\theta = \arccos(\langle \mathbf{z}_i, \mathbf{z}_j \rangle) \tag{61}$$

From (58), we can conclude that,

$$d_{\mathcal{M}} = \theta = \arccos(\langle \mathbf{z}_i, \mathbf{z}_j \rangle). \tag{62}$$

$\square$

### J.2. Proof of Theorem 2.3

**Theorem J.2.** *Let $\mathbf{z}$ be a random vector distributed uniformly on the unit hypersphere $\mathbb{S}^{d-1}$ equipped with the uniform surface probability measure $\sigma$. For any fixed reference axis $\mathbf{u} \in \mathbb{S}^{d-1}$ (e.g., the North pole), let $h(\mathbf{z}) = \langle \mathbf{z}, \mathbf{u} \rangle$ be the projection of $\mathbf{z}$ onto that axis.*

*For any $\epsilon > 0$, the probability that $\mathbf{z}$ resides in the "polar cap" defined by a distance $\epsilon$ from the equator is bounded by:*

$$\sigma\left(\{\mathbf{z} \in \mathbb{S}^{d-1} : |\langle \mathbf{z}, \mathbf{u} \rangle| \geq \epsilon\}\right) \leq 2 \exp\left(-\frac{d\epsilon^2}{2}\right) \tag{63}$$

*Proof.* We can use the concentration of measure in higher-dimensional statistics to prove this theorem. Consider a fixed reference axis $\mathbf{u} \in \mathbb{S}^{d-1}$ where $h(\mathbf{z}) = \langle \mathbf{z}, \mathbf{u} \rangle$ is the projection of $z$ onto the reference axis. Let the measure to be found be $\sigma(|h(\mathbf{z})| > \epsilon)$ for any $\epsilon > 0$. Consider slicing the sphere $\mathbb{S}^{d-1}$ with hyperplanes perpendicular to the axis of $h(\mathbf{z})$ at some value $t$. The intersection of the sphere $\mathbb{S}^{d-2}$ and the plane $z_1 = t$ is a lower dimensional sphere $\mathbb{S}^{d-2}$ with radius $r = \sqrt{1 - t^2}$. The surface area $A$ of the slice has the following relation,

$$A(t) \propto (\sqrt{1 - t^2})^{d-3} \tag{64}$$

Therefore, the probability density for the coordinate $h(z)$ is proportional to the area defined in (64) as follows,

$$P(t) \propto (1 - t^2)^{\frac{d-3}{2}} \tag{65}$$

The detailed derivation of (65) by change of variables is given in Appendix S. The probability of falling in the polar cap $t \geq \epsilon$ is the ratio of the area of the cap to the total area.

$$P(t \geq \epsilon) = \frac{\int_\epsilon^1 (1 - t^2)^{\frac{d-3}{2}} dt}{\int_{-1}^1 (1 - t^2)^{\frac{d-3}{2}} dt} \tag{66}$$

Now, we have the inequality $1 - x \leq e^{-x}$, which can be written as $1 - t^2 \leq e^{-t^2}$ by taking $x = t^2$ for our convenience. Further, for large $d$, we can approximate $\frac{d-3}{2}$ as $\frac{d}{2}$. Therefore, we can write,

$$(1 - t^2)^{\frac{d}{2}} \leq e^{-\frac{dt^2}{2}} \tag{67}$$

So, now we have the following bound,

$$P(t \geq \epsilon) = \frac{\int_\epsilon^1 \exp(\frac{-dt^2}{2}) dt}{\int_{-1}^1 \exp(\frac{-dt^2}{2}) dt} \tag{68}$$

We can express the above integral using the error function

$$\int \exp(\frac{-dt^2}{2}) dt = \sqrt{\frac{\pi}{2d}} \operatorname{erf}\left(t\sqrt{\frac{d}{2}}\right) \tag{69}$$

We derive (69) in appendix S.2. Therefore, the integral now becomes,

$$P(t \geq \varepsilon) = \frac{\operatorname{erf}\left(\sqrt{\frac{d}{2}}\right) - \operatorname{erf}\left(\varepsilon\sqrt{\frac{d}{2}}\right)}{2\operatorname{erf}\left(\sqrt{\frac{d}{2}}\right)} \tag{70}$$

As $d$ becomes large, $\operatorname{erf}(\sqrt{\frac{d}{2}}) \to 1$

$$P(t \geq \epsilon) \approx \frac{1}{2}[1 - \operatorname{erf}(\epsilon\sqrt{\frac{d}{2}})] \tag{71}$$

We know,

$$\operatorname{erfc}(x) = 1 - \operatorname{erf}(x) \tag{72}$$

Therefore,

$$P(t \geq \epsilon) \approx \frac{1}{2}\operatorname{erfc}(\epsilon\sqrt{\frac{d}{2}}) \tag{73}$$

Now, for large $x$, $\operatorname{erfc}(x) \leq \exp(-x^2)$, therefore, we can obtain,

$$P(t \geq \epsilon) \leq \exp(\frac{-d\epsilon^2}{2}) \tag{74}$$

On accounting for both the North pole cap and South pole cap, i.e $P(h(\mathbf{z}) \geq \epsilon)$ and $P(h(\mathbf{z}) \leq -\epsilon)$, since the distribution is symmetric, we can finally say that,

$$P(|h(\mathbf{z})| \geq \epsilon) \leq 2\exp\left(-\frac{d\epsilon^2}{2}\right) \tag{75}$$

$\square$

### J.3. Proof of Theorem 2.4

**Theorem J.3.** *Let the uniform distribution on $\mathbb{S}^{d-1}$ be generated by normalizing a standard multivariate Gaussian vector $\mathbf{x} \sim \mathcal{N}(\mathbf{0}, \mathbf{I}_d)$, such that $\mathbf{z} = \mathbf{x}/\|\mathbf{x}\|_2$. Let $\mathbf{u} \in \mathbb{S}^{d-1}$ be any fixed reference axis (e.g., the polar axis).*

*By the Weak Law of Large Numbers (WLLN), as the dimension $d \to \infty$, the projection of a random embedding $\mathbf{z}$ onto $\mathbf{u}$ converges in probability to zero:*

$$\langle \mathbf{z}, \mathbf{u} \rangle \xrightarrow{P} 0 \tag{76}$$

*Proof.* Let $\mathbf{x} \sim \mathcal{N}(\mathbf{0}, \mathbf{I}_d)$ be a standard multivariate Gaussian vector where $x_1, \ldots, x_d$ are independent and identically distributed (i.i.d.) with $x_i \sim \mathcal{N}(0, 1)$. The random vector on the sphere is given by the normalization $\mathbf{z} = \frac{\mathbf{x}}{\|\mathbf{x}\|_2}$.

Let $\mathbf{u} \in \mathbb{S}^{d-1}$ be an arbitrary fixed unit vector. We seek to analyze the convergence of the projection $Y_d = \langle \mathbf{z}, \mathbf{u} \rangle$. We can express this projection as:

$$\langle \mathbf{z}, \mathbf{u} \rangle = \frac{\langle \mathbf{x}, \mathbf{u} \rangle}{\|\mathbf{x}\|_2} = \frac{\sum_{i=1}^{d} x_i u_i}{\sqrt{\sum_{i=1}^{d} x_i^2}}. \tag{77}$$

Consider the numerator term $N = \langle \mathbf{x}, \mathbf{u} \rangle = \sum_{i=1}^{d} u_i x_i$. Since the $x_i$ are independent Gaussian variables, their linear combination $N$ is also a Gaussian random variable. We calculate the mean and variance of $N$:

$$\mathbb{E}[N] = \sum_{i=1}^{d} u_i \mathbb{E}[x_i] = 0,$$

$$\operatorname{Var}(N) = \sum_{i=1}^{d} u_i^2 \operatorname{Var}(x_i) = \sum_{i=1}^{d} u_i^2(1) = \|\mathbf{u}\|_2^2.$$

Since $\mathbf{u}$ is a unit vector, $\|\mathbf{u}\|_2^2 = 1$. Therefore, regardless of the dimension $d$, the numerator is distributed as a standard normal variable:

$$N \sim \mathcal{N}(0, 1).$$

Thus, the numerator is bounded in probability (it is $O_p(1)$).

Consider the squared norm in the denominator, denoted as $D^2 = \|\mathbf{x}\|_2^2 = \sum_{i=1}^{d} x_i^2$. The terms $x_i^2$ are i.i.d. variables following a Chi-squared distribution with 1 degree of freedom, having expected value $\mathbb{E}[x_i^2] = 1$.

We apply the Weak Law of Large Numbers (WLLN) to the average of these terms. As $d \to \infty$:

$$\frac{1}{d} \sum_{i=1}^{d} x_i^2 \xrightarrow{P} \mathbb{E}[x_1^2] = 1.$$

By the Continuous Mapping Theorem, taking the square root gives:

$$\sqrt{\frac{1}{d} \|\mathbf{x}\|_2^2} \xrightarrow{P} \sqrt{1} = 1.$$

We can rewrite the original expression for the projection by dividing both the numerator and the denominator by $\sqrt{d}$:

$$\langle \mathbf{z}, \mathbf{u} \rangle = \frac{N}{\sqrt{d} \cdot \sqrt{\frac{1}{d} \|\mathbf{x}\|_2^2}}.$$

As $d \to \infty$:

1. The term $\frac{N}{\sqrt{d}}$ converges to 0 in probability because $N$ is a finite random variable ($\mathcal{N}(0, 1)$) and $\sqrt{d} \to \infty$.

2. The term $\sqrt{\frac{1}{d} \|\mathbf{x}\|_2^2}$ converges to 1 in probability.

Finally, by Slutsky's Theorem, the product of these converging terms is:

$$\langle \mathbf{z}, \mathbf{u} \rangle = \left( \frac{N}{\sqrt{d}} \right) \cdot \left( \frac{1}{\sqrt{\frac{1}{d} \|\mathbf{x}\|_2^2}} \right) \xrightarrow{P} 0 \cdot \frac{1}{1} = 0.$$

Thus, the projection converges in probability to zero. $\qquad\square$

## K. Additional Discussion on SVGD

As discussed in section 2.3.2, the behavior of the regularization is governed by the topology of the target score function, $s(\mathbf{z}) = \nabla_z \log p(\mathbf{z})$. This score function is interpreted as a superposition of physical forces designed to perform work against the inherent entropic barriers of the hypersphere. We define the effective potential energy of a concept embedding space $\mathbf{z}$ as the negative log likelihood of the target distribution,

$$U_{\text{total}}(\mathbf{z}) = -\log p(\mathbf{z}) = \underbrace{-\log p_{\text{struct}}(\mathbf{z})}_{U_{\text{polar}}} + \underbrace{-\log p_{\text{align}}(\mathbf{z})}_{U_{\text{semantic}}} \tag{78}$$

Gradient descent on this landscape subjects the particle to a conservative force $\mathbf{F} = -\nabla U_{\text{total}}(\mathbf{z})$. The equilibrium state of the system is determined by the interplay of these potentials against the curvature of the manifold.

### K.1. Structural Drift

The structural component of the score function is derived from a prior favoring the poles $\mathbf{p}$. We know that the surface area element of a thin sheet that slices a sphere scales as,

$$d\text{Area}(\psi) \propto \sin^{d-2}(\psi) d\psi$$

From theorem 2.3, we can establish that for large $d$, $\sin^{d-2}(\psi)$ becomes an extremely sharp peak at the equator. Therefore, if points are sampled randomly, they will never be at the poles, i.e, high/low hierarchy levels. Now, to represent hierarchy effectively, we need the distribution of an embedding $\mathbf{z}$ occupying a longitudinal coordinate $\theta$ to be Uniform. Therefore, we must introduce a prior $p_{\text{struct}}(\mathbf{z})$ that cancels out the geometric concentration factor $\sin^{d-2}\psi$. So, the target prior density must be inversely proportional to the surface area element,

$$p_{\text{struct}}(\mathbf{z}) \propto \frac{1}{\sin^{d-2}(\psi)}$$

Now, we derive the structural score for this prior. Along the polar axis, $z_d = \cos \psi$. Note that $\psi$ denotes a derived polar angle with respect to a chosen structural axis, not one of the hyperspherical chart variables $\psi_1, \psi_2, ...\psi_{d-2}$. Therefore, we can obtain,

$$\sin \psi = \sqrt{1 - z_d^2}$$

Substituting the above expression in the ratio,

$$p_{\text{struct}}(z_d) \propto (1 - z_d^2)^{\frac{-k}{2}}$$

where $k$ is a scaling factor. On taking the log likelihood,

$$\log p_{\text{struct}}(z_d) = -\frac{k}{2} \log(1 - z_d^2) + C \tag{79}$$

On taking the derivative of the log likelihood with respect to $z_d$,

$$\frac{\partial}{\partial z_d} \log p_{\text{struct}}(z_d) = -\frac{k}{2} \cdot \frac{1}{1 - z_d^2} \cdot (-2z_d) = k \cdot \frac{z_d}{1 - z_d^2} \tag{80}$$

In our implementation, we absorb $k$ into the kernel weights and add $\epsilon$ to the denominator for numerical stability. This yields,

$$\nabla_{z_d} \log p_{\text{struct}} = \frac{z_d}{1 - z_d^2 + \epsilon} \tag{81}$$

Since we align the $\mathbf{p}_N$ with the basis vector,

$$\mathbf{e}_d = \begin{bmatrix} 0 & \cdots & 0 & 1 \end{bmatrix}^\top$$

we obtain the final gradient vector as,

$$\nabla_z \log p_{\text{struct}}(\mathbf{z})^\top = \begin{bmatrix} 0 & \cdots & 0 & \dfrac{z_d}{1 - z_d^2 + \epsilon} \end{bmatrix} \tag{82}$$

This gradient acts as a Pole Attracting Force Field. At the Northern Hemisphere, $z_d > 0$, it pushes the particles closer to the North pole while at the Southern Hemisphere, $z_d < 0$, it pushes the particles closer to the South pole.

### K.2. Semantic Alignment

The alignment component corresponds to the vMF likelihood centered at the pretrained embeddings $\boldsymbol{\mu}$.

$$\mathbf{F}_{\text{sem}} = \nabla \mathbf{z}(\kappa_{\text{align}} \langle \mathbf{z}, \boldsymbol{\mu} \rangle) = \kappa_{\text{align}} \boldsymbol{\mu} \tag{83}$$

This acts as a spring connecting the learnable embedding $\mathbf{z}$ to its semantic anchor $\boldsymbol{\mu}$. As the structural gradient pulls $\mathbf{z}$ towards the pole to satisfy hierarchy constraints, the spring stretches. The force $\mathbf{F}_{\text{sem}}$ exerts a restoring pull to preserve semantic fidelity. This term sculpts local minima into the global gravitational landscape, ensuring that while the constellation of points drift towards the Poles, individual points maintain their relative constellations.

## L. Derivation of KL Divergence of vMF Distributions

In this section, we derive the KL divergence between two vMF distributions. The PDF for a vMF distribution with mean direction $\boldsymbol{\mu}$ and concentration $\kappa$ is,

$$f(\mathbf{x}|\boldsymbol{\mu}, \kappa) = C_d(\kappa) \exp\big(\kappa \langle \mathbf{x}, \boldsymbol{\mu} \rangle\big) \tag{84}$$

where $\mathbf{x} \in \mathbb{S}^{d-1}$ is a unit vector and $C_d(\kappa)$ is a normalization constant. Let the child distribution be denoted as $f_c$ and the parent distribution be denoted as $f_p$. Both the distributions are parameterized by $\mu$ and $\kappa$ as $\mu_c, \kappa_c$ and $\mu_p, \kappa_p$ respectively.

$$D_{KL}(f_c \| f_p) = \int_{\mathbb{S}^{d-1}} f_c(\mathbf{x}) \log \frac{f_c(\mathbf{x})}{f_p(\mathbf{x})} \, d\mathbf{x} = \mathbb{E}_{\mathbf{x} \sim f_c}[\log f_c(\mathbf{x}) - \log f_p(\mathbf{x})] \tag{85}$$

Now, we can expand the terms as,

$$\log f_c(\mathbf{x}) = \log C_d(\kappa_c) + \kappa_c \langle \mathbf{x}, \mu_c \rangle \tag{86}$$

$$\log f_p(\mathbf{x}) = \log C_d(\kappa_p) + \kappa_p \langle \mathbf{x}, \mu_p \rangle \tag{87}$$

Subtracting (86) and (87), we get,

$$\log f_c(\mathbf{x}) - \log f_p(\mathbf{x}) = \log C_d(\kappa_c) - \log C_d(\kappa_p) + \langle \mathbf{x}, \kappa_c \mu_c - \kappa_p \mu_p \rangle \tag{88}$$

On taking the expectation of the above expression, we get,

$$D_{KL} = \log C_d(\kappa_c) - \log C_d(\kappa_p) + \langle \mathbb{E}_{\mathbf{x} \sim f_c}[\mathbf{x}], \kappa_c \mu_c - \kappa_p \mu_p \rangle \tag{89}$$

The $\mathbb{E}[\mathbf{x}]$ on the unit hypersphere $\mathbb{S}^{d-1}$ is given by,

$$\mathbb{E}[\mathbf{x}] = \mathcal{A}_d(\kappa)\mu \tag{90}$$

where,

$$\mathcal{A}_d(\kappa) = \frac{I_{\frac{d}{2}}(\kappa)}{I_{\frac{d}{2}-1}(\kappa)} \tag{91}$$

Substituting (90) in (89), we get,

$$D_{KL}(f_c \| f_p) = \log C_d(\kappa_c) - \log C_d(\kappa_p) + \mathcal{A}_d(\kappa_c) \langle \mu_c, \kappa_c \mu_c - \kappa_p \mu_p \rangle \tag{92}$$

Since $\mu_c$ is a unit vector, $\langle \mu_c, \mu_c \rangle = 1$, which simplifies the expression to:

$$D_{KL}(f_c \| f_p) = \log C_d(\kappa_c) - \log C_d(\kappa_p) + \mathcal{A}_d(\kappa_c)(\kappa_c - \kappa_p \langle \mu_c, \mu_p \rangle) \tag{93}$$

## M. Algorithm for Training and Coupled Orbital Inference

Algorithm 1 details the training process step by step and algorithm 2 details the inference process. We calculate each node's radius based on its depth and the number of descendants, which is later used to rank candidates.

## N. Performance on the Full CUB-200-2011 Dataset

As described in Section B, we evaluate multimodal performance using an image-to-label ranking formulation. To assess scalability and robustness beyond the reduced experimental setting, we additionally report results on the complete CUB-200-2011 dataset. Table 13 summarizes the corresponding performance comparisons. The results demonstrate that `Polaris` consistently outperforms strong geometric and hyperbolic baselines across all evaluation metrics on the full dataset. In particular, the improvements in Precision@1 and MRR indicate superior retrieval quality and ranking consistency, while the substantially lower MR reflects more accurate hierarchical alignment overall.

---

**Algorithm 1** Training Procedure for `Polaris`

---

**Require:** Training taxonomy $\mathcal{T}$, encoder $f_\theta$, batch size $B$, learning rate $\eta$, SVGD temperature $\kappa$, margins $\gamma_{\text{geom}}, \gamma_{\text{prob}}$
1: Initialize encoder parameters $\theta$
2: Initialize spherical projection network $f_{\text{sphere}}$
3: Initialize probabilistic heads $\Theta_\mu, w_\kappa, b_\kappa$
4: **for** each training iteration **do**
5:     Sample mini-batch $\mathcal{B} = \{(p_i, c_i, n_i)\}_{i=1}^B$ of parent–child–negative triplets
6:     **for** each triplet $(p_i, c_i, n_i) \in \mathcal{B}$ **do**
7:         Encode latent representations: $h_p, h_c, h_n \leftarrow f_\theta(p_i), f_\theta(c_i), f_\theta(n_i)$
8:         Project each of $h \in \{h_p, h_c, h_n\}$ onto the tangent space at the North pole: $u \leftarrow h - \langle h, \mathbf{p}_N \rangle \mathbf{p}_N$
9:         Map to hypersphere using exponential map: $y \leftarrow \exp_{\mathbf{p}_N}(u)$
10:        Apply spherical projection network and normalize: $z \leftarrow f_{\text{sphere}}(y)$
11:        Compute geodesic angular distances: $\theta_{pc} \leftarrow \arccos(\langle z_p, z_c \rangle), \quad \theta_{nc} \leftarrow \arccos(\langle z_n, z_c \rangle)$
12:        Compute robust geometric loss: $\mathcal{L}_{\text{geom}} \leftarrow \max\left(0, \gamma_{\text{geom}} + \mathcal{W}(\theta_{pc}) - \mathcal{W}(\theta_{nc})\right)$
13:        Predict vMF distribution parameters: $\boldsymbol{\mu} \leftarrow f_{\text{sphere}}(z; \Theta_{\boldsymbol{\mu}}), \quad \kappa \leftarrow \text{Softplus}(w_\kappa^\top z + b_\kappa)$
14:        Compute probabilistic containment loss: $\mathcal{L}_{\text{prob}} \leftarrow \max\left(0, \gamma_{\text{prob}} + D_{\text{KL}}(\text{vMF}_c \,\|\, \text{vMF}_p) - D_{\text{KL}}(\text{vMF}_c \,\|\, \text{vMF}_n)\right)$
15:     **end for**
16:     Compute SVGD transport field over batch embeddings: $\phi(z_i) \leftarrow \frac{1}{B} \sum_{j=1}^B \left[ k(z_j, z_i) \nabla_{z_j} \log p(z_j) + \nabla_{z_j} k(z_j, z_i) \right]$
17:     Project SVGD updates onto tangent space: $\phi^\top(z_i) \leftarrow \phi(z_i) - \langle \phi(z_i), z_i \rangle z_i$
18:     Compute SVGD loss over the batch: $\mathcal{L}_{\text{svgd}} \leftarrow \frac{1}{B} \sum_{i=1}^B \|\phi^\top(\mathbf{z}_i)\|_2$
19:     Compute total objective: $\mathcal{L} \leftarrow \lambda_{\text{geo}} \mathcal{L}_{\text{geom}} + \lambda_{\text{prob}} \mathcal{L}_{\text{prob}} + \lambda_{\text{svgd}} \mathcal{L}_{\text{svgd}}$
20:     Compute Riemannian gradients on the hypersphere
21:     Update parameters using Riemannian Adam: $z^{(t+1)} \leftarrow \exp_{z^{(t)}}\left(-\eta \frac{m_t}{\sqrt{v_t} + \epsilon}\right)$
22: **end for**
23: **Return** Trained Embedding Model `Polaris`

---

**Algorithm 2** Coupled Orbital Inference

---

1: **Input:** Query encoder $f_q$, Candidate set $\mathcal{C}$, Depth Map $D(\cdot)$, Descendant Map $N(\cdot)$, Margin $\epsilon$, Strength $\gamma$
2: **Output:** Ranked List of Children
3: Let $\mathbf{z}_q \leftarrow f_q(\text{query})$ projected to $\mathbb{S}^{d-1}$
4: **Define Radius:** $R(node) \leftarrow D(node) + 1 + \frac{\log(1 + N(node))}{\log 2}$
5: $r_q \leftarrow 1 - \text{Normalize}(R(\text{query}))$
6: **for each** candidate $c \in \mathcal{C}$ **do**
7:     $\mathbf{z}_c \leftarrow \text{Embedding}(c)$
8:     $r_c \leftarrow 1 - \text{Normalize}(R(c))$
9:     Compute Intrinsic Similarity:
10:     $\theta_{qc} \leftarrow \arccos(\langle \mathbf{z}_q, \mathbf{z}_c \rangle)$
11:     $\mathcal{S}_{\text{ang}} \leftarrow \cos(\theta_{qc})$ {Cosine similarity}
12:     Compute Radial Coupling:
13:     $\Delta r \leftarrow |r_q - r_c|$
14:     $\tau_{\text{thresh}} \leftarrow 1 - \gamma (\Delta r)^2$ {Dynamic Decision Boundary}
15:     Apply Coupled Gate:
16:     **if** $\mathcal{S}_{\text{ang}} > \tau_{\text{thresh}}$ **then**
17:         $\text{Score}_c \leftarrow \mathcal{S}_{\text{ang}}$
18:     **else**
19:         $\text{Score}_c \leftarrow 0$ {Outside valid orbital aperture}
20:     **end if**
21: **end for**
22: **Return** Top-$k(\{\text{Score}_c\}_{c \in \mathcal{C}})$

---

## O. Hyperparameter Analysis and Reproducibility

In this section, we provide an analysis on key hyperparameters, namely, embedding dimension $d$ and Welsch loss scale parameter $c$. Also, table 14 reports the hyperparameter configurations employed for all datasets on which `Polaris` was evaluated. These hyperparameters were used for most of the experiments conducted to validate `Polaris`.

*Table 13.* **Results on the full CUB-200-2011 dataset.** First ( ), Second ( ), and Third ( ) best models are highlighted. Performance is reported as $mean^{std}$.

| Method | Precision@1 ($\uparrow$) | MR ($\downarrow$) | MRR ($\uparrow$) |
|---|---|---|---|
| CLIP-2 | $10.2^{1.5}$ | $35.4^{3.2}$ | $21.5^{1.4}$ |
| HyperExpan | $24.5^{1.8}$ | $14.1^{2.1}$ | $39.7^{1.7}$ |
| Box | $31.8^{2.3}$ | $17.6^{1.5}$ | $41.2^{2.0}$ |
| Gumbel Box | $35.2^{2.1}$ | $15.4^{1.2}$ | $45.1^{1.8}$ |
| **Polaris** | $\mathbf{44.2^{1.2}}$ | $\mathbf{8.9^{1.0}}$ | $\mathbf{57.2^{0.3}}$ |

*Table 14.* **Dataset specific hyperparameter configurations used in Polaris.** Most hyperparameters remain stable across datasets, indicating consistent optimization behavior and low sensitivity to extensive tuning.

| Hyperparameter | Science | Environment | WordNet | MeSH | Verb | Food | Birds |
|---|---|---|---|---|---|---|---|
| *Architecture* | | | | | | | |
| Embedding dimension $d$ | 128 | 64 | 128 | 128 | 256 | 128 | 64 |
| Projection hidden size | 64 | 64 | 64 | 64 | 64 | 64 | 64 |
| Projection depth | 2 | 2 | 2 | 2 | 2 | 2 | 2 |
| *Optimization* | | | | | | | |
| PLM learning rate | $9 \times 10^{-5}$ | $9 \times 10^{-5}$ | $9 \times 10^{-5}$ | $9 \times 10^{-5}$ | $9 \times 10^{-5}$ | $9 \times 10^{-5}$ | – |
| Projection learning rate | $1 \times 10^{-3}$ | $1 \times 10^{-3}$ | $1 \times 10^{-3}$ | $1 \times 10^{-3}$ | $1 \times 10^{-3}$ | $1 \times 10^{-3}$ | $1 \times 10^{-3}$ |
| Number of epochs | 50 | 50 | 30 | 75 | 75 | 50 | 50 |
| Gradient accumulation steps | 3 | 3 | 5 | 5 | 5 | 5 | 3 |
| Batch size | 64 | 64 | 128 | 128 | 128 | 64 | 32 |
| Negative samples | 50 | 50 | 10 | 20 | 20 | 20 | 5 |
| *Probabilistic Alignment* | | | | | | | |
| vMF margin | 0.3 | 0.3 | 0.3 | 0.3 | 0.3 | 0.3 | 0.3 |
| Probabilistic weight | 0.3 | 0.3 | 0.3 | 0.3 | 0.3 | 0.3 | 0.3 |
| *SVGD Dynamics* | | | | | | | |
| $\kappa_{align}$ | 1.0 | 1.0 | 1.0 | 1.0 | 1.0 | 1.0 | 1.0 |
| $\kappa_{repel}$ | 2.0 | 2.0 | 2.0 | 4.5 | 4.5 | 2.0 | 4.5 |
| SVGD weight | 0.1 | 0.1 | 0.1 | 0.1 | 0.1 | 0.1 | 0.1 |
| *Geometric Regularization* | | | | | | | |
| Geometric margin | 0.5 | 0.5 | 0.5 | 0.5 | 0.5 | 0.5 | 0.5 |
| Geometric weight | 0.7 | 0.7 | 0.7 | 0.7 | 0.7 | 0.7 | 0.7 |
| Welsch loss scale $c$ | 0.7 | 0.4 | 0.4 | 0.4 | 0.4 | 0.4 | 0.4 |

### O.1. Embedding Dimension

We conduct an analysis on the number of dimensions used to represent **z** on Science and Environment as shown in figure 4a. For both datasets, performance peaks at $d = 64$ to $d = 128$ and degrades at higher dimensions. This indicates that the coupled orbital geometry captures hierarchical relationships without requiring very high dimensional latent spaces.

### O.2. Welsch Loss Parameter $c$

We conduct experiments on the Welsch loss parameter $c$ as shown in figure 4b. The parameter $c$ controls the transition of the loss from quadratic to saturated behavior: for small residuals, the Welsch loss approximates the Mean Squared Error (MSE), while for large residuals the penalty saturates, effectively downweighting outliers. Therefore, $c$ acts as a robustness threshold that balances sensitivity to local geometric misalignment against stability under large structural deviations. As shown in ablations for science and environment, moderate values of $c$ consistently yield improved performance across datasets, whereas excessively small or large values degrade performance due to over-robustification and insufficient noise suppression, respectively. Thereby a value between 0.4 and 0.7 is ideal for parameter $c$.

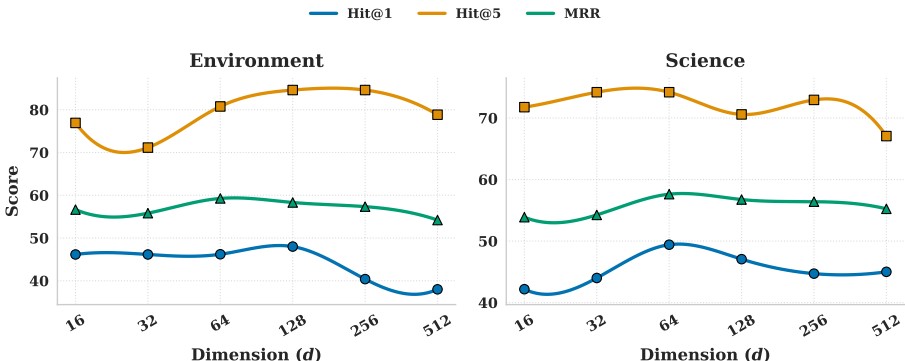

*(a)* **Embedding dimension sensitivity.** Performance improves with larger latent dimensionality before gradually saturating at higher dimensions, indicating stable scaling behavior across datasets.

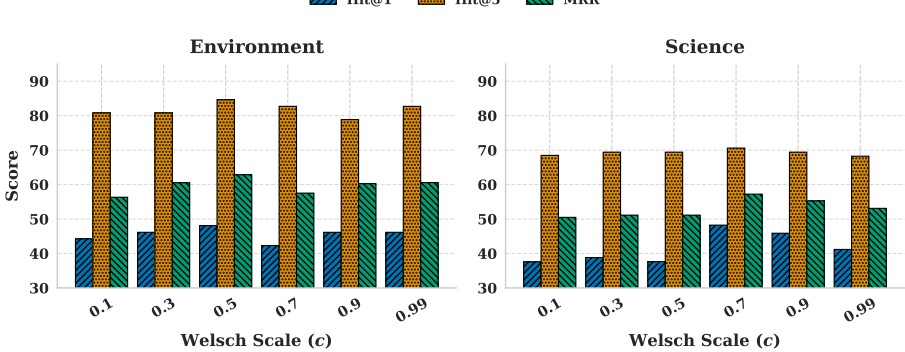

*(b)* **Sensitivity to Welsch loss scale.** Moderate Welsch scale values consistently yield the strongest performance, while degradation away from the optimum remains gradual and stable.

*Figure 4.* **Hyperparameter sensitivity analysis.** Influence of embedding dimensionality ($d$) and Welsch loss scale ($c$) on retrieval performance across the Science and Environment datasets.

## P. Robustness to Hierarchy Noise

Since radius $r$ is calculated deterministically, we perform edge removal tests to show the robustness of `Polaris` to incomplete taxonomies. Table 15 demonstrates these results. We notice that performance degrades smoothly as corruption increases, avoiding abrupt collapse. `Polaris` is highly resilient to small perturbations (5% to 15% missing edges) and approaches a stable lower bound under severe sparsity, proving it leverages available structure without becoming brittle.

## Q. Time Complexity Analysis

In this section, we analyze the computational complexity and scalability characteristics of SVGD within `Polaris`İn our implementation, particle interactions are restricted to the mini-batch $B$, yielding a computational complexity of $\mathcal{O}(B^2 d)$, where $d$ denotes the embedding dimension. Since SVGD is applied locally within each batch rather than globally across the dataset, the method remains computationally tractable even at large scales. We further report empirical wall-clock evaluations on MeSH using a batch size of 256. The additional overhead introduced by SVGD is modest, contributing approximately 20 seconds per training epoch relative to the non-SVGD variant. Although `Polaris` requires roughly $1.9\times$ the training time of the fastest baseline, TaxoExpan(Shen et al., 2020b), inference remains highly efficient with a latency of only 92 ms per query, making the framework suitable for real-time deployment scenarios. Overall, the computational overhead introduced by SVGD is small relative to the substantial improvements in geometric stability, hierarchical consistency, and downstream predictive performance. Detailed efficiency comparisons are presented in Table 16.

*Table 15.* **Robustness to structural corruption.** Performance of Polaris under progressive random edge removal. Polaris exhibits smooth degradation under increasing graph corruption, indicating resilience to missing hierarchical relations. Retention denotes performance relative to the uncorrupted graph.

| Removed | Environment | | | WordNet | | |
|---|---|---|---|---|---|---|
| | Hit@1 | Retention | $\Delta \downarrow$ | Hit@1 | Retention | $\Delta \downarrow$ |
| **0%** | **53.0** | **100%** | **0.0** | **26.5** | **100%** | **0.0** |
| 5% | 52.2 | 98.5% | 0.8 | 24.3 | 91.7% | 2.2 |
| 10% | 51.0 | 96.2% | 2.0 | 23.3 | 87.9% | 3.2 |
| 15% | 51.0 | 96.2% | 2.0 | 22.0 | 83.0% | 4.5 |
| 20% | 50.0 | 94.3% | 3.0 | 20.7 | 78.1% | 5.8 |
| 25% | 47.2 | 89.1% | 5.8 | 18.2 | 68.7% | 8.3 |
| 30% | 45.2 | 85.3% | 7.8 | 16.4 | 61.9% | 10.1 |
| 35% | 42.3 | 79.8% | 10.7 | 15.4 | 58.1% | 11.1 |
| 40% | 40.4 | 76.2% | 12.6 | 15.0 | 56.6% | 11.5 |
| 45% | 36.5 | 68.9% | 16.5 | 15.0 | 56.6% | 11.5 |
| **50%** | 32.7 | 61.7% | 20.3 | 12.6 | 47.5% | 13.9 |

*Table 16.* **Computational efficiency comparison.** Polaris maintains competitive training and inference overhead despite its complex geometric objectives.

| Model | Inference Time (ms/query) | Training Time (/Epoch) |
|---|---|---|
| BERT+MLP | 45 | 6m 25s |
| TaxoExpan | 62 | 4m 40s |
| HyperExpan | 78 | 6m 15s |
| Box Embeddings | 75 | 8m 35s |
| ConE | 72 | 6m 50s |
| **Polaris** | **92** | **9m 00s** |
| Polaris w/o SVGD | 92 | 8m 40s |
| Polaris w/o vMF | 92 | 8m 47s |

# R. Statistical Tests

We perform statistical tests to quantify the significance of Polaris's improvements across hierarchical expansion tasks. Table 17 reports aggregated $z$-scores and $p$-values, demonstrating significant gains over baselines for single parent, DAG expansion and multimodal hierarchies with all $p$-values falling below the standard significance threshold of 0.05. This confirms that observed gains are not due to random chance and are statistically significant across all evaluated domains.

# S. Additional Derivations required in Proofs

In this section, we elaborate on derivations that we used in proofs.

*Table 17.* Statistical significance analysis using $z$-test across hierarchical expansion tasks.

| Task | Dataset | $z$ | $p$-**value** |
|---|---|---|---|
| **Single-Parent Taxonomy Expansion** | Science | 9.90 | $7.2e-09$ |
| | WordNet | 8.14 | $2.1e-07$ |
| | Environment | 10.39 | $2.3e-12$ |
| **DAG Expansion** | MeSH | 12.51 | $3.1e-19$ |
| | Verb | 4.26 | $8.9e-03$ |
| | Semeval Food | 3.45 | $3.5e-02$ |
| **Multimodal Hierarchy** | Birds | 3.53 | $9.7e-03$ |

## S.1. Detailed Derivation of Surface Area of Strip by Change of Variables

Equation (65) is obtained by change of variables from angle $\psi$ to linear coordinate $t$. On a unit hypersphere, let $\psi$ be the latitudinal angle ranging from 0 to $\pi$. The surface area of a strip at angle $\psi$ is proportional to,

$$A(\psi) \propto \sin^{d-2}(\psi) \tag{94}$$

Therefore, the probability distribution with respect to the angle is

$$P(\psi)d\psi \propto \sin^{d-2}(\psi)d\psi \tag{95}$$

Let the projection $h(z) = t$. The relationship between the linear coordinate $t$ and $\psi$ is $t = \cos\psi$. Now through change of variables,

$$dt = -\sin\psi\,d\psi \tag{96}$$

$$d\psi = \frac{-dt}{\sin\psi} \tag{97}$$

Since $\sin\psi = \sqrt{1 - \cos^2\psi}$ and $\cos\psi = t$, we obtain,

$$d\psi = \frac{dt}{\sqrt{1 - t^2}} \tag{98}$$

Substituting back in (95), we obtain

$$P(t)dt \propto \frac{(\sqrt{1 - t^2})^{d-2}}{\sqrt{1 - t^2}}dt \tag{99}$$

$$P(t) \propto (\sqrt{1 - t^2})^{d-3} \tag{100}$$

## S.2. Solution of $\int \exp\frac{-dt^2}{2}dt$

To solve this integral, we make use of the error function. The error function is defined as follows,

$$\mathrm{erf}(x) = \frac{2}{\sqrt{\pi}} \int_0^x \exp(-t^2)dt \tag{101}$$

Taking the derivative on both sides,

$$\frac{d}{dx}(\mathrm{erf}(x)) = \frac{2}{\sqrt{\pi}} \exp\left(-x^2\right) \tag{102}$$

Let $I = \int \exp\frac{-dt^2}{2}dt$. Consider the substitution,

$$u = t\sqrt{\frac{d}{2}} \tag{103}$$

$$du = \sqrt{\frac{d}{2}}dt \tag{104}$$

Plugging (104) in the integral $I$, we get,

$$I = \sqrt{\frac{2}{d}} \int \exp{-u^2}du \tag{105}$$

From (102), we can now write $I$ as,

$$I = \sqrt{\frac{\pi}{2d}} \int \frac{d}{du}(\mathrm{erf}(u))du \tag{106}$$

By taking the integral of the derivative, we get,

$$I = \sqrt{\frac{\pi}{2d}}\,\mathrm{erf}(u) \tag{107}$$

On substituting $u$, we finally have,

$$I = \sqrt{\frac{\pi}{2d}}\,\mathrm{erf}\left(t\sqrt{\frac{d}{2}}\right) \tag{108}$$

