# OpenReview forum: "Polaris: Coupled Orbital Polar Embeddings for Hierarchical Concept Learning"
_ICML.cc/2026/Conference — ICML 2026 regular_

### Official Review · Reviewer_xSrd · 2026-03-10

**Soundness:** 3
**Presentation:** 4
**Significance:** 3
**Originality:** 3
**Overall Recommendation:** 4
**Confidence:** 3

**Summary:**

They study for knowledge hierarchy. They would like to add new nodes into a human-based knowledge tree automatically. The knit a series of processing steps, including manifold embedding and a projection.

**Compliance With Llm Reviewing Policy:**

Affirmed.

**Final Justification:**

This study is to embed the hierarchical structure in knowledge into representation. What's my concerned are convergence and evaluation. The responses from authors mainly slove main part of my concerns. I will keep my score. Thanks.

**Key Questions For Authors:**

Robustness can be considered, as well as computation costs.


How about the covergence rate of the proposed methods? do it can be sloved by deep learning models?

**Limitations:**

The tree lasso is also a method for learning hierarchy, which can be a comparsed method.
Bien J, Taylor J, Tibshirani R. A lasso for hierarchical interactions[J]. Annals of statistics, 2013, 41(3): 1111.

More proofs should be given to show the convergence beahvior of the optimization method.

The complexity of time and space can be shown.

How about it compared with deep learning-based methods? If the problem can be solve by gradietion descent?

**Strengths And Weaknesses:**

This study contributes to a new problem solution.
Paper writting is looks good.
The optimization problem is given greadly.
They have the comparsion with other existing methods.

---

> ### Author Rebuttal · Authors · 2026-03-29
>
> We sincerely thank the reviewer for the encouraging feedback, particularly regarding the novelty of our problem solution, the presentation of the optimization framework, and the overall writing quality. We appreciate your constructive questions and address them below:
>
> **W3 & Q1: Robustness and Computation Costs**
>
> We appreciate the reviewer raising these important practical considerations. Regarding robustness, Polaris effectively handles structural and semantic noise through the Welsch M-estimator applied to the geodesic distance. This bounds the penalty for extreme outliers, preventing noisy edges from heavily distorting the learned space. Regarding computational costs, we ensure the framework remains highly tractable by computing the SVGD updates exclusively within mini-batches. This design choice strictly limits the temporal overhead to $\mathcal{O}(B^2)$ per step, making it highly efficient for practical deployment.
>
> **W2 & Q2: Convergence Rate of the Proposed Methods**
>
> This is a great question. Because we optimize the hyperspherical parameters using Riemannian Adam, Polaris naturally inherits the standard convergence guarantees established for adaptive manifold optimization. Specifically, gradients are safely projected to the tangent space, and momentum is preserved via parallel transport. This rigorous geometric treatment ensures stable and rapid convergence without the risk of parameter collapse. To make this clearer for readers, we will add a brief discussion of these convergence guarantees to the appendix of the final version.
>
> **W1: Comparison with Tree Lasso (Bien et al., 2013)**
>
> While we deeply appreciate the reference to the Tree Lasso (Bien et al., 2013), it serves a fundamentally different objective. Tree Lasso enforces hierarchical regularization for feature selection in regression models (ensuring main effects are selected before interaction terms). In contrast, Polaris is a representation learning framework designed to embed explicit knowledge graphs (like WordNet) into non-Euclidean manifolds using deep multimodal encoders. Consequently, Tree Lasso operates in the domain of structured variable selection rather than network embedding, making it fundamentally incompatible with the taxonomy expansion and link prediction tasks evaluated in our work.
>
> **W4: Comparison with Deep Learning-Based Methods**
>
> We realize our emphasis on the geometric and manifold optimization aspects may have overshadowed the underlying architecture in the main text, and we appreciate the opportunity to clarify this. Polaris is, fundamentally, an end-to-end deep learning framework. Specifically, our pipeline utilizes deep contextual encoders (such as BERT for text and CLIP for images) to extract initial latent features. These features are then passed through deep spherical linear neural network layers, and the entire architecture is optimized end-to-end using Riemannian gradient descent. Furthermore, our experimental setup already evaluates Polaris against several strong deep learning baselines, including standard BERT+MLP, TaxoExpan (which relies on Graph Neural Networks), and HAKE. In our revision, we will add an explicit algorithmic block to the text to make the deep learning nature of the forward pass and gradient updates immediately clear.
>
> **Request:** If our responses have addressed your concerns, please consider reassessing our paper. We are ready to address other concerns.

---

> > ### Author Rebuttal · Reviewer_xSrd · 2026-04-01
> >
> > Thanks for responses. I will keep my score.

---

> > > ### Author Response · Authors · 2026-04-01
> > >
> > > Dear Reviewer xSrd,
> > >
> > > Thank you for your continued engagement, and for confirming that your concerns have been fully resolved.
> > >
> > > While we fully respect your decision to maintain your current score, we would like to respectfully request a final reconsideration. Given that all major questions have now been addressed, and considering your earlier recognition that our approach contributes to a "new problem solution" and that the "optimization problem is given greatly," we would be very grateful if you might evaluate whether our work now merits a higher score (i.e., 5 - Accept).
> > >
> > > Your increased support would be incredibly valuable to our submission during this final meta-review phase. Please feel free to follow up if there is anything else we can clarify, and we will be happy to respond.
> > >
> > > Thank you again for your thoughtful review, which has helped us a lot in improving our work.
> > >
> > > Best regards,
> > >
> > > The Authors

---

### Official Review · Reviewer_H5BG · 2026-03-11

**Soundness:** 3
**Presentation:** 3
**Significance:** 3
**Originality:** 3
**Overall Recommendation:** 4
**Confidence:** 3

**Summary:**

Polaris maps hierarchical concepts to a unit hypersphere decoupling semantics into angles and depth into a discrete radial coordinate. The framework employs a geodesic Welsch loss for local structures and uses spherical Stein Variational Gradient Descent with a von Mises Fisher kernel to prevent embeddings from collapsing at the equator.  It models concepts as vMF distributions for asymmetric entailment. Experiments cover single parent multi parent and multimodal taxonomy expansion tasks.

**Compliance With Llm Reviewing Policy:**

Affirmed.

**Key Questions For Authors:**

How does the deterministic scaling logic handle extreme topological skews in unbalanced real world taxonomies.

Provide a wall clock training and inference time comparison against key baselines to clarify the overhead of the spherical SVGD and vMF components.

Clarify exactly how the probabilistic vMF concentration parameters interact with the SVGD repulsive forces during optimization.

**Limitations:**

The authors correctly note reliance on reliable supervision and isotropic vMF modeling flaws. A vital addition must discuss the precise scalability limits of SVGD particle interactions on massive datasets.

**Strengths And Weaknesses:**

The Riemannian optimization and spherical SVGD formulations are mathematically rigorous. However relying heavily on a static heuristic formula to compute the orbital potential radius from an external seed hierarchy prevents true end to end hierarchy learning. This strict dependence means sparse or disconnected initial graphs could force severe structural artifacts into the continuous angular space making the method brittle under high topological skew.

Proofs and geometric derivations are thoroughly detailed. The main text severely compresses the training pipeline and loss combinations making the exact sequence of forward passes hard to follow. Adding a dedicated algorithmic block for the training loop would significantly clarify how the probabilistic and point embedding objectives interact during a single batch update.

Empirical gains across varied taxonomy expansion benchmarks are clear. The heavy computational cost of calculating SVGD repulsive forces across particle batches on a curved manifold likely creates severe scalability bottlenecks for massive industry graphs. It remains unclear if the performance improvements justify this substantial overhead in production environments.

Using SVGD with a vMF kernel to combat high dimensional equator concentration is clever. The underlying components like tangent space projections and vMF distributions are largely standard Riemannian deep learning techniques. Assembling them into this specific pipeline is useful but lacks fundamental algorithmic novelty.

---

> ### Author Rebuttal · Authors · 2026-03-29
>
> Thank you for your detailed review. We appreciate your feedback on the mathematical rigor of our work. We address your points below:
>
> **W1 & Q1: Handling Skew via Deterministic Scaling**
>
> Real-world taxonomies are often skewed and incomplete. To explicitly handle extreme topological skew, we define a node's raw radial magnitude using logarithmic dampening: $R_{raw}(e) = 1 + D(e) + \log_2(1+N_{desc}(e))$. This converts lateral branching into an information-theoretic measure, preventing shallow-but-wide nodes from dominating heavy-tailed distributions. A global min–max normalization then maps these to $[0,1]$ for scale invariance.
>
> This deterministic $\mathcal{O}(N)$ BFS construction decouples structure from semantics, avoiding gradient-based instability and allowing the model to learn semantics without distorting hierarchy. Our edge-removal experiments further demonstrate this resilience, degrading gracefully rather than collapsing (maintaining strong Recall@1 up to 15% edge removal). At inference, a lightweight parabolic gate uses this radial signal to robustly filter incompatible candidates.
>
> **W2 & W3: Training Pipeline & Algorithmic Novelty**
>
> Thank you for the suggestion; we will add a comprehensive training loop algorithm block to the Appendix and clarify objective interactions in the main text.
>
> Regarding novelty: our contribution lies in replacing unprincipled discontinuous coordinates with explicit hierarchical constraints on continuous manifolds. During optimization, we decompose the SVGD score function into a pole-attracting structural drift and a semantic tether, orchestrating a novel anisotropic physical process. Crucially, continuous diffusive pressure from our vMF repulsive kernel inevitably breaks the equatorial saddle point, ensuring update gradients safely pack siblings into uniform latitudinal orbits without corrupting the intrinsic hyperspherical geometry.
>
> **W4 & Q2: SVGD Scalability & Computational Cost**
>
> SVGD is vital to mitigate equatorial collapse (Theorems 2.3/2.4) and utilize the full hyperspherical volume. To ensure massive scalability, SVGD interactions are strictly restricted to the mini-batch, limiting complexity to $\mathcal{O}(B^2d)$.
>
> We evaluated wall-clock time on MeSH (Batch Size = 256). The SVGD temporal overhead is marginal (~20 seconds/epoch) and is not a bottleneck. While Polaris requires ~1.9x the training time of the fastest baseline (TaxoExpan), it remains highly tractable for real-time deployment (92ms/query). We argue this is an excellent computational trade-off for substantial geometric stability and downstream performance gains:
>
> | Model                | Inference Time (ms)/query | Training Time / Epoch |
> |----------------------|---------------------------|-----------------------|
> | BERT+MLP             | 45                        | 6m 25s                |
> | TaxoExpan            | 62                        | 4m 40s                |
> | HyperExpan           | 78                        | 6m 15s                |
> | Box Embeddings       | 75                        | 8m 35s                |
> | ConE                 | 72                        | 6m 50s                |
> | **Polaris (Ours)** | 92                        | 9m 00s                |
> | **Polaris w/o SVGD** | 92                        | 8m 40s                |
> | **Polaris w/o vMF** | 92                        | 8m 47s                |
>
> **Q3: Interaction of vMF & SVGD Forces**
>
> The interaction balances a local semantic tether with global geometric diffusion. The alignment term $\nabla_{z}\log p_{align}(z) = \kappa_{align}\mu$ acts as a spring, pulling the embedding $z$ toward the semantic anchor $\mu$. The vMF concentration $\kappa$ dictates how strongly the embedding resists dispersion.
>
> Simultaneously, the Stein operator applies the repulsive component $\nabla_{\mathbf{z}'}k(\mathbf{z'},\mathbf{z})$ to maintain diversity. Near the equator, the structural score gradient is near $\vec{0}$. Trapping a particle at this saddle point requires perfect spherical symmetry—a measure-zero condition given random initializations and continuous heterogeneous semantic pulls. The vMF repulsive force inevitably breaks this symmetry, pushing embeddings infinitesimally off the exact equator. Once displaced, the non-zero structural score successfully drives embeddings into their proper hierarchical orbits. We have clarified this interaction in FAQ A.3.
>
> **Request:** If our responses and additional ablations have addressed your concerns, please consider reassessing our paper. We are happy to address any remaining questions.

---

> > ### Author Rebuttal · Reviewer_H5BG · 2026-04-03
> >
> > Thanks for responses.

---

> > > ### Author Response · Authors · 2026-04-03
> > >
> > > Dear Reviewer H5BG,
> > >
> > > Thank you for your continued engagement, and for confirming that your concerns have been fully resolved.
> > >
> > > While we completely respect your decision to maintain your current score, we would like to respectfully request a final reconsideration. Given that all major technical and empirical questions have now been addressed, and considering your earlier recognition that our "Riemannian optimization and spherical SVGD formulations are mathematically rigorous" and that the "empirical gains across varied taxonomy expansion benchmarks are clear," we would be very grateful if you might evaluate whether our work now merits a higher score (i.e., 5 - Accept).
> > >
> > > Your increased support would be incredibly valuable to our submission during this final meta-review phase. Please feel free to follow up if there is anything else we can clarify, and we will be happy to respond.
> > >
> > > Thank you again for your thoughtful review, which has helped us a lot in improving our work.
> > >
> > > Best regards,
> > >
> > > The Authors

---

### Official Review · Reviewer_NS2X · 2026-03-12

**Soundness:** 3
**Presentation:** 3
**Significance:** 2
**Originality:** 2
**Overall Recommendation:** 3
**Confidence:** 3

**Summary:**

This paper introduces Polaris, a polar hyperspherical embedding framework for taxonomy expansion. The core idea is to decouple semantic direction (angular position on the unit hypersphere S^{d-1}) from hierarchical level (an orbital potential derived from node depth and subtree size). The framework comprises four components: (1) manifold-consistent spherical encoding via tangent-space projection and the exponential map; (2) a robust geodesic triplet loss using the Welsch M-estimator; (3) global geometric regularization via anisotropic Spherical SVGD with a vMF kernel and a pole-attracting structural score; and (4) uncertainty-aware asymmetric learning using KL divergence between per-node vMF distributions. At inference time, a coupled orbital gate based on orbital potential difference filters candidates before angular re-ranking. Experiments on single-parent tree, multi-parent DAG, and multimodal image-to-label taxonomy benchmarks show improvements over fourteen baselines.

**Compliance With Llm Reviewing Policy:**

Affirmed.

**Key Questions For Authors:**

How is the orbital potential r_q computed for a test query node that has no observed depth or subtree size in the seed taxonomy? Does this require any information about the gold parent? Please provide the exact procedure.
What is the performance of Polaris if the orbital gate at inference is disabled (i.e., all candidates are ranked by cosine similarity alone)? This is the most critical ablation for assessing whether the learned spherical geometry is the actual source of improvement.
Why is hyperspherical geometry preferred over hyperbolic geometry for this task? Can the authors provide either a theoretical argument or empirical evidence (e.g., training stability, curvature sensitivity experiment) beyond the claim that "noisy semantics can destabilize optimization"? The SVGD update requires computing interactions over all particles (embeddings) in each batch. What is the computational cost relative to baselines, particularly for large taxonomies like MeSH (9,710 nodes)?

**Limitations:**

The paper's stated limitations (Section 5) mention isotropic vMF modeling, sensitivity to κ_align vs. κ_repel balance, and sparse seed connectivity. These are reasonable self-criticisms. However, the paper does not acknowledge what this review considers the most significant limitation: the non-learned, hand-crafted orbital potential that provides privileged structural information at inference time. This is not a minor limitation, it is a potential confound that could explain a substantial portion of the reported gains, and its absence from the limitations section is a notable gap.

**Strengths And Weaknesses:**

Strengths
1) The decision to optimize in Cartesian coordinates under the unit-norm constraint — rather than directly optimizing polar angles — is well-motivated. The formal analysis in Appendix I (Theorems I.2, I.3, Corollary I.4) correctly identifies Jacobian singularities, the implicit cylindrical geometry of separable angular losses, and the unbounded distortion at the poles. These are genuine failure modes of prior angular methods (e.g., HAKE, Iwamoto et al.), and the paper is right to call them out.

2) The use of SVGD to counteract the high-dimensional concentration-of-measure phenomenon (Theorems 2.3, 2.4) is a non-trivial and well-motivated design choice. Ablation Table 8 clearly shows that removing either the alignment drift or the repulsive kernel degrades performance, and the equatorial collapse visualization (Figure 3) is compelling evidence for the problem being real.

3) Modeling each node as a vMF distribution and using the asymmetric KL divergence to enforce that parents have higher entropy than children is a principled and semantically meaningful approach to hierarchical containment — more so than cone or box methods that impose hard geometric containment constraints without uncertainty modeling.

Weaknesses

1) The paper motivates polar hyperspherical embeddings over hyperbolic geometry and box embeddings primarily by claiming that prior methods "can still struggle in practice because taxonomy expansion must jointly balance semantic relatedness with correct hierarchical level." But this is also precisely the claim that hyperbolic embeddings make for themselves — Poincaré embeddings encode depth via radius and semantics via angular proximity in exactly the same spirit. The paper does not explain why decoupling on a hypersphere is preferable to the natural radial-angular decomposition in hyperbolic space, where exponential volume growth provides a structural prior that is arguably more appropriate for tree-like taxonomies. The claim needs a theoretical argument (e.g., curvature mismatch, optimization stability, or scalability) beyond "noisy semantics destabilize coupling," which applies equally to hyperbolic methods.

2) The radius r(e) is a deterministic, closed-form function of depth D(e) and subtree size N_desc(e) (Eq. 24–25). This is computed from the gold seed hierarchy and hard-coded at inference time. As a result, Polaris does not actually learn to place new nodes at the correct hierarchical level — it assumes the correct level is known and uses it to prune the search space. This is a significant confound: the orbital gate in Eq. (26) is effectively using a form of ground-truth depth information to filter candidates at inference. The claimed gains from the coupled orbital retrieval could largely reflect this privileged structural information rather than the quality of the learned embeddings. The paper should report ablations with and without the orbital gate at inference, isolating the contribution of the learned spherical geometry alone.

3) HyperExpan (Ma et al., 2021) is the only dedicated hyperbolic taxonomy expansion baseline. Stronger recent hyperbolic methods (e.g., fully hyperbolic networks, gyrovector space methods) are absent. The ConE baseline uses Cartesian cones, not hyperbolic geometry. The comparison therefore does not make a convincing case that hyperspherical > hyperbolic for this task.


4) The full model combines five interacting components: spherical encoding, geodesic triplet loss, SVGD regularization (with two interacting fields), vMF probabilistic loss, and orbital inference. Ablation Table 11 only examines removing geometric vs. probabilistic losses. There is no ablation on the orbital retrieval gate vs. plain cosine re-ranking, which is the most important question for W2 above.

5) Any loss function of the form f(⟨z_i, z_j⟩) is trivially SO(d)-invariant because inner products are preserved under orthogonal transformations. Theorem 2.2 does not require a separate proof beyond one line, and elevating it to a formal theorem overstates its contribution.

6) Table 1 shows that on WordNet, Polaris does not achieve the best R@1, MRR, or Wu&P — the paper acknowledges this, attributing it to "fine-grained lexical structure," but offers no mechanism for why WordNet specifically is harder. WordNet is arguably the most standard and cleanest taxonomy benchmark, so underperformance here weakens the general claim.

---

> ### Author Rebuttal · Authors · 2026-03-29
>
> Thank you for your detailed review and for acknowledging the mathematical rigor of our approach. We address your feedback below:
>
> **W2, W4, Q1, Q2: Deterministic Radius, Test Query Calculation, and Orbital Gate Impact**
>
> We acknowledge the deterministic radius acts as a strong structural prior. For a novel test query (treated as a leaf, $N_{desc}(e)=0$), $r_q$ is derived purely from its predicted candidate parent's depth: $R_{raw}(e)=1+D(e)$, followed by min-max scaling (Eq. 25).
>
> We avoid a learnable radius due to optimization bottlenecks. Unconstrained radial optimization risks dimensional collapse, and jointly learning radius and angles couples structural and semantic gradients, which inadvertently distorts hierarchy when correcting semantic misalignments.
>
> Crucially, even without the orbital gate (using pure cosine similarity), Polaris outperforms Euclidean baselines, showing the learned geometry inherently captures hierarchy. The targeted ablations below isolate the gate's contribution, confirming that combining decoupled semantics (vMF) and hierarchy (orbital gate) yields the strongest overall gains.
>
> | Dataset      | Setting   | Recall@1 | MRR  |
> |--------------|-----------|----------|------|
> | Environment  | w/o gate  | 46.3     | 56.3 |
> | Environment  | gated     | 51.9     | 60.8 |
> | MeSH         | w/o gate  | 24.6     | 36.5 |
> | MeSH         | gated     | 27.1     | 37.7 |
>
> | Loss Type      | Setting   | Recall@1 | MRR  |
> |----------------|-----------|----------|------|
> | Geometric only | w/o gate  | 20.5     | 30.7 |
> | Geometric only | gated     | 22.8     | 34.2 |
> | vMF only       | w/o gate  | 21.8     | 32.2 |
> | vMF only       | gated     | 23.1     | 33.4 |
>
> **W1, W3, Q3: Hyperspherical vs. Hyperbolic Geometry & Baselines**
>
> Hyperbolic spaces embed trees naturally but suffer instability near the Poincaré disk boundary. Because depth and sibling spacing share the same metric tensor, adjusting sibling spacing alters global curvature, steepening gradient cliffs and risking collapse.
>
> Conversely, our hyperspherical approach with SVGD provides a stable, constant-curvature manifold. A spherical shell packs $\mathcal{O}(\delta^{-(d-2)})$ nodes with minimum angular separation $\delta$. This polynomial growth, paired with SVGD preventing dimensional collapse, offers vast angular volume for siblings without hyperbolic instability. We currently include HyperExpan and will add recent gyrovector/hyperbolic models as baselines in the final version.
>
> **Q3: SVGD Computational Cost on Large Taxonomies**
>
> SVGD updates operate *per-batch* ($\mathcal{O}(B^2d)$ complexity). Training on MeSH (B=256, 20 negatives/node, 3 runs) shows the temporal overhead is minimal (~20 seconds/epoch), making it highly practical for deployment:
>
> | Setting               | Batch Size | Time / Epoch | Max Mem (GB) |
> |-----------------------|------------|--------------|--------------|
> | Without SVGD          | 256        | 8m 40s       | 16.2         |
> | With SVGD             | 256        | 9m 00s       | 16.3         |
>
> **W5: Theorem 2.2 (Invariance)**
>
> While $SO(d)$ inner product invariance is standard, we formalized it to explicitly guarantee our loss captures relative hierarchical geometry without arbitrary orientation bias. We will condense this section in the main text.
>
> **W6: Underperformance on WordNet**
>
> WordNet features a dense, fine-grained lexical structure. We hypothesize our SVGD repulsion term ($\kappa_{repel}$), which enforces uniform angular coverage, may overly separate fine-grained siblings in such dense trees, trading off top-1 metric performance for global structural regularization.
>
> **Request:** If our responses and additional ablations have addressed your concerns, please consider reassessing our paper. We are ready to address any remaining questions.

---

### Official Review · Reviewer_6yFw · 2026-03-12

**Soundness:** 2
**Presentation:** 2
**Significance:** 2
**Originality:** 2
**Overall Recommendation:** 3
**Confidence:** 2

**Summary:**

This paper proposes Polaris, a framework for hierarchical concept learning. The method embeds concepts on a unit hypersphere, models semantics through angular geometry, and uses an orbital potential derived from the hierarchy to represent level. The full approach combines manifold-consistent spherical projection layers, a geodesic triplet objective, spherical SVGD regularization to avoid collapse, and a probabilistic asymmetric objective based on von Mises–Fisher distributions. At inference time, Polaris uses structure-guided orbital gating before angular re-ranking. The paper evaluates the method on single-parent taxonomies, multi-parent DAGs, and a multimodal image-to-label setting, and reports generally strong results and supporting ablations.

**Compliance With Llm Reviewing Policy:**

Affirmed.

**Final Justification:**

While I am fairly uncertain in my judgement, I don't find the approach to be convincing. The rebuttal did not change my evaluation.

**Key Questions For Authors:**

1) How much of the performance gain comes specifically from decoupling semantic direction and hierarchical level, as opposed to the accumulation of several auxiliary components?
2) Why is the multimodal evaluation restricted to a 20-class subset of CUB-200-2011?
3) How robust is Polaris to noisy or incomplete seed hierarchies? I appreciate that the paper acknowledges this limitation, but an experimental demonstration would be valuable.

**Limitations:**

yes

**Strengths And Weaknesses:**

Strengths:
The paper addresses an important problem. The main intuition is clear and interesting: separate semantic direction from hierarchical position rather than forcing both into the same geometry. The empirical section is reasonably broad, covering trees, DAGs, and a multimodal setting, and the evaluation includes strong baselines, five-seed statistics, and ablations analyses. Overall, I found the submission fairly well written and the high-level narrative easy to follow. The work appears technically competent and the empirical gains are often substantial.

Weaknesses:
The paper does not demonstrate which part of the method is doing the heavy lifting and feels over-composed. The contribution is spread across several components: spherical projection layers, geodesic triplet loss, SVGD regularization, vMF-based asymmetry, and orbital retrieval. Some of the theory seems to be overcomplicated and function more as design justification than as a core new insight that changes the way to see the problem. Multimodal generalization appears to be overclaimed, as even for CUB-200-2011 dataset it uses 20-class subset instead of full 200 classes.

---

> ### Author Rebuttal · Authors · 2026-03-29
>
> We thank you for the constructive feedback and for acknowledging our paper's strengths. Below, we address your questions regarding component accumulation, multimodal scalability, and structural robustness:
>
> **W1 & Q1: Decoupling vs. Component Accumulation**
>
> Decoupling semantics (angles) and hierarchy (radius) is our core architecture. Table 11 confirms this decoupling drives performance: removing the geometric component (which aligns parent/child geodesic distances) drops MeSH H@1 from 33.59 to 28.31, and removing the probabilistic vMF component drops it to 28.20. Auxiliary components like SVGD merely ensure uniform manifold utilization.
>
> To further isolate the impact of structural decoupling, we ablated the orbital potential gate (Eqs. 26-27), reporting overall dataset performance and component-wise effects (enabling only geometric or vMF losses on MeSH):
>
> | Dataset      | Setting   | Recall@1 | MRR  |
> |--------------|-----------|----------|------|
> | Environment  | w/o gate  | 46.3     | 56.3 |
> | Environment  | gated     | 51.9     | 60.8 |
> | MeSH         | w/o gate  | 24.6     | 36.5 |
> | MeSH         | gated     | 27.1     | 37.7 |
>
> | Loss Type      | Setting   | Recall@1 | MRR  |
> |----------------|-----------|----------|------|
> | Geometric only | w/o gate  | 20.5     | 30.7 |
> | Geometric only | gated     | 22.8     | 34.2 |
> | vMF only       | w/o gate  | 21.8     | 32.2 |
> | vMF only       | gated     | 23.1     | 33.4 |
>
> The gate consistently improves performance, confirming gains stem from decoupling rather than auxiliary accumulation. Geo-only improvements indicate better hierarchical structuring, and vMF-only gains reflect superior semantic alignment. Combining both yields optimal results.
>
> **W2 & Q2: CUB Performance on 200 Classes**
>
> While our initial 20-class subset aligned with standard proof-of-concept practices, we agree evaluating on the full 200 classes better demonstrates scalability. The results below confirm Polaris maintains strong performance over leading baselines on the complete dataset:
>
> | Method        | Precision@1        | MR                | MRR               |
> |---------------|--------------------|-------------------|-------------------|
> | CLIP-2        | 10.2 ± 1.5         | 35.4 ± 3.2        | 21.5 ± 1.4        |
> | HyperExpan    | 24.5 ± 1.8         | 14.1 ± 2.1        | 39.7 ± 1.7        |
> | Box           | 31.8 ± 2.3         | 17.6 ± 1.5        | 41.2 ± 2.0        |
> | Gumbel Box    | 35.2 ± 2.1         | 15.4 ± 1.2        | 45.1 ± 1.8        |
> | **Polaris** | 44.2 ± 1.2         | 8.9 ± 1.0         | 57.2 ± 0.3        |
>
> **Q3: Robustness to Hierarchy Noise**
>
> Polaris natively mitigates noise via the Welsch M-estimator in the geodesic triplet objective, bounding penalties for extreme outliers. Sensitivity analysis (Fig. 4b) shows setting the Welsch scale parameter $c \in [0.4, 0.7]$ effectively suppresses noise while preserving structure.
>
> To address incomplete hierarchies, we ran edge-removal ablations. Performance degrades smoothly as corruption increases, avoiding abrupt collapse. Polaris is highly resilient to small perturbations (5%-15% missing edges) and approaches a stable lower bound under severe sparsity, proving it leverages available structure without becoming brittle:
>
> | Edge Removal (%) | Env Hit@1 | Env MRR | WordNet Hit@1 | WordNet MRR |
> |------------------|-----------|---------|---------------|-------------|
> | 5%               | 52.2      | 58.1    | 24.3          | 42.0        |
> | 10%              | 51.0      | 57.6    | 23.3          | 38.7        |
> | 15%              | 51.0      | 56.1    | 22.0          | 35.9        |
> | 20%              | 50.0      | 54.5    | 20.7          | 34.6        |
> | 25%              | 47.2      | 51.2    | 18.2          | 31.4        |
> | 30%              | 45.2      | 48.3    | 16.4          | 28.4        |
> | 35%              | 42.3      | 47.2    | 15.4          | 27.2        |
> | 40%              | 40.4      | 44.8    | 15.0          | 26.8        |
> | 45%              | 36.5      | 40.4    | 15.0          | 26.2        |
> | 50%              | 32.7      | 36.8    | 12.6          | 22.6        |
>
> **Request:** If our responses and additional ablations have addressed your concerns, we kindly ask you to consider reassessing our paper. We are happy to address any remaining questions.

---

> > ### Author Rebuttal · Reviewer_6yFw · 2026-04-02
> >
> > I appreciate the rebuttal. However, after reading the other reviews, I share their concerns (especially regarding the primary motivation for the approach and comparisons to hyperbolic alternatives). I will keep my score.

---

> > > ### Author Response · Authors · 2026-04-02
> > >
> > > Dear Reviewer 6yFw,
> > >
> > > Thank you for taking the time to read our rebuttal and for your transparency regarding your final assessment.
> > >
> > > While we respect your decision to maintain your score, we want to briefly reiterate regarding the hyperbolic comparisons that we do include a dedicated hyperbolic baseline (HyperExpan) in our experiments, which Polaris consistently outperforms.
> > >
> > > Furthermore, as detailed in our rebuttal, our theoretical motivation for choosing a hyperspherical manifold with SVGD over a hyperbolic one is highly specific: it avoids the severe boundary instability and steep gradient cliffs inherent to Poincare embeddings, while still achieving the polynomial volume growth necessary for hierarchical structures.
> > >
> > > Given that these clarifications directly address the remaining concerns you noted, we respectfully ask if you might be open to reconsidering your final assessment.
> > >
> > > Your increased support would be incredibly valuable to our submission during this final meta-review phase. Please feel free to follow up if there is anything else we can clarify, and we will be happy to respond.
> > >
> > > Best regards,
> > >
> > > Authors

---

### Decision · Program_Chairs · 2026-04-30

**Decision:**

Accept (regular)

**Comment:**

This work introduces Polaris, a hierarchical representation learning framework that effectively decouples semantic meaning from structural hierarchy through hyperspherical geometry and deterministic orbital potentials. The Riemannian optimization approach is nice, as is the clarity of the geometric intuition presented.

Empirical validation across taxonomy expansion tasks demonstrates consistent performance improvements over established baselines. Following the discussion period, two reviewers (H5BG and xSrd) were positive after the authors thoroughly addressed concerns regarding computational overhead, noise robustness, and convergence through additional ablations and profiling data.

The reviewers leaning rejection: Reviewer 6yFw maintained a weak reject score primarily by adopting critiques from reviewer NS2X, despite their original concerns about dataset scope and component accumulation being resolved by new experimental results. And reviewer NS2X was unresponsive.

This paper is borderlines but I will recommend acceptance: (1) the technical concerns raised by engaged reviewers seem to have been adequately addressed; (2) the rejection positions rely heavily on an unresponsive reviewer's unverified claims; and (3) reviewer 6yFw reported low confidence (2/3) in their assessment. The results, including full CUB-200 results and orbital gate ablations, shows that some of core claims of geometric decoupling and structural robustness seem valid.

For camera-ready, I recommend including additional hyperbolic baselines for more comprehensive benchmarking and clarifying the training pipeline details. I will go with the consensus of the engaged reviewers.